# A prefrontal network model operating near steady and oscillatory states links spike desynchronization and synaptic deficits in schizophrenia

David A Crowe[1], Andrew Willow[1], Rachael K Blackman[2,3,4], Adele L DeNicola[2,4], Matthew V Chafee[2,4,5]*, Bagrat Amirikian[2,4,5]*

[1]Department of Biology, Augsburg University, Minneapolis, United States; [2]Department of Neuroscience, University of Minnesota, Minneapolis, United States; [3]Medical Scientist Training Program (MD/PhD), University of Minnesota, Minneapolis, United States; [4]Brain Sciences Center, VA Medical Center, Minneapolis, United States; [5]Center for Cognitive Sciences, University of Minnesota, Minneapolis, United States

*For correspondence:
chafe001@umn.edu (MVC);
amiri001@umn.edu (BA)

**Competing interest:** The authors declare that no competing interests exist.

**Abstract** Schizophrenia results in part from a failure of prefrontal networks but we lack full understanding of how disruptions at a synaptic level cause failures at the network level. This is a crucial gap in our understanding because it prevents us from discovering how genetic mutations and environmental risks that alter synaptic function cause prefrontal network to fail in schizophrenia. To address that question, we developed a recurrent spiking network model of prefrontal local circuits that can explain the link between NMDAR synaptic and 0-lag spike synchrony deficits we recently observed in a pharmacological monkey model of prefrontal network failure in schizophrenia. We analyze how the balance between AMPA and NMDA components of recurrent excitation and GABA inhibition in the network influence oscillatory spike synchrony to inform the biological data. We show that reducing recurrent NMDAR synaptic currents prevents the network from shifting from a steady to oscillatory state in response to extrinsic inputs such as might occur during behavior. These findings strongly parallel dynamic modulation of 0-lag spike synchrony we observed between neurons in monkey prefrontal cortex during behavior, as well as the suppression of this 0-lag spiking by administration of NMDAR antagonists. As such, our cortical network model provides a plausible mechanism explaining the link between NMDAR synaptic and 0-lag spike synchrony deficits observed in a pharmacological monkey model of prefrontal network failure in schizophrenia.

## Editor's evaluation

This valuable modeling study proposes a local circuit mechanism based on a network of recurrently connected excitatory and inhibitory neurons for the recently reported effect that NMDA receptor antagonists cause a drastic reduction of prefrontal neural synchronization in preparation for motor responses in a cognitive task. This mechanism is convincingly supported by simulations of spiking networks and a thorough analysis of the parameter dependency of network dynamics using mean-field theory. The work will be of general interest to computational neuroscientists, and especially for those interested in computational psychiatry.

**eLife digest** Schizophrenia is a long-term mental health condition that can cause a person to see, hear or believe things that are not real. Although researchers do not fully understand the causes of schizophrenia, it is known to disrupt synapses, which connect neurons in the brain to form circuits that carry out a specific function when activated. This disruption alters the pattern of activity among the neurons, distorting the way that information is processed and leading to symptoms. Development of schizophrenia is thought to be due to interactions between many factors, including genetic makeup, changes in how the brain matures during development, and environmental stress.

Despite animal studies revealing how neural circuits can fail at the level of individual cells, it remains difficult to predict or understand the complex ways that this damage affects advanced brain functions. Previous research in monkeys showed that mimicking schizophrenia using a drug that blocks a particular type of synapse prevented neurons from coordinating their activity. However, this did not address how synaptic and cellular changes lead to disrupted neural circuits.

To better understand this, Crowe et al. developed a computational model of neural circuits to study how they respond to synapse disruption. To replicate the brain, the model consisted of two types of neurons – those that activate connecting cells in response to received signals and those that suppress them. This model could replicate the complex network behavior that causes brain cells to respond to sensory inputs. Increasing the strength of inputs to the network caused it to switch from a state in which the cells fired independently to one where the cells fired at the same time. As was previously seen in monkeys, blocking a particular type of synapse thought to be involved in schizophrenia prevented the cells from coordinating their signaling.

The findings suggest that schizophrenia-causing factors can reduce the ability of neurons to fire at the same instant. Disrupting this process could lead to weaker and fewer synapses forming during brain development or loss of synapses in adults. If that is the case, and scientists can understand how factors combine to trigger this process, the mechanism of coordinated activity failure revealed by the model could help identify treatments that prevent or reverse the synapse disruption seen in schizophrenia.

## Introduction

NMDAR synaptic malfunction has been implicated as causal in schizophrenia (*Fromer et al., 2014*; *Schizophrenia Working Group of the Psychiatric Genomics Consortium, 2014*; *Timms et al., 2013*), and loss of NMDAR synaptic function in prefrontal networks is believed to contribute to cognitive deficits as well as clinical symptoms in the disease (*Goldman-Rakic, 1999*; *Javitt et al., 2012*; *Wang et al., 2013*). However, we do not have a complete understanding of how NMDAR synaptic mechanisms influence neural dynamics in prefrontal networks, nor how the disruption of NMDAR synaptic mechanisms might cause prefrontal networks to malfunction. To address these questions, we recently investigated how blocking NMDAR altered neural dynamics and effective communication between neurons in prefrontal cortex of monkeys performing a cognitive control task measuring deficits in schizophrenia (*Blackman et al., 2013*; *Jones et al., 2010*; *Kummerfeld et al., 2020*; *Zick et al., 2018*). We found that reducing NMDAR synaptic communication reduced the frequency of synchronous ('0-lag') spiking between neurons, as well as effective communication between neurons on timescales consistent with monosynaptic interactions between them (*Kummerfeld et al., 2020*; *Zick et al., 2022*; *Zick et al., 2018*). Whereas these studies suggested that NMDAR synaptic function and spike timing in prefrontal networks were linked, they did not elucidate the circuit mechanisms responsible.

In the current study, we developed a spiking neural network model to understand mechanisms that might mediate the link between NMDAR synaptic malfunction and neural dynamics (reduced 0-lag synchronous spiking) we observed in biological data (*Kummerfeld et al., 2020*; *Zick et al., 2022*; *Zick et al., 2018*). The network is comprised of leaky integrate-and-fire neurons embedded in a sparsely connected recurrent network employing realistic NMDAR, GABAR, and AMPAR mediated synaptic currents. We use network stability and mean field analyses to investigate how the balance between NMDA and AMPA components of recurrent excitatory and GABA inhibitory currents influence regimes of network dynamics and spiking synchrony.

For cortical neurons synchrony can occur naturally due to the local recurrent network connectivity, even when external afferent inputs are entirely uncorrelated. Theoretical studies have shown that such synchrony can arise in randomly connected recurrent networks operating in asynchronous irregular (*Amit, 1989*; *Amit and Brunel, 1997*; *Brunel, 2000*; *Renart et al., 2010*; *van Vreeswijk and Sompolinsky, 1996*; *Vicente et al., 2008*) and synchronous irregular regimes (*Brunel, 2000*; *Brunel and Hakim, 1999*; *Brunel and Wang, 2003*; *Ledoux and Brunel, 2011*). In both regimes, individual neurons fire spikes highly irregularly at low rates, a typical situation in a cortex. The major distinction is that in an asynchronous regime population spike rate is steady in time, whereas in a synchronous regime it becomes oscillatory.

We show that simulated prefrontal networks operating near the boundary between steady (asynchronous irregular) and oscillatory (synchronous irregular) regimes in the synaptic parameter space can explain several key experimental observations. First, such networks achieve biologically realistic stochastic spike trains and firing rates of excitatory and inhibitory neurons in prefrontal cortex. Second, increased extrinsic inputs, such as those that might occur during behavior, shift these networks from a steady to an oscillatory regime that causes the emergence of 0-lag spiking between neurons as they stochastically entrain to oscillatory population activity. Third, and perhaps most importantly, we show that reducing recurrent NMDAR synaptic currents prevents these networks from transitioning into oscillatory activity in response to extrinsic inputs, thereby preventing the emergence of 0-lag spike synchrony. Although prior modeling studies have addressed the relationship between NMDAR function and oscillatory activity in prefrontal networks (*Brunel and Wang, 2003*; *Compte et al., 2000*; *Kirli et al., 2014*; *Wang, 1999*), none account for this range of experimental observations. The current results allow us to establish strong parallels between simulated and biological data, including the emergence of 0-lag synchronous spiking via recurrent synaptic interactions between neurons during behavior, the association between synchronous spiking and oscillatory population activity, as well as their joint dependence on NMDAR synaptic mechanisms, both in our current simulation and in the neural data (*Zick et al., 2018*).

## Results

### Summary of experimental results

In this section, we summarize main experimental findings reported previously by our group (*Zick et al., 2018*). In that study, spike trains of ensembles of single neurons were recorded simultaneously from PFC of monkeys while they performed the dot-pattern expectancy (DPX) task, a task that measures specific deficits in cognitive control in schizophrenia (*Jones et al., 2010*). In the DPX task, the correct response (left or right joystick movement) to a probe stimulus depends on a preceding cue followed by a delay period (Materials and methods).

In the present study, we focus on PFC population spike dynamics recorded in the DPX task under two conditions: drug-naive and drug. The drug naive data were collected before monkeys were administered drug, phencyclidine, which is an NMDA receptor antagonist. *Figure 1* shows the population average pairwise correlation between spike trains of neurons recorded in drug-naive (black) and drug (magenta) conditions. The strength of spike correlation was quantified by the ratio between the observed frequency of synchronous spikes (1ms resolution) and the frequency expected if the spike trains were uncorrelated (we subtracted 1 from this ratio so that correlation value is zero for uncorrelated, positive for correlated, and negative for anticorrelated spike activity, Materials and methods). The frequency of spike synchrony was determined from activity observed during a short (100 ms-long) window that was slid across time of task performance. *Figure 1A* shows that spike synchrony obtained from trials aligned to the cue onset (time 0) remained relatively weak and unchanged during the cue and delay periods, until the probe onset, in both drug-naive and drug conditions. The corresponding population average spike rates during these periods are shown in *Figure 1C*. Because the instant of response after probe presentation varied from trial to trial, to appreciate the time course of synchrony and spike rate after the delay period immediately preceding the response, in *Figure 1B and D* we aligned trials to response time (time 0). It is seen that synchrony started to increase sharply about 200ms before the motor response in the drug-naive condition and reached its peak at the time of the response (*Figure 1B*, black). The spike rate also started to increase before the response but more gradually and starting earlier before the response (*Figure 1D*, black). Both spike synchrony (*Figure 1B*)

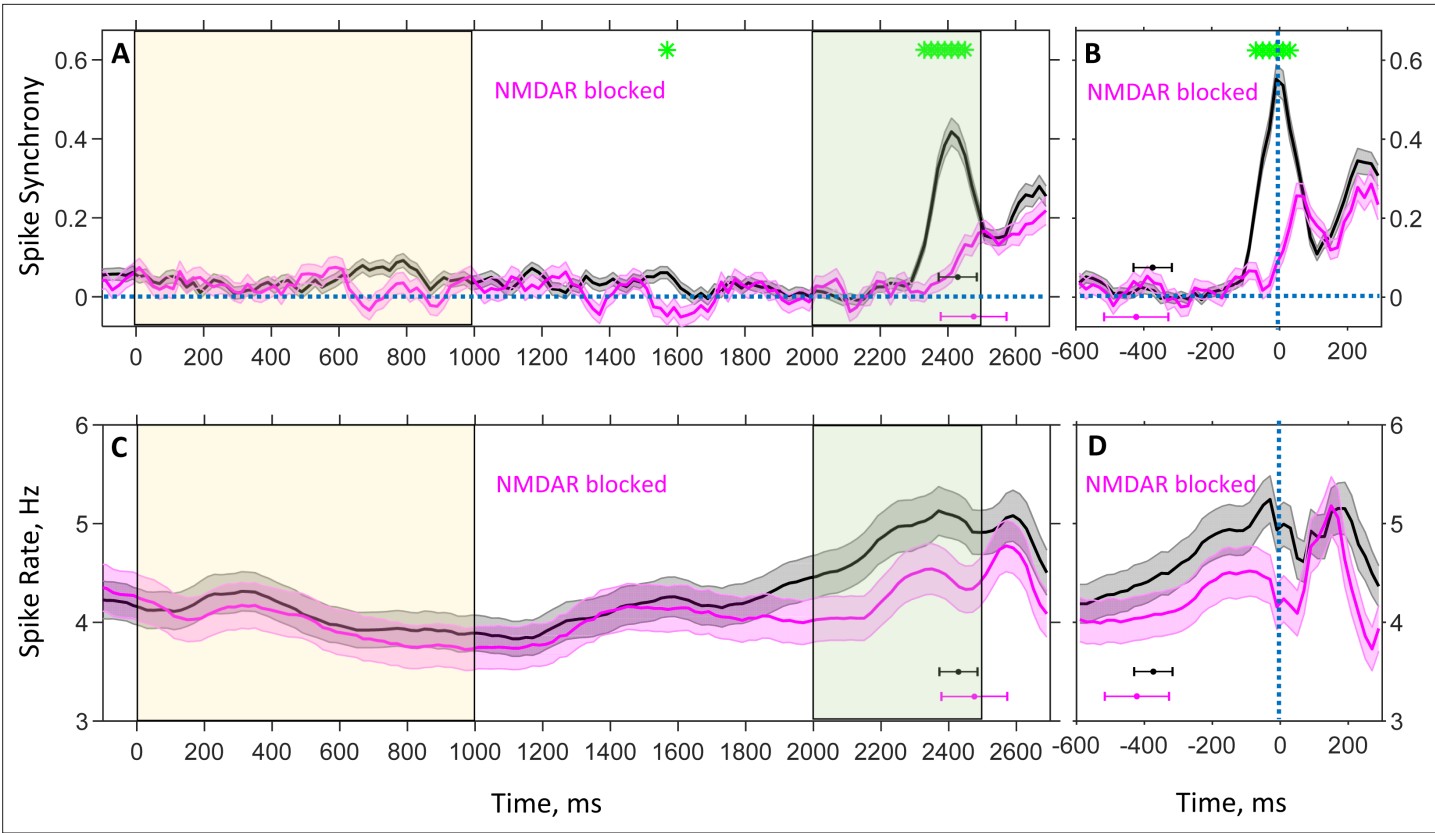

**Figure 1.** Population average spike rate and synchrony between spike trains of neuron pairs recorded during the DPX task as a function of time. Plots show time evolution of spike synchrony (**A, B**) and spike rate (**C, D**) estimated with 100 ms temporal resolution for drug-naive (black) and drug (magenta) conditions. Spike synchrony was measured with 1 ms resolution, and only neuron pairs for which a reasonably reliable estimation of synchrony could be achieved contributed to the plots (see Materials and methods). (**A, C**): Trials are aligned to the cue onset ($t = 0$ ms); in all trials, the cue was presented until $t = 1,000$ ms (yellow shaded area), followed by a 1,000 ms delay period, after which the probe was presented at $t = 2,000$ ms for 500 ms (green shaded area). Color-coded horizontal error-bars indicate the mean and standard deviation of the motor response time for the corresponding drug condition. The numbers of contributing pairs for drug-naive and drug conditions are 524 and 195 (**A**), and the number of neurons are, correspondingly, 514 and 343 (**C**). (**B, D**): Trials are aligned to the time of motor response ($t = 0$ ms) to show the temporal modulation of synchrony and spike rate during the last 600 ms immediately preceding the response. Color-coded horizontal error-bars indicate the mean and standard deviation of the probe presentation time for the corresponding drug condition. The numbers of contributing pairs for drug-naive and drug conditions are 661 and 223 (**B**), and the number of neurons are, correspondingly, 538 and 343 (**D**). Shaded grey and magenta bands show the standard errors for spike synchrony (**A, B**) and rate (**C, D**). Green asterisks show the instances of times when the drug-naive and drug conditions are statistically different (false discovery rate 0.05 [**Benjamini and Hochberg, 1995**] using two-sample *t*-test *p*-values).

and spike rate (***Figure 1D***) exhibited secondary peaks occurring approximately 150–250ms after the response. In the drug condition, however, the increase in spike synchrony at the time of the response was markedly weakened (***Figure 1B***, magenta). The increase in spike rate was also reduced, although less dramatically (***Figure 1D***, magenta). We term this effect as NMDAR blockage induced desynchronization of spiking activity.

## Network model and theoretical framework

To understand the phenomenon of drug-induced desynchronization of spiking activity and the role played by various components of synaptic currents, we considered a spiking network model representing a local circuit of monkey PFC. Details of the model and the theoretical framework are given in Materials and methods. Here, we only highlight their main aspects.

The network comprises excitatory and inhibitory neurons representing populations of pyramidal cells and interneurons, respectively. All neurons are modeled as leaky integrate-and-fire units (see, e.g., ***Dayan and Abbott, 2001***). Synaptic connections are random and sparse, but the number of connections received by individual neurons is large. In addition to the recurrent local connections,

each neuron also receives external connections from excitatory neurons outside of the network that fire spikes with rate $\nu_X$.

Recurrent synaptic currents of excitatory connections are two-component, mediated by AMPA and NMDA receptors, whereas currents of inhibitory connections are mediated by GABA$_A$ receptors (GABA thereafter). External currents represent the noisy inputs due to the background synaptic activity and are mediated by AMPA receptors. Thus, the model entails eight maximal synaptic conductance parameters $g_{X,\alpha}$, $g_{AMPA,\alpha}$, $g_{NMDA,\alpha}$, $g_{GABA,\alpha}$ corresponding to the external AMPA, recurrent AMPA, NMDA, and GABA currents ($\alpha = E, I$ for excitatory and inhibitory neurons, respectively).

To produce a desired regime of network dynamics (asynchronous or synchronous) with a given firing rate of excitatory and inhibitory neurons $\nu_E$ and $\nu_I$, respectively, the values of the conductance parameters should be properly adjusted. For this purpose, we used mean field analysis. In this framework, population mean firing rates $v_E^0$ and $v_I^0$ in the asynchronous stationary state of the network can be effectively parametrized by three parameters expressed as ratios of component synaptic currents: $I_{AMPA}/I_{GABA}$, $I_{NMDA}/I_{GABA}$, and $I_{X,E}/I_{\theta,E}$, where $I_R$ is the mean current of the $R$-receptor mediated synapse ($R = X$, AMPA, NMDA, GABA), and $I_{\theta,E}$ is the current that is needed for an excitatory neuron to reach firing threshold $\theta$ in absence of recurrent feedback. These parameters characterize the balance between recurrent excitation and inhibition, and the balance between external input and firing threshold. Once they are specified, for a given external spike rate $\nu_X$ one can solve the mean field equations to obtain the underlying eight synaptic conductances providing the desired population mean firing rates $v_E^0$ and $v_I^0$ in asynchronous state of the network.

While the mean field analysis allows us to determine synaptic conductances that achieve desired firing rates of neurons, whether these rates remain stable over time is another issue. To address it, we conduct a linear stability analysis of the asynchronous state to understand if the network develops oscillatory instability caused by small fluctuations in population firing rates. This analysis entails two parameters, $\lambda$ and $\omega$, describing the rate of instability growth and the oscillation frequency. The asynchronous state is stable when $\lambda < 0$; in this case small perturbations of firing rates cause exponentially damped oscillation of network activity. The case $\lambda = 0$ corresponds to the onset of instability of the asynchronous state and the emergence of sustained sinusoidal oscillations of population average firing rates with frequency $\omega$; in the oscillatory regime spike trains remain sparse and irregular but at each oscillation cycle a random subset of network neurons fire synchronously giving rise to the synchronous irregular state. Lastly, when $\lambda > 0$, small fluctuations in the stationary rates develop

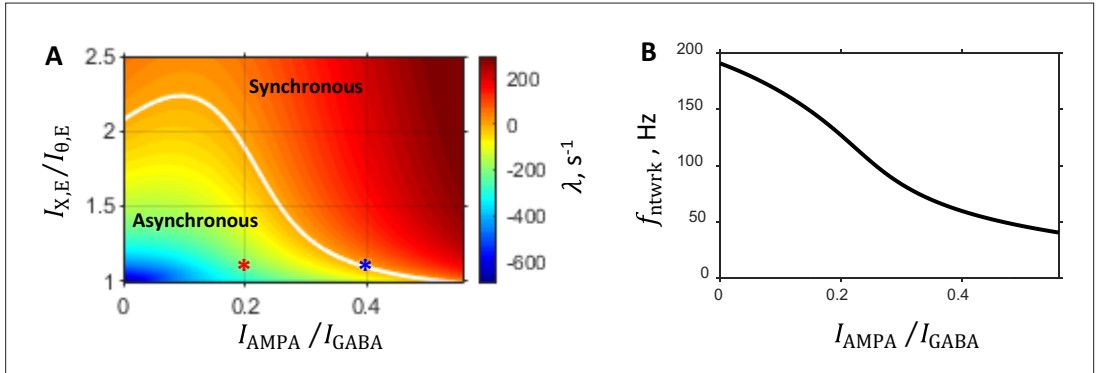

**Figure 2.** Characteristics of the system predicted by the linear stability analysis. Parameters are as follows: prescribed firing rates of excitatory and inhibitory populations are 5 Hz and 20 Hz, respectively; external input spike rate is 5 Hz; and the balance between NMDA and GABA currents is fixed at 0.15. (**A:**) State diagram in the $\left(I_{AMPA}/I_{GABA}, I_{X,E}/I_{\theta,E}\right)$ parameter plane showing color coded value of the rate of instability growth $\lambda$: in the region of the parameter space where $\lambda < 0$ the asynchronous state is stable, whereas the region where $\lambda > 0$ corresponds to the synchronous oscillation state. The two regimes are separated by a critical line on which $\lambda = 0$. This boundary, shown by a white line, is the locus where the stationary network dynamic becomes unstable, and oscillatory population activity develops. Each point in this parameter plane corresponds to a network with a specific set of eight synaptic conductances provided by the mean field approximation. Red and blue asterisks are the points in the state diagram corresponding to the steady and critical primary networks, respectively (see Selection of Primary Networks in Results). (**B:**) Network oscillation frequency that develops on the critical line as a function of the balance between AMPA component of recurrent excitation and inhibition.

The online version of this article includes the following figure supplement(s) for figure 2:

**Figure supplement 1.** Dependence of the characteristic features of the network on the balance between the NMDA and GABA currents.

oscillatory instability with the amplitude of oscillations growing exponentially in time; however, higher order terms neglected in linear analysis can eventually saturate the instability growth (**Brunel and Hakim, 1999**), resulting in a stable oscillation with a finite amplitude.

To examine the boundary between the regions of asynchronous and synchronous states, we fix the balance of tonic NMDA current relative to GABA current, $I_{NMDA}/I_{GABA}$, and vary the remaining two parameters: the balance between recurrent excitation and inhibition, $I_{AMPA}/I_{GABA}$, and the balance between external excitation and firing threshold, $I_{X,E}/I_{\theta,E}$. For a given point in this $\left(I_{AMPA}/I_{GABA}, I_{X,E}/I_{\theta,E}\right)$ parameter plane we solve the mean field equations to find the underlying set of eight synaptic conductances that provide the prescribed rates $v_E^0$ and $v_I^0$ given external spike rate $\nu_X$, and then carry out linear stability analysis to find out if these rates are stable. **Figure 2A** shows a state diagram of the system for which external spike rate is set to $\nu_X = 5$ Hz, the rates of excitatory and inhibitory populations are set to $v_E^0 = 5$ Hz, $v_I^0 = 20$ Hz, and the NMDA current balance is fixed at $I_{NMDA}/I_{GABA} = 0.15$. The diagram shows solutions for $\lambda$ obtained from the linear stability analysis in the $\left(I_{AMPA}/I_{GABA}, I_{X,E}/I_{\theta,E}\right)$ parameter space. The asynchronous stationary state corresponds to the region where $\lambda < 0$, whereas the synchronous oscillation state is realized in the region where $\lambda > 0$. The asynchronous and synchronous states are separated by a "critical" or instability line on which $\lambda = 0$ (shown in white color in **Figure 2A**). This boundary is the locus where the stationary network dynamics becomes unstable, and the sinusoidal oscillation of network activity develops. The oscillation frequency on the critical line, $f_{ntwrk} = \omega/2\pi$, as a function of the balance between the recurrent AMPA and GABA currents, $I_{AMPA}/I_{GABA}$, is shown in **Figure 2B**.

The characteristic features of the state diagram qualitatively remain unchanged when the balance between the NMDA and GABA currents is varied (**Figure 2—figure supplement 1A**). Furthermore, the network frequency at the onset of oscillation, $f_{ntwrk}$, essentially is independent of the $I_{NMDA}/I_{GABA}$ balance (**Figure 2—figure supplement 1B**).

## Integration of DPX task context and drug condition into the model

To study spike synchrony in asynchronous and synchronous networks in the context of the DPX task performed in drug-naive and drug conditions (**Zick et al., 2018**), we make two assumptions regarding neural and synaptic activity: (1) the increase in spike synchronization observed before the monkey's response in **Zick et al., 2018** is due to task-specific external afferent signals received by PFC neurons after probe presentation; (2) administration of NMDAR antagonist results in blocking NMDAR mediated synaptic currents. In the framework of our model, we implemented these assumptions as follows: task specific external signals were accounted for by an increase in the external spike rate from its background level $\nu_X$, whereas the effect of drug administration was modeled by setting NMDAR conductances $g_{NMDA,E}$ and $g_{NMDA,I}$ to zero.

Next, to investigate how spike synchrony in asynchronous and synchronous networks depends on the modulations of $v_X$ and $g_{NMDA,\alpha}$, for each network regime we proceed with the following three steps. First, we choose proper values for conductances, so that the underlying network operates in a desired regime providing the prescribed population firing rates $v_E^*$ and $v_I^*$ for a given external spike rate $v_X^*$. We shall designate this network as the *primary network* relating to the underlying regime and distinguish the corresponding values of all its parameters by the asterisk (*). Second, we carry out a series of network simulations, in which external spike rate $v_X$ and NMDAR conductance $g_{NMDA,\alpha}$ are varied relative to their standard values $v_X^*$ and $g_{NMDA,\alpha}^*$, respectively. Lastly, for each simulated network, we compute population average pairwise correlation between spike trains of neurons and analyze how this correlation depends on the external spike rate and NMDAR conductance.

## Selection of primary networks

To perform a comparison between the primary networks, we need to choose appropriate values for their parameters. We begin with the parameters that are common to both networks. First, we set the excitatory and inhibitory population mean firing rates to $v_E^* = 5$ Hz and $v_I^* = 20$ Hz, respectively, which are on the order of magnitude of spontaneous rates observed for PFC neurons. Second, since external inputs represent activity of excitatory neurons outside the PFC circuit model, we choose the background external rate $v_X^*$ to be the same as the excitatory population rate $v_E^*$ inside the model and, thus, set $v_X^* = 5$ Hz. Lastly, for both networks, we fix the balance between NMDA and GABA currents at $I_{NMDA}^*/I_{GABA}^* = 0.15$. Note that the state diagram in the $\left(I_{AMPA}/I_{GABA}, I_{X,E}/I_{\theta,E}\right)$ space

shown in *Figure 2A* was obtained exactly for these values of the above listed parameters. We use this state diagram for selecting the primary networks and determining the remaining parameters that are network specific.

In this regard, we note that each point in the $\left(I_{\text{AMPA}}/I_{\text{GABA}}, I_{\text{X,E}}/I_{\theta,\text{E}}\right)$ plane corresponds to a network with a specific set of synaptic conductances. For synchronous regime, we look for a network on the critical line ($\lambda = 0$, white line in *Figure 2A*), at the onset of oscillatory instability with a frequency in the $\gamma$-band (a frequency band associated with the LFPs recorded from prefrontal areas [*Bastos et al., 2018*; *Lundqvist et al., 2016*]). For instance, the point marked by a blue asterisk in *Figure 2A* located at $\left(I^*_{\text{AMPA}}/I^*_{\text{GABA}} = 0.4, I^*_{\text{X,E}}/I^*_{\theta,\text{E}} = 1.09\right)$ corresponds to such a network with oscillation frequency $f^*_{\text{ntwrk}} \sim 58$ Hz (*Figure 2B*). In the following, we refer to this network as the *critical state primary network*.

Correspondingly, for the asynchronous regime, we need to select a network that is far from the critical line and deep in the region of stable network dynamics ($\lambda < 0$). The point marked by a red asterisk in *Figure 2A* located at $\left(I^*_{\text{AMPA}}/I^*_{\text{GABA}} = 0.2, I^*_{\text{X,E}}/I^*_{\theta,\text{E}} = 1.09\right)$ is an example of such a network. We shall refer to this network as the *steady state primary network*. For each primary network, we obtain the underlying set of eight synaptic conductance parameters $g^*_{\text{GABA},\alpha}, g^*_{\text{NMDA},\alpha}, g^*_{\text{AMPA},\alpha}, g^*_{\text{xAMPA},\alpha}$ ($\alpha = \text{E}, \text{I}$) by numerically solving the mean field equations.

## Correlation of spiking activity and synchrony in the asynchronous and synchronous states

To investigate characteristic features of spiking dynamics in asynchronous and synchronous regimes, we carried out direct simulations of the primary networks. Both networks comprise $N = 5,000$ neurons, of which $N_{\text{E}} = 4,000$ are excitatory and $N_{\text{I}} = 1,000$ inhibitory. Neurons are connected randomly with a probability $p = 0.2$. *Figure 3* illustrate the behavior of simulated networks with synaptic conductance parameters corresponding to the steady and critical primary networks indicated by the red and blue asterisks, respectively, in the state diagram presented in *Figure 2A*. The dynamic behavior is shown at the level of individual cell activity (spike rasters, top of panels in *Figure 3*), as well as whole population activity (bottom of panels in *Figure 3*).

In simulations shown in *Figure 3* panels A1 and A2 external spike rate $\nu_{\text{X}}$ was fixed at the level of $\nu^*_{\text{X}} = 5$ Hz chosen for the primary networks. It is seen that excitatory and inhibitory neurons exhibit highly irregular firing with average rates, $\nu_{\text{E}}$ and $\nu_{\text{I}}$, about 5.2 Hz and 20 Hz in the steady state primary network (*Figure 3A1*) and 5.5 Hz and 21 Hz in the critical state primary network (*Figure 3A2*). These observed in simulations rates $\nu_{\text{E}}$ and $\nu_{\text{I}}$ are in good agreement with the prescribed rates $\nu^*_{\text{E}} = 5$ Hz and $\nu^*_{\text{I}} = 20$ Hz that were used to derive the synaptic conductance parameters of the simulated networks. Moreover, *Figure 3A1* demonstrates that population activity of the steady state primary network is rather stationary in time, whereas activity of the critical primary network shown in *Figure 3A2* exhibits signs of developing of oscillatory instability (compare *Figure 3—figure supplement 1A1* vs *Figure 3—figure supplement 1A2*). Thus, spiking dynamics observed in the simulated steady state primary network displays basic characteristics of the asynchronous regime—irregular firing of individual neurons and stationary population activity. Correspondingly, the behavior of the simulated critical state primary network exhibits similarity with the boundary regime on which the asynchronous stationary state destabilizes and oscillatory behavior of the population activity emerges.

Panels B1 and B2 in *Figure 3* demonstrate results of simulations in which external spike rate $\nu_{\text{X}}$ was increased by 5% relative to the rate $\nu^*_{\text{X}}$ used in simulations illustrated in *Figure 3* panels A1 and A2. For the steady state primary network (*Figure 3B1*), the firing rates of excitatory and inhibitory neurons increase with the external drive. However, the regime of network dynamics qualitatively does not change and remains asynchronous (compare *Figure 3—figure supplement 1A1* vs *Figure 3—figure supplement 1B1*). In contrast, stronger external inputs received by the critical state primary network synchronize population activity (*Figure 3B2*). It is seen that while individual neurons continue to fire irregularly, population activity now clearly exhibits oscillatory behavior, indicating that the network is in synchronous irregular regime in which the average firing frequency of neurons is low, about 20 Hz, compared to the frequency of network oscillation, which is about 50 Hz (see *Figure 3—figure supplement 1B2*). This frequency is close to the theoretically predicted network frequency of 58 Hz near the onset of oscillation.

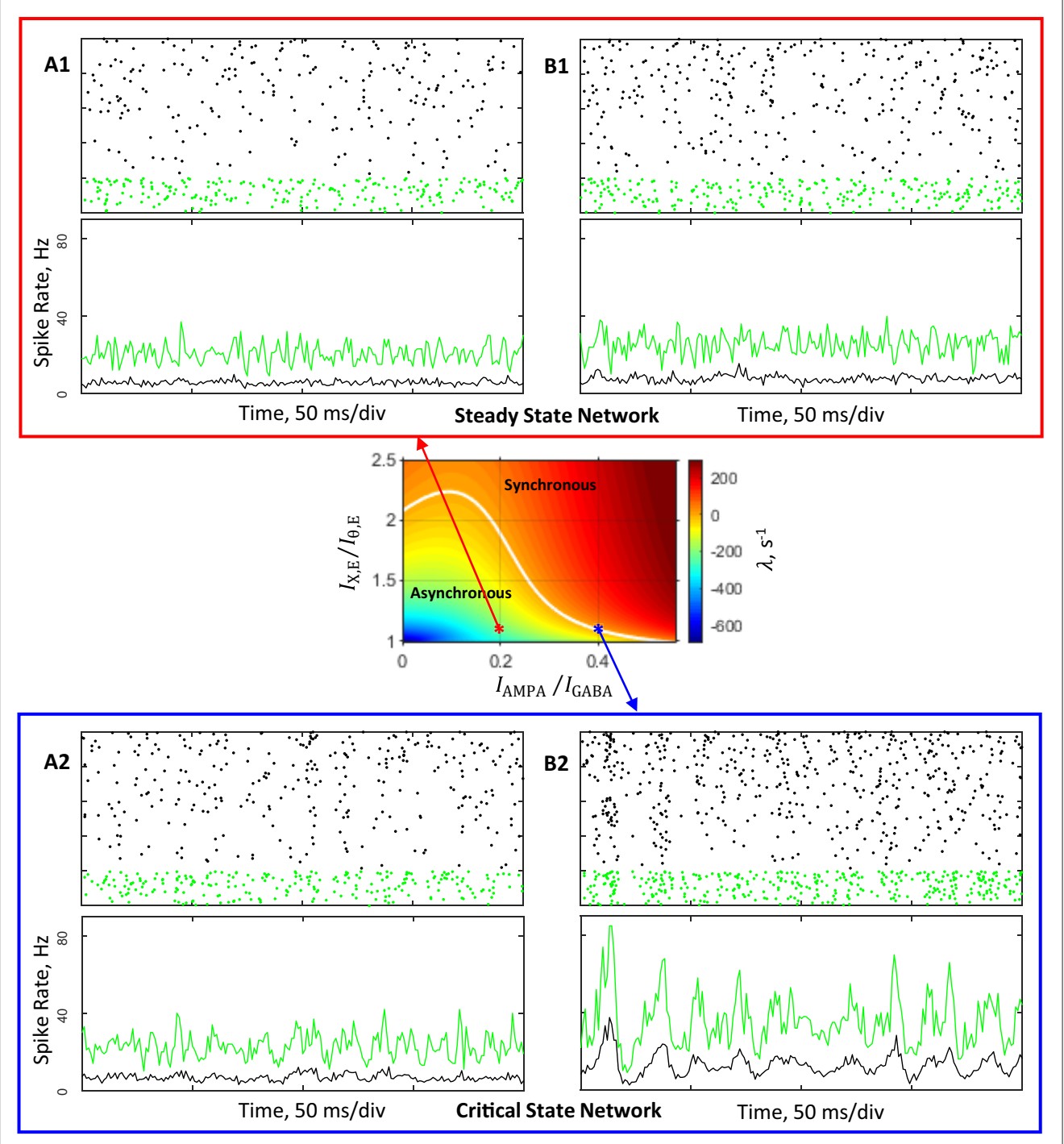

**Figure 3.** Simulations of networks composed of 4,000 excitatory and 1,000 inhibitory neurons connected randomly with probability 0.2. Conductance parameters are solutions of mean field equations for the steady state primary network (**A1, B1**) and the critical state primary network (**A2, B2**) corresponding to the red and blue asterisks, respectively, in the $\left(I_{AMPA}/I_{GABA},\ I_{X,E}/I_{\theta,E}\right)$ state plane shown in *Figure 2A* and inset. (**A1, B1**), (**A2, B2**): Top, spike rasters (sorted by rate) of 200 excitatory (black) and 50 inhibitory (green) neurons. Bottom, time-varying activity (1ms resolution) of excitatory (black) and inhibitory (green) populations. (**A1, A2**): External input spike rate $\nu_X = 5$ Hz. Excitatory and inhibitory neurons display average firing rates of, respectively, 5.3 Hz and 20 Hz (**A1**), and 6.3 Hz and 22 Hz (**A2**). (**B1, B2**): In these simulations $\nu_X$ was increased by 5%. Excitatory and inhibitory neurons display average firing rates of, respectively, 7.5 Hz and 25 Hz (**B1**), and 12 Hz and 34 Hz (**B2**).

The online version of this article includes the following figure supplement(s) for figure 3:

**Figure supplement 1.** Power spectra of population spiking activity observed in network simulations.

Thus, direct simulations confirm that analytically derived network parameters for both steady and critical primary networks provide the anticipated regimes of network dynamics.

To facilitate the comparison of characteristic features exhibited by a simulated network with experimentally measurable quantities, we compute temporal correlation of spiking activity that quantifies average pairwise correlation between spike trains of excitatory neurons. In the context of the DPX task performed in drug-naive and drug conditions studied in *Zick et al., 2018* and with the purpose of elucidating the mechanism of drug-induced desynchronization of spiking activity, we investigated how temporal correlations depend on the strength of external drive and the NMDAR mediated synaptic current. To this end, we varied external input rate $v_X$ and the NMDAR conductance parameters $g_{NMDA,E}$ and $g_{NMDA,I}$ relative to their respective standard values $v_X^*$, and $g_{NMDA,E}^*$ and $g_{NMDA,I}^*$, while keeping all other system parameters fixed, and performed simulations of the ensuing networks. Conductances for excitatory and inhibitory neurons were scaled with the same factor and, therefore, their relative values $g_{NMDA,E}/g_{NMDA,E}^*$ and $g_{NMDA,I}/g_{NMDA,I}^*$ are the same; in the following we drop the E, I designation.

*Figure 4* displays correlation of spiking activity (panels A1, A2, C1, C2) and synchrony (0-lag correlation, panels B1, B2, D1, D2) obtained from spike trains of simulated steady (panels A1, B1, C1, D1) and critical (panels A2, B2, C2, D2) networks for a range of $v_X/v_X^*$ (panels A1, A2, B1, B2) and $g_{NMDA}/g_{NMDA}^*$ (panels C1, C2, D1, D2) values. It is seen that in the steady state primary network correlations are weak and insensitive to the modulations of external input rate or NMDAR conductance (*Figure 4* panels A1, B1, C1, D1). In contrast, in the critical state primary network temporal correlations show sharp dependence on these parameters (*Figure 4* panels A2, C2), and with decreasing external drive or decreasing NMDAR conductance profoundly attenuating spike synchrony (*Figure 4* panels B2, D2).

## Circuit mechanisms of spike synchronization modulation

What are the network mechanisms of external drive and NMDA conductance dependent spike synchronization? Why in the network close to the boundary between asynchronous and synchronous regimes, are spike correlations strongly affected by the modulations of external inputs and recurrent NMDA currents, but in the network far from this boundary and deep in the region of the asynchronous regime, correlations are essentially independent of these modulations? How does the interplay between synchronous and asynchronous regimes at their boundary lead to spike synchronization when external input rate $v_X$ increases, and to desynchronization when the NMDA conductance $g_{NMDA}$ decreases?

To answer these questions and to illuminate the role of asynchronous and synchronous regimes in the shaping of network-wide synchronization of spiking activity, we carried out linear stability analysis in the $(v_X/v_X^*, g_{NMDA}/g_{NMDA}^*)$ parameter plane while keeping the remaining parameters fixed. For both steady and critical state primary networks, stability is investigated in the vicinity of the standard values of the external input spike rate and NMDAR conductances corresponding to the respective networks.

*Figure 5* illustrates state diagrams in the $(v_X/v_X^*, g_{NMDA}/g_{NMDA}^*)$ plane in the neighborhood of the steady (*Figure 5A*) and critical (*Figure 5B*) state primary networks. As in *Figure 2A*, the critical line ($\lambda = 0$) separating the asynchronous stationary ($\lambda < 0$) and synchronous oscillatory ($\lambda > 0$) states is shown in white color. Asterisks correspond to the loci of the steady (*Figure 5A*) and critical (*Figure 5B*) state primary networks in these parameter planes. It is seen that the modulations of $v_X$ and $g_{NMDA}$ in the steady state primary network (*Figure 5A*) do not change the network state; these modulations have no impact on the spike correlation and the strength of synchrony (*Figure 4B1 and D1* and *Figure 5A* insets).

In contrast, the modulations of $v_X$ and $g_{NMDA}$ in the critical state primary network (*Figure 5B*) induce transitions between the network states. Specifically, as the external input spike rate $v_X$ increases (horizontal yellow arrow in *Figure 5B*) the system crosses the boundary between asynchronous and synchronous regimes and the network state changes from stationary to oscillatory; this transition is accompanied by a sharp increase in spike synchrony (*Figure 4B2* and *Figure 5B* bottom inset). The decrease of NMDAR conductance $g_{NMDA}$ (vertical magenta arrow in *Figure 5B*) causes the system to cross the boundary again, and the network state changes now from oscillatory to stationary; this transition is accompanied by a sharp decrease in spike synchrony (*Figure 4D2* and *Figure 5B* right inset).

Thus, this analysis reveals that networks that are close to the boundary between asynchronous and synchronous regimes, in contrast to asynchronous networks that are far from this boundary, have a rich

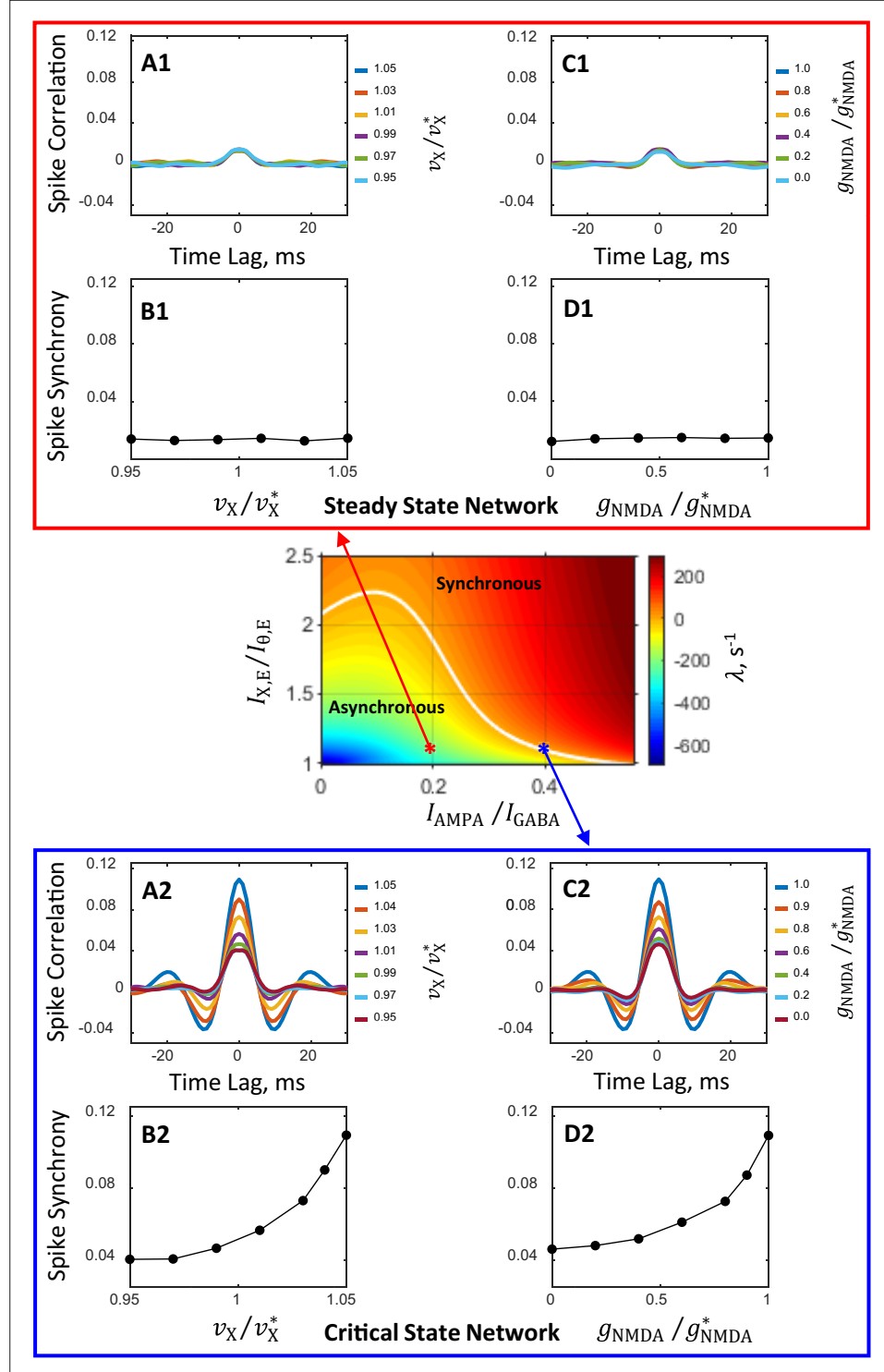

**Figure 4.** Spiking activity correlation and synchrony computed from spike trains of simulated networks. Conductance parameters are solutions of mean field equations for the steady state primary network (**A1, B1, C1, D1**) and the critical state primary network (**A2, B2, C2, D2**) corresponding to the red and blue asterisks, respectively, in the $\left(I_{\text{AMPA}}/I_{\text{GABA}},\ I_{\text{X,E}}/I_{\theta,\text{E}}\right)$ state plane shown in **Figure 2A** and inset. For the steady state network, correlation and synchrony are weak and insensitive to the modulation of external input spike rate $\nu_{\text{X}}$ (**A1, B1**) and NMDAR conductance $g_{\text{NMDA}}$ (**C1, D1**). In contrast, for the critical state network spike correlation depends strongly on the external spike rate (**A2**) and NMDAR conductance (**C2**) and the degree of spike synchrony could be modulated from relatively weak to strong (**B2, D2**). Results shown in (**C1, D1, C2, D2**) are obtained from

*Figure 4 continued on next page*

*Figure 4 continued*

simulations in which $\nu_X$ is increased by 5%. The magnitudes of modulation of $\nu_X$ and $g_{NMDA}$ are normalized by their standard values $v_X^*$ and $g_{NMDA}^*$, respectively. The numbers next to color-coded lines for spike correlation plots show the normalized magnitudes of external input spike rates, $v_X/v_X^*$, (**A1, A2**) and NMDAR conductance, $g_{NMDA}/g_{NMDA}^*$, (**C1, C2**).

dynamic behavior. The dynamic states of these networks could be easily switched around by modulations in the external drive and the strength of recurrent excitation by NMDAR mediated currents. Switching between the network states, in turn, results in sharp changes in the degree of network-wide synchronization of spiking activity in response to these modulations.

## Explaining the effects of blocking of NMDAR observed in primate PFC by the prefrontal circuit model

As illustrated in *Figure 1B*, spiking activity observed in monkey PFC in the DPX task (*Zick et al., 2018*) remains practically desynchronized after probe presentation for about 200ms but it begins to increase sharply about 200ms before the motor response. To get a deeper insight into the properties of spike timing dynamics, we show in *Figure 6* temporal correlations of spiking activity during the 200ms period following probe presentation (*Figure 6A1*) and during the 200ms period preceding the motor response (*Figure 6B1*) in drug-naive (black) and drug (magenta) conditions. It can be now appreciated that in drug-naive condition, population activity during the pre-response period develops characteristics of synchronized oscillation behavior, as signaled by the appearance of time lagged peaks of correlation (blue arrows, *Figure 6B1*, black). However, administration of a drug blocking NMDAR desynchronizes neuronal activity during this period (*Figure 6B1*, magenta).

The presence of strong spike synchrony (0ms lag) together with the correlation peaks at ±18 ms lags in the pre-response period (*Figure 6B1*), and the absence of these characteristics in the initial probe period (*Figure 6A1*) suggest that after probe presentation but before motor response network dynamics switches from the asynchronous stationary state to the synchronous oscillation state with a $\gamma$-frequency around 55 Hz. Desynchronization of neuronal activity produced by drug administration implies that NMDAR blockage prevents PFC circuits operating in the asynchronous regime from switching to synchronous dynamics.

These experimental findings could be readily explained by a prefrontal network model that operates on the boundary between asynchronous and synchronous regimes. We start by recalling that in the framework of our approach the pre-response afferent signals, which we assume are received by PFC neurons before the monkey's response, are modeled as an increase in the external spike rate from its background level $\nu_X$. This assumption is supported by the increase in the population spike rate preceding the monkey's response observed in neural data shown in *Figure 1D*. Secondly, the effect of drug administration is modeled by setting NMDAR conductances $g_{NMDA,E}$ and $g_{NMDA,I}$ to zero. The capacity of the prefrontal network model to provide a circuit mechanism for the emergence of synchrony in spiking activity and drug-dependent desynchronization can be illustrated by considering the system's behavior in the $(v_X/v_X^*, g_{NMDA}/g_{NMDA}^*)$ state plane around the point $(v_X/v_X^* = 1, g_{NMDA}/g_{NMDA}^* = 1)$ corresponding to the critical state primary network (*Figure 6C*). In this space, the effects of probe presentation on the spiking dynamics of the prefrontal circuit model under drug-naive ($g_{NMDA}/g_{NMDA}^* = 1.25$) and drug ($g_{NMDA}/g_{NMDA}^* = 0$) conditions are represented, respectively, by black and magenta horizontal arrows (*Figure 6C*). The arrows are pointing from the state of the network corresponding to the initial probe period ($v_X/v_X^* = 0.97$) to the network state corresponding to the pre-response period ($v_X/v_X^* = 1.03$).

In drug-naive condition, increase in the external spike rate $v_X$ switches the circuit model from asynchronous to synchronous regime (*Figure 6C*, black arrow crosses the boundary between the regimes). The oscillation frequency is about 50 Hz, which is manifested in the temporal correlations of spiking activity as a sharp increase in synchrony and appearance of peaks at ±20 ms lags (*Figure 6A2 vs B2*, black line). This is very similar to what is observed in monkey PFC during the initial probe and pre-response periods in the DPX task (*Fig. 6A1 vs A2 and B1 vs B2*, black line). In the drug condition, setting NMDAR conductance to zero prevents the circuit model from switching to the synchronous regime in response to an increase in the external spike rate $v_X$ (*Figure 6C*, magenta arrow does not cross the boundary between the regimes). This, in turn, considerably reduces the degree of spike

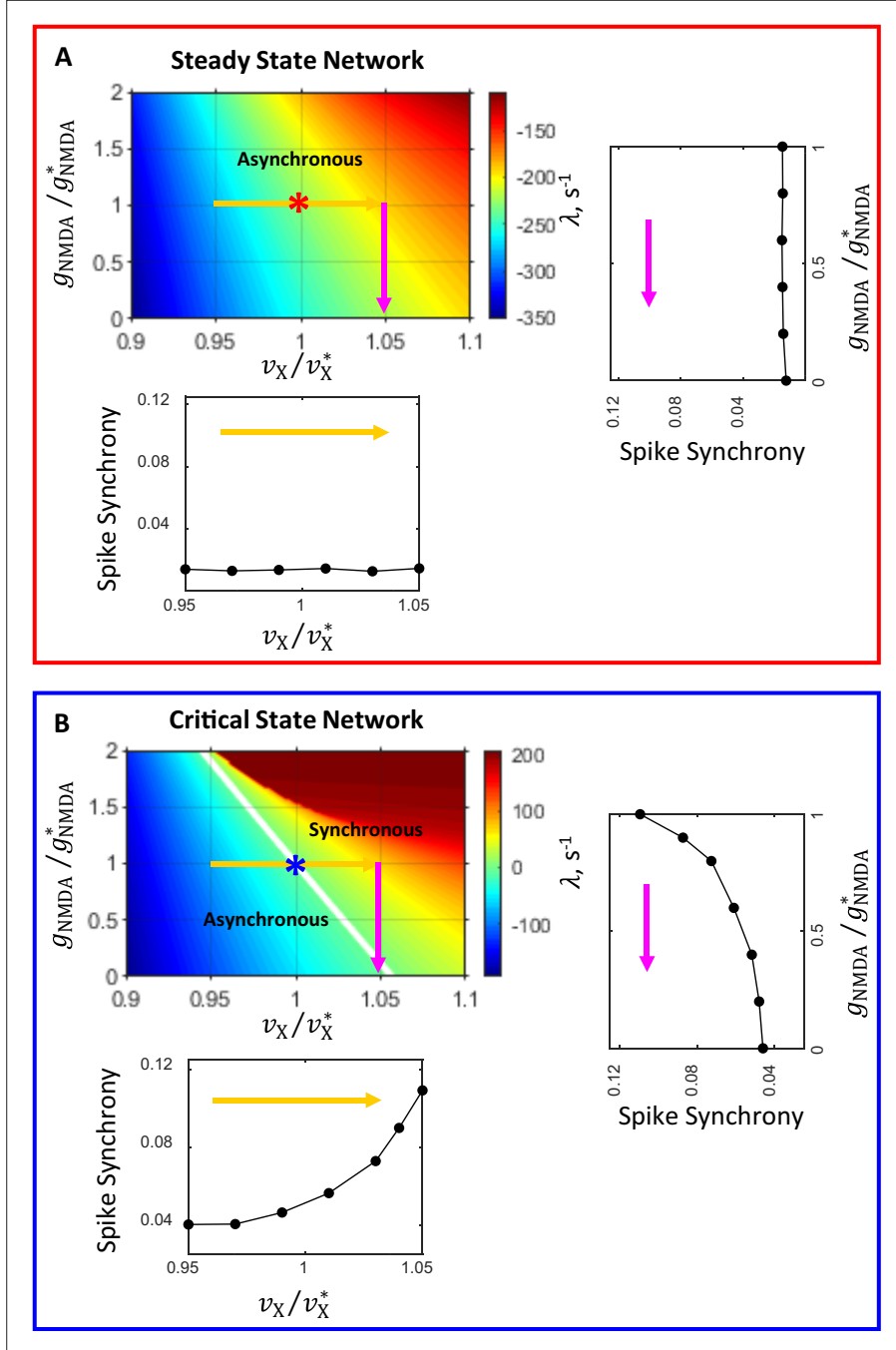

**Figure 5.** Network state diagrams in the $\left(v_{\mathrm{X}}/v_{\mathrm{X}}^*,\ g_{\mathrm{NMDA}}/g_{\mathrm{NMDA}}^*\right)$ plane. The critical line ($\lambda = 0$, white line) separates the parameter plane into regions of asynchronous stationary ($\lambda < 0$) and synchronous oscillatory ($\lambda > 0$) regimes. In the state diagram for the steady state network (**A**) the critical line is beyond the area covered by the diagram. Asterisks correspond to the steady (**A**) and critical (**B**) state primary networks in these planes. Color-coded arrows show the range of modulation of $\nu_{\mathrm{X}}$ (yellow) and $g_{\mathrm{NMDA}}$ (magenta) corresponding to the range of modulation of these parameters for which temporal correlations of spiking activity and synchrony are shown in *Figure 4*. The insets show how spike synchrony changes along the corresponding arrows in the state diagrams. These insets display the same plots for spike synchrony that are shown in panels **B1** and **B2** (bottom insets in **A** and **B**, correspondingly) and **D1** and **D2** (right insets in **A** and **B**, correspondingly) in *Figure 4*.

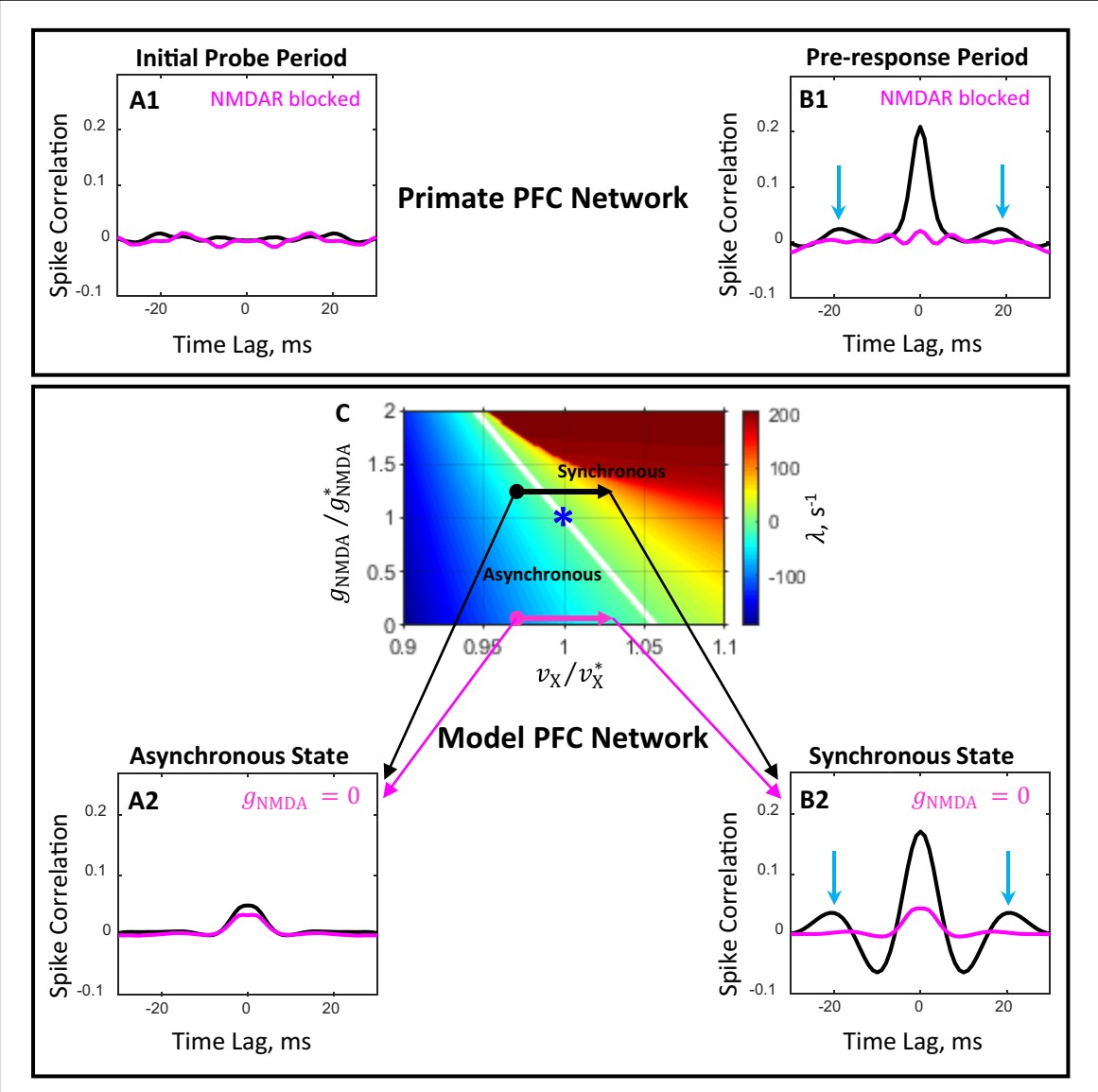

**Figure 6.** Comparison of the effects of blocking of NMDAR in primate PFC and in the prefrontal circuit model. (**A1, B1**) Plots show population average temporal correlations between spiking activity of neuron pairs recorded from PFC during the 200 ms period immediately following probe presentation (**A1**) and the 200 ms period immediately preceding the motor response (**B1**) in the DPX task (*Zick et al., 2018*). In the drug-naive condition (black line), population activity during the pre-response period develops characteristics of synchronous oscillation with a frequency of ~55 Hz (peaks at time lags ±18 ms, blue arrows, **B1**). Administration of a drug blocking NMDAR (magenta line) desynchronizes neuronal activity during the pre-response period (**B1**). (**A2, B2, C**) Temporal correlations (**A2, B2**) computed from spike trains of simulated networks corresponding to four conditions shown in the $\left(\nu_X/\nu_X^*, \ g_{NMDA}/g_{NMDA}^*\right)$ state plane (**C**) by bold dots and arrow heads: *initial probe* ( $v_X/v_X^* = 0.97$, **A2**) and *pre-response* ( $v_X/v_X^* = 1.03$, **B2**) periods for *drug-naive* ( $g_{NMDA}/g_{NMDA}^* = 1.25$, black line) and *drug* ( $g_{NMDA}/g_{NMDA}^* = 0$, magenta line) conditions. The critical line ($\lambda = 0$, white line in panel **C**) separates the parameter plane into regions of asynchronous stationary ($\lambda < 0$) and synchronous oscillation ($\lambda > 0$) regimes. The locus of the blue asterisk corresponds to the critical state primary network in this plane.

The online version of this article includes the following figure supplement(s) for figure 6:

**Figure supplement 1.** Dependence of spike synchrony on the network size.

**Figure supplement 2.** Dependence of spike synchrony on the fraction of neurons receiving increased external input.

synchrony compared to drug-naive condition (*Figure 6B2*, magenta vs black line), similar to the desynchronizing effect of NMDAR antagonist administration on spiking activity in monkey PFC (*Figure 6 B1*, magenta vs black line).

In the consideration above, we investigated the network spiking dynamics in the asynchronous and synchronous states during stationary external input at a decreased ($v_X/v_X^* = 0.97$) and increased ($v_X/v_X^* = 1.03$) external rate. To simulate a more biologically realistic scenario, we also examined the network behavior in response to transient external input. In this analysis, external input rate had a trapezoid-like temporal profile (*Figure 7A*). First, external rate was fixed at a lower level ($v_X/v_X^* = 0.97$) setting the network in the asynchronous state. Then, throughout 100ms period the rate was linearly increased to a higher level ($v_X/v_X^* = 1.05$) and kept constant for 400ms, pushing the network across the boundary to the synchronous state. Finally, the rate was decreased to the initial level during the next 100ms to switch the network back to the asynchronous state. *Figure 7* shows time evolution of population spike rate (*Figure 7B*) and synchrony (*Figure 7C*) in response to such transient external input (*Figure 7A*) for $g_{NMDA}/g_{NMDA}^* = 1.25$ (black) and $g_{NMDA}/g_{NMDA}^* = 0$ (magenta) corresponding to drug-naive and drug conditions, respectively. These simulated temporal profiles can be compared with the temporal profiles shown in *Figure 1* for population average spike rate (*Figure 1D*) and synchrony (*Figure 1B*) obtained from experimental data. Parallels between the simulated and recorded neural data are evident. Spike synchrony and spike rate peak at about the same time both in simulated (*Figure 7B and C*) and recorded (*Figure 1B and D*) neural activity. Further, the increase in spike rate is early and gradual in comparison to the increase in spike synchrony which is delayed and abrupt both in simulated (*Figure 7B and C*) and recorded (*Figure 1B and D*) neural activity. While our relatively simple model qualitatively is consistent with dynamical features of the firing rate and synchrony observed in primate PFC, there are, however, some quantitative discrepancies in firing rates. In addition, recorded neural activity exhibits complex dynamics following the response (*Figure 1B and D*), that are not evident in the simulation (*Figure 7B and D*). This presumably reflects temporal modulation of synaptic inputs to the recorded neurons in the biological data that are more complex than the ramp transient we implemented in the simulation.

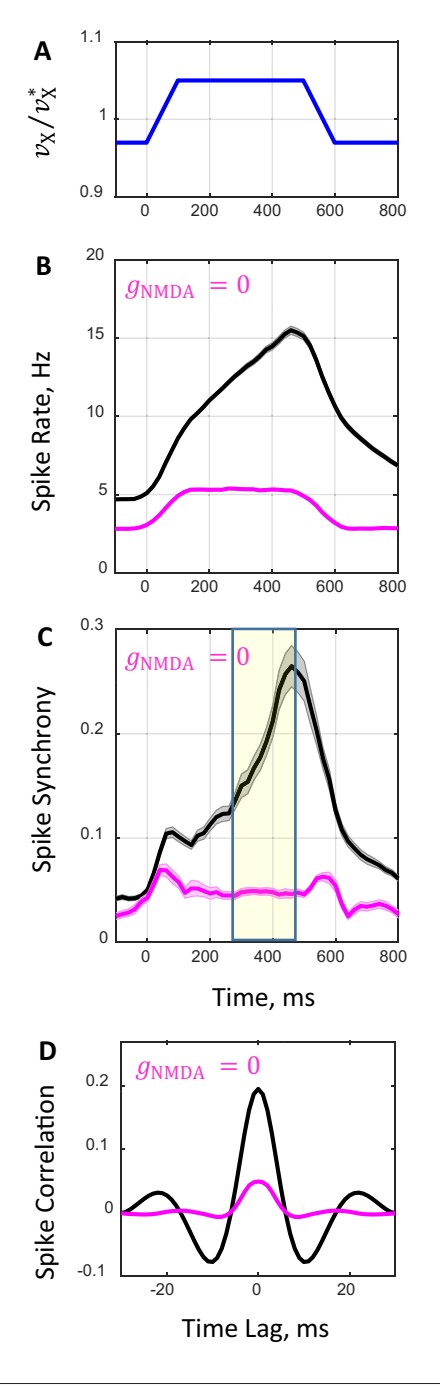

**Figure 7.** Network spiking dynamics in response to transient input. (**A**) Temporal profile of external rate. Initially, external rate is fixed at a lower level ($v_X/v_X^* = 0.97$) and the network is in the asynchronous state. At time $t = 0$ the rate begins to increase and in the next 100 ms it crosses the boundary between the asynchronous and synchronous states reaching a higher level ($v_X/v_X^* = 1.05$). The rate is kept constant for the next 400 ms and, afterwards, it decreases within 100 ms and returns to its initial level corresponding to the asynchronous state. (**B**, **C**) Average population

*Figure 7 continued on next page*

*Figure 7 continued*

spike rate (**B**) and synchrony (**C**) obtained from spike trains of 100 network simulations that received the transient external input shown in **A** for drug-naive ($g_{\text{NMDA}}/g^*_{\text{NMDA}} = 1.25$, black line) and drug conditions ($g_{\text{NMDA}}/g^*_{\text{NMDA}} = 0$, magenta line). Shaded grey and magenta bands show the standard errors for spike rate (**B**) and synchrony (**C**). (**D**) Population average temporal correlations between spiking activity of neuron pairs obtained in simulations during the 200 ms period shown in (**C**) by yellow shaded area.

In summary, the analyses of simulations with stationary and transient external inputs suggest that when the prefrontal network model operates close to the boundary between asynchronous stationary and synchronous oscillatory regimes it has a considerable capacity to capture experimentally observed aspects of spike synchrony in both drug-naive and drug conditions.

## Role of the balance between NMDAR mediated recurrent excitation and GABA inhibition

So far, in most of our analyses, we did not vary the balance between the tonic component of recurrent excitation mediated by NMDA and GABA inhibition, keeping it fixed at $I^*_{\text{NMDA}}/I^*_{\text{GABA}} = 0.15$. We have only shown that the network frequency at the onset of oscillation essentially is independent of the $I_{\text{NMDA}}/I_{\text{GABA}}$ balance (*Figure 2—figure supplement 1B*), and that the characteristic features of the $(I_{\text{AMPA}}/I_{\text{GABA}}, I_{\text{X,E}}/I_{\theta,\text{E}})$ state diagram qualitatively remain unchanged when this balance is varied (*Figure 2—figure supplement 1A*). Could, however, the $I^*_{\text{NMDA}}/I^*_{\text{GABA}}$ balance be crucial for the prefrontal circuit model capacity to provide the underlying mechanism for external input and NMDA conductance dependent spike synchronization? To investigate this issue, we analyzed how characteristic features of the $(v_{\text{X}}/v^*_{\text{X}}, g_{\text{NMDA}}/g^*_{\text{NMDA}})$ state diagram shown in *Figure 6C* depend on the $I^*_{\text{NMDA}}/I^*_{\text{GABA}}$ balance.

*Figure 8* shows state diagrams in the $(v_{\text{X}}/v^*_{\text{X}}, g_{\text{NMDA}}/g^*_{\text{NMDA}})$ plane obtained for several $I^*_{\text{NMDA}}/I^*_{\text{GABA}}$ balance values. It is seen that the orientation of the critical line in the state space depends on the $I^*_{\text{NMDA}}/I^*_{\text{GABA}}$ balance. When the balance is shifted toward stronger inhibition ($I^*_{\text{NMDA}}/I^*_{\text{GABA}} < 0.15$, *Figure 8A*), the critical line becomes too steep: in the drug condition, blocking NMDA current may not necessarily lead to spike desynchronization because the external spike modulation could trigger the network to switch to the synchronous regime (magenta arrow in *Figure 8A*). On the other hand, when the balance is shifted toward stronger tonic excitation ($I^*_{\text{NMDA}}/I^*_{\text{GABA}} > 0.15$, *Figure 8C*), the critical line becomes too flat: in the drug-naive condition the external spike modulation may not be able to produce strong enough synchrony because the system would be too close to the critical line, and not shift deep enough into the region of the synchronous regime (black arrow in *Figure 8C*).

## Dependence of oscillatory instability growth rate on synaptic parameters

Further insights into how synaptic conductances and external rate affect synchrony can be achieved by obtaining an analytic expression describing the dependence of the rate of oscillatory instability

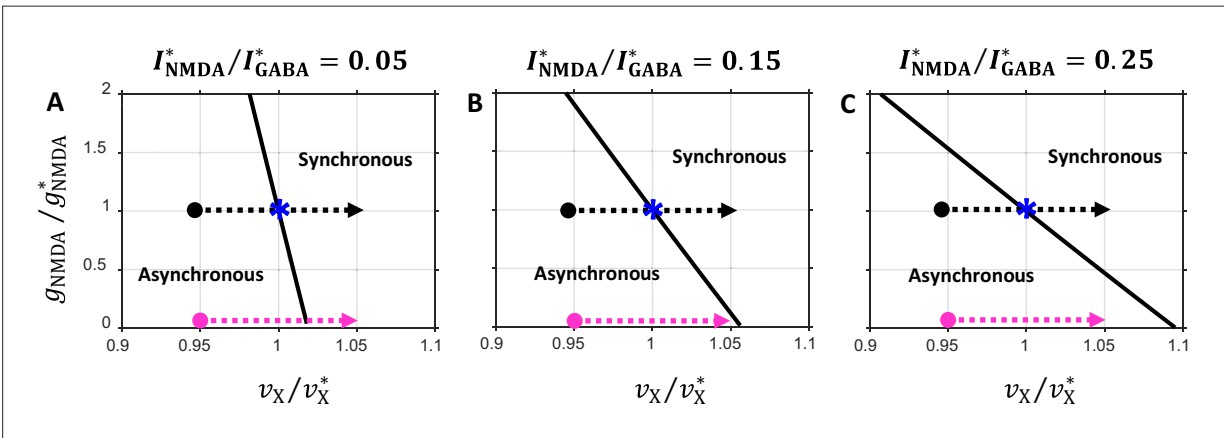

**Figure 8.** State diagrams in the $(v_{\text{X}}/v^*_{\text{X}}, g_{\text{NMDA}}/g^*_{\text{NMDA}})$ plane obtained for several values of the balance between the NMDA and GABA currents. Notations are the same as in *Figure 6C*. (**A**) $I^*_{\text{NMDA}}/I^*_{\text{GABA}} = 0.05$; (**B**) $I^*_{\text{NMDA}}/I^*_{\text{GABA}} = 0.15$; (**C**) $I^*_{\text{NMDA}}/I^*_{\text{GABA}} = 0.25$. Note that the critical line orientation depends on the $I^*_{\text{NMDA}}/I^*_{\text{GABA}}$ balance.

growth $\lambda$ on these parameters near the boundary between the asynchronous and synchronous states. Such expression can be derived by linearizing the stability analysis equations in the limit of small relative changes $\Delta g_{\text{AMPA}}/g^*_{\text{AMPA}}$, $\Delta g_{\text{NMDA}}/g^*_{\text{NMDA}}$, $\Delta g_{\text{GABA}}/g^*_{\text{GABA}}$, and $\Delta v_X/v^*_X$ of the synaptic parameters around the critical point $\left\{ g^*_{\text{AMPA},\{\text{E,I}\}}, g^*_{\text{NMDA},\{\text{E,I}\}}, g^*_{\text{GABA},\{\text{E,I}\}}, v^*_X \right\}$ corresponding to the onset of oscillatory instability where $\lambda = 0$ (conductances $g_{R,\text{E}}$ and $g_{R,\text{I}}$ of excitatory and inhibitory neurons ($R$ = AMPA, NMDA, GABA) are again scaled with the same factors and, thus, their relative changes are equal: $\Delta g_{R,\text{E}}/g^*_{R,\text{E}} = \Delta g_{R,\text{I}}/g^*_{R,\text{I}}$). The calculation is detailed in the Materials and methods section. The result is that $\lambda$ in the vicinity of the critical point on the boundary between the steady and oscillatory states can be approximated by

$$\lambda = \Lambda_{\text{AMPA}} \left( \frac{\Delta g_{\text{AMPA}}}{g^*_{\text{AMPA}}} + \frac{\Delta \phi'_{I_{\text{syn,E}}}}{\phi'_{I_{\text{syn,E}}}} \right) + \Lambda_{\text{NMDA}} \left( \frac{\Delta g_{\text{NMDA}}}{g^*_{\text{NMDA}}} + \frac{\Delta \phi'_{I_{\text{syn,E}}}}{\phi'_{I_{\text{syn,E}}}} \right) - \Lambda_{\text{GABA}} \left( \frac{\Delta g_{\text{GABA}}}{g^*_{\text{GABA}}} + \frac{\Delta \phi'_{I_{\text{syn,I}}}}{\phi'_{I_{\text{syn,I}}}} \right), \quad (1)$$

where $\Lambda_{\text{AMPA}}$, $\Lambda_{\text{NMDA}}$, and $\Lambda_{\text{GABA}}$ are quantities defined by the parameters of the critical state network around which the equations are linearized, $\phi'_{I_{\text{syn,}\alpha}}$ is the slope of the neuron's current-frequency response function at the critical state, and $\Delta \phi'_{I_{\text{syn,}\alpha}}$ is the change in the slope of the response function due to the deviations of the synaptic parameters from their critical values ($\alpha = \text{E, I}$ for excitatory and inhibitory neurons, respectively). The deviations of the synaptic conductances $\Delta g_{\text{AMPA}}$, $\Delta g_{\text{NMDA}}$, $\Delta g_{\text{GABA}}$, and external rate $\Delta v_X$ give rise to the changes in the corresponding average recurrent $I_{\text{AMPA}}$, $I_{\text{NMDA}}$, $I_{\text{GABA}}$ and external $I_X$ synaptic currents. This produces the change $\Delta I_{\text{syn}}$ in the average total current $I_{\text{syn}} = I_X + I_{\text{AMPA}} + I_{\text{NMDA}} - I_{\text{GABA}}$ and shifts the operating point of the current-frequency response function $v = \phi(I_{\text{syn}})$ that describes the relationship between the average total input current $I_{\text{syn}}$ and the output firing frequency of the neuron $v$. For the leaky integrate-and-fire neuron model, $\phi$ is a monotonically increasing non-linear function (see, e.g., **Renart et al., 2003**). Thus, the shift of the operating point of the neuron's response function $\phi$ due to the change $\Delta I_{\text{syn}}$ in the total average synaptic current results not only in the change of the firing rate (i.e. $\Delta \phi$), but also in the change of the slope of the response function $\Delta \phi'_{I_{\text{syn}}}$. The latter can be calculated by linearizing the self-consistent mean field equations (see Materials and methods). As a result, $\Delta \phi'_{I_{\text{syn}}}$ is approximated as

$$\frac{\Delta \phi'_{I_{\text{syn,}\alpha}}}{\phi'_{I_{\text{syn,}\alpha}}} = U_\alpha \left( \frac{I_X}{I_{\text{GABA}}} \frac{\Delta v_X}{v^*_X} + \frac{I_{\text{AMPA}}}{I_{\text{GABA}}} \frac{\Delta g_{\text{AMPA}}}{g^*_{\text{AMPA}}} + \frac{I_{\text{NMDA}}}{I_{\text{GABA}}} \frac{\Delta g_{\text{NMDA}}}{g^*_{\text{NMDA}}} - \frac{\Delta g_{\text{GABA}}}{g^*_{\text{GABA}}} \right), \qquad \alpha = \text{E, I}, \quad (2)$$

where $U_\alpha$ is a positive constant defined by the parameters of the critical state network around which the mean field equations are linearized. The analytical expression for $\lambda$ given by **Equation 1, 2** provides a very good approximation of the exact relationship (see Appendix 1).

Within the linear approximation, the change $\Delta \phi'_{I_{\text{syn}}}$ is proportional to the change $\Delta I_{\text{syn}}$:

$$\frac{\Delta \phi'_{I_{\text{syn,}\alpha}}}{\phi'_{I_{\text{syn,}\alpha}}} = U_\alpha \frac{\Delta I_{\text{syn}}}{I_0}, \tag{3}$$

where $I_0$ is a positive constant. Hence, from **Equation 2** it follows that $\Delta I_{\text{syn}}$ is proportional to the expression in the brackets:

$$\Delta I_{\text{syn}} = I_0 \left( \frac{I_X}{I_{\text{GABA}}} \frac{\Delta v_X}{v^*_X} + \frac{I_{\text{AMPA}}}{I_{\text{GABA}}} \frac{\Delta g_{\text{AMPA}}}{g^*_{\text{AMPA}}} + \frac{I_{\text{NMDA}}}{I_{\text{GABA}}} \frac{\Delta g_{\text{NMDA}}}{g^*_{\text{NMDA}}} - \frac{\Delta g_{\text{GABA}}}{g^*_{\text{GABA}}} \right). \tag{4}$$

From **Equation 1** it follows that the rate of oscillatory instability growth $\lambda$ directly depends on the changes in the synaptic conductances but does not explicitly depend on the external rate variation $\Delta v_X$. However, $\lambda$ depends on $\Delta v_X$ indirectly via the terms involving the change in the slope $\Delta \phi'_{I_{\text{syn}}}$ due to the change in the average total synaptic current $\Delta I_{\text{syn}}$ (**Equations 3, 4**). In fact, $\Delta I_{\text{syn}}$ is affected by the variations of the synaptic conductances as well. Thus, the rate of instability growth $\lambda$ not only directly depends on the synaptic conductances, but also indirectly via the effect of the recurrent excitatory and inhibitory currents mediated by them on the average total synaptic current and, therefore, the operating point of the current-frequency response function.

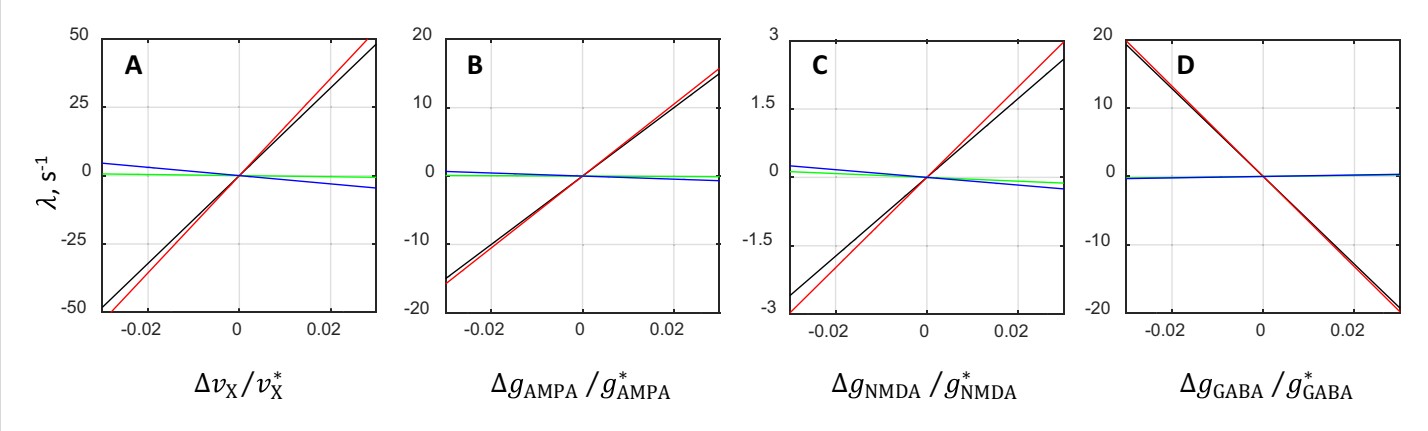

**Figure 9.** Contributions from various terms in the analytical approximation of the oscillatory instability growth rate $\lambda$. The plots show separately the rate $\lambda$ and its individual terms $\Lambda_{\text{AMPA}}$, $\Lambda_{\text{NMDA}}$, and $\Lambda_{\text{GABA}}$ (*Equation 1*) as functions of the relative deviations from the critical value of external rate (**A**), AMPAR conductance (**B**), NMDAR conductance (**C**), and GABAR conductance (**D**). The comparison is performed by varying the underlying parameter while keeping the other parameters at their critical values. Black lines correspond to the rate $\lambda$, whereas red, green, and blue lines correspond to the terms involving $\Lambda_{\text{AMPA}}$, $\Lambda_{\text{NMDA}}$, and $\Lambda_{\text{GABA}}$, respectively. In each plot, the values corresponding to red, blue, and green lines add up to the values of black lines. Note that red lines run very close to black lines, and blue and green lines are nearly horizontal. This indicates that the term $\Lambda_{\text{AMPA}}$ alone approximates the dependence of $\lambda$ on the synaptic parameters rather accurately and that the contributions from the remaining terms $\Lambda_{\text{NMDA}}$ and $\Lambda_{\text{GABA}}$ are rather small.

The factors $\Lambda_{\text{AMPA}}$, $\Lambda_{\text{NMDA}}$, and $\Lambda_{\text{GABA}}$ govern the strength of the direct and indirect contributions of the changes in the synaptic conductances $\Delta g_{\text{AMPA}}$, $\Delta g_{\text{NMDA}}$, and $\Delta g_{\text{GABA}}$ to the oscillatory instability. By inspecting *Equation 1*, one can see that the strength of the direct contribution of the change $\Delta g_R$ ($R$ = AMAPA, NMDA, GABA) is determined only by the corresponding factor $\Lambda_R$ via the term $\Lambda_R \Delta g_R / g_R^*$. However, the strength of its indirect contribution is determined by all three factors, $\Lambda_{\text{AMPA}}$, $\Lambda_{\text{NMDA}}$, and $\Lambda_{\text{GABA}}$, through the changes in the slopes $\Delta \phi'_{I_{\text{syn,E}}}$ and $\Delta \phi'_{I_{\text{syn,I}}}$, which depend on $\Delta g_R$ (*Equation 2*). For example, the strength of direct contribution to $\lambda$ due to the change in the GABAR conductance $\Delta g_{\text{GABA}}$ is determined only by $\Lambda_{\text{GABA}}$ via the term $\Lambda_{\text{GABA}} \Delta g_{\text{GABA}} / g_{\text{GABA}}^*$ in *Equation 1*. However, the strength of indirect contribution from $\Delta g_{\text{GABA}}$ is determined by all three factors $\Lambda_{\text{AMPA}}$, $\Lambda_{\text{NMDA}}$, and $\Lambda_{\text{GABA}}$ via the terms $\Lambda_{\text{AMPA}} \Delta \phi'_{I_{\text{syn,E}}} / \phi'_{I_{\text{syn,E}}}$, $\Lambda_{\text{NMDA}} \Delta \phi'_{I_{\text{syn,E}}} / \phi'_{I_{\text{syn,E}}}$, and $\Lambda_{\text{GABA}} \Delta \phi'_{I_{\text{syn,I}}} / \phi'_{I_{\text{syn,I}}}$ in *Equation 1* because $\Delta \phi'_{I_{\text{syn,E}}}$ and $\Delta \phi'_{I_{\text{syn,I}}}$ themselves depend on $\Delta g_{\text{GABA}}$ (*Equation 2*). As noted above, this indirect contribution is due to the change in the average total synaptic current and, therefore, the change in the operating point of the current-frequency response function.

*Figure 9* illustrates the contributions of individual terms involving $\Lambda_{\text{AMPA}}$, $\Lambda_{\text{NMDA}}$, and $\Lambda_{\text{GABA}}$ in *Equation 1* to the oscillatory instability growth rate $\lambda$. The panels display separately four cases in which one of the synaptic parameters is varied while the remaining three are kept constant at their critical values. It is seen that in all four cases the dominant contribution to $\lambda$ is coming from the term involving $\Lambda_{\text{AMPA}}$. The contribution related to $\Lambda_{\text{NMDA}}$ is nearly zero, whereas the contribution from $\Lambda_{\text{GABA}}$ term is much smaller than the one from $\Lambda_{\text{AMPA}}$. While both $\Lambda_{\text{NMDA}}$ and $\Lambda_{\text{GABA}} \ll \Lambda_{\text{AMPA}}$, the primary reasons are different (see Appendix 2).

It should be noted that even though $\Lambda_{\text{NMDA}}$ and $\Lambda_{\text{GABA}}$ are negligibly small, this does not mean that changes in the NMDAR and GABAR conductances do not affect oscillatory instability (black lines, panels C and D, *Figure 9*). The fact that $\Lambda_{\text{NMDA}}$ and $\Lambda_{\text{GABA}}$ are small only means that $\Delta g_{\text{NMDA}}$ and $\Delta g_{\text{GABA}}$ do not affect the oscillatory instability directly. However, the changes in the NMDAR and GABAR conductances still affect the instability growth rate $\lambda$ indirectly via the term involving the product of $\Lambda_{\text{AMPA}}$ and $\Delta \phi'_{I_{\text{syn,E}}} / \phi'_{I_{\text{syn,E}}}$ in *Equation 1*, as mentioned above (and summarized below).

Since $\Lambda_{\text{NMDA}}, \Lambda_{\text{GABA}} \ll \Lambda_{\text{AMPA}}$, we can neglect the terms involving $\Lambda_{\text{NMDA}}$ and $\Lambda_{\text{GABA}}$ in *Equation 1* for the oscillatory instability growth rate $\lambda$. With this approximation, the equation for $\lambda$ simplifies to

$$\frac{\lambda}{\Lambda_{\text{AMPA}}} = \frac{\Delta g_{\text{AMPA}}}{g_{\text{AMPA}}^*} + \frac{\Delta \phi'_{I_{\text{syn,E}}}}{\phi'_{I_{\text{syn,E}}}} = \frac{\Delta g_{\text{AMPA}}}{g_{\text{AMPA}}^*} + U_E \frac{\Delta I_{\text{syn}}}{I_0}. \tag{5}$$

Inserting the expression for $\Delta I_{\text{syn}}$ from **Equation 4**, we obtain

$$\frac{\lambda}{\Lambda_{\text{AMPA}}} = \frac{\Delta g_{\text{AMPA}}}{g^*_{\text{AMPA}}} + U_{\text{E}}\left(\frac{I_{\text{X}}}{I_{\text{GABA}}}\frac{\Delta v_{\text{X}}}{v^*_{\text{X}}} + \frac{I_{\text{AMPA}}}{I_{\text{GABA}}}\frac{\Delta g_{\text{AMPA}}}{g^*_{\text{AMPA}}} + \frac{I_{\text{NMDA}}}{I_{\text{GABA}}}\frac{\Delta g_{\text{NMDA}}}{g^*_{\text{NMDA}}} - \frac{\Delta g_{\text{GABA}}}{g^*_{\text{GABA}}}\right), \quad (6)$$

Thus, the instability growth rate $\lambda$, in essence, directly depends only on the AMPAR conductance via the first term in **Equation 6**. The term in the brackets describes the dependence on the NMDAR mediated excitation, GABAR mediated inhibition, and external rate $v_{\text{X}}$ that affect $\lambda$ only indirectly through their effect on the operating point of the response function. In addition, $\lambda$ also depends indirectly on the AMPAR conductance. For the critical state network $U_{\text{E}} = 2.5$ and $I_{\text{AMPA}}/I_{\text{GABA}} = 0.4$. Therefore, half of the contribution to $\lambda$ is due to the indirect and the second half due to the direct dependence on the AMPAR conductance. A more detailed consideration of the direct and indirect pathways by which modulations of synaptic conductances and external rate affect synchrony is given in Appendix 3.

Since in our network model we vary only the NMDAR conductance and external rate, **Equation 6** for $\lambda$ simplifies to

$$\frac{\lambda}{\Lambda_{\text{AMPA}}} = U_{\text{E}}\left(\frac{I_{\text{X}}}{I_{\text{GABA}}}\frac{\Delta v_{\text{X}}}{v^*_{\text{X}}} + \frac{I_{\text{NMDA}}}{I_{\text{GABA}}}\frac{\Delta g_{\text{NMDA}}}{g^*_{\text{NMDA}}}\right). \quad (7)$$

The expression in the brackets is proportional to the change in the average total synaptic current $I_{\text{syn}}$ (**Equation 4**). The transition to synchrony in the model simulations is achieved by increasing external input (drug-naive condition in **Zick et al., 2018**), whereas reducing the NMDAR conductance prevents the network from such transition (drug condition in **Zick et al., 2018**). These simulation results and the mechanism implemented in our model for the transition between the steady and oscillatory states, and the lack thereof when the NMDAR conductance is blocked can be explained in terms of **Equation 7** for the instability growth rate $\lambda$. As explained above, changes in external rate $\Delta v_{\text{X}}$ and NMDAR conductance $\Delta g_{\text{NMDA}}$ both affect synchrony via indirect mechanism by changing the excitatory drive $I_{\text{syn}}$ and, therefore, shifting the operating point of the neuron's response function. In the drug-naive condition, increase in external rate ($\Delta v_{\text{X}} > 0$) increases the excitatory drive. As a result, $\lambda$ becomes positive (see **Equation 7**) and the network switches to the synchronous regime. However, in the drug condition, when NMDAR is blocked ($\Delta g_{\text{NMDA}} < 0$), the initial excitatory drive is reduced compared to the drug-naive condition, and now the same increase in external rate $\Delta v_{\text{X}}$ becomes insufficient to offset the reduced excitatory drive caused by the NMDAR blockage. As a result, $\lambda$ stays negative and the network remains in asynchronous regime.

A more formal consideration of the mechanism implemented in our model for the transition between the steady and oscillatory states as well as an analytical approximation for the critical line separating these two states are given in Appendix 4. In Appendix 5, we provide theoretical explanations in terms of the equation for the oscillatory instability growth rate $\lambda$ for some other simulation results obtained earlier.

## Discussion

To better understand how synaptic mechanisms influence neural synchrony in recurrent local circuits in monkey prefrontal cortex, we developed a theoretical framework employing a sparsely connected recurrent network model accounting for AMPAR, NMDAR, and GABAR mediated synaptic currents. This allowed us to examine how varying combinations of synaptic transmission in the recurrent network influenced spike timing at the level of pairs of neurons and oscillatory dynamics at the level of neural populations. Our motivation to pursue this question derives from recent neurophysiological experiments investigating the impact of pharmacological NMDAR blockade on spike timing dynamics in monkey prefrontal cortex (**Kummerfeld et al., 2020**; **Zick et al., 2022**; **Zick et al., 2018**). These studies were initiated to investigate how risk factors associated with schizophrenia alter neural dynamics in prefrontal cortex. Those studies found that pharmacological and genetic factors associated with schizophrenia convergently reduce 0-lag synchronous spiking between pairs of prefrontal neurons in monkeys and mice (**Zick et al., 2022**). The spiking network model we develop in the present study provides a circuit mechanism capable of explaining the biological data. The principal

features of this circuit mechanism are as follows: (i) synaptic conductance parameters of the underlying circuit are such that it is in an asynchronous state near a critical boundary in the (NMDAR conductance – external input) parameter plane separating asynchronous and synchronous network states, (ii) small increases in extrinsic inputs push the circuit past this critical boundary into the region of a synchronous state, causing emergence of gamma oscillations in population activity, (iii) 0-lag synchronous spiking between neurons emerges as they stochastically entrain to the gamma population rhythm, (iv) blocking NMDAR currents prevents the circuit from switching to a synchronous regime in response to external inputs, (v) thereby precluding emergence of 0-lag synchronous spiking in neurons.

This circuit mechanism offers a reasonable explanation accounting for the task-locked increase in 0-lag spike synchrony that occurs in monkey prefrontal cortex just before the motor response in the cognitive control task (*Zick et al., 2018*): the increase in synchrony could reflect increased synaptic input to prefrontal networks at around this time, potentially from mediodorsal nucleus of thalamus (*DeNicola et al., 2020*). It also explains why pharmacological blockade of NMDAR attenuates 0-lag spike synchrony before the motor response: the deficit in NMDAR mediated synaptic currents prevents prefrontal networks from switching to a synchronous regime in response to external inputs.

In the circuit model, the balance between the AMPA component of recurrent excitation and GABA inhibition controls the network frequency at the onset of oscillation, consistent with results in *Brunel and Wang, 2003*. This frequency is virtually independent of the balance between the tonic component of recurrent excitation mediated by the NMDAR and GABA inhibition. However, the balance between the NMDA and GABA currents determines the strength of modulation of the external synaptic input needed for switching between the asynchronous stationary and synchronous oscillatory states in the absence and presence of NMDAR antagonist.

## Firing rate and synaptic mechanisms jointly influence synchronous spiking

To gain further insights into how specifically synaptic conductances and external rate affect emergence of synchronous oscillations, we obtained an analytic approximation for the oscillatory instability growth rate $\lambda$ describing the dependence on these parameters near the boundary between the asynchronous and synchronous states where $\lambda = 0$. We showed that $\lambda$, in essence, directly depends only on the AMPAR synaptic conductance; it is virtually independent of the NMDAR conductance due to the slow synaptic decay time constant, while the dependence on the GABAR conductance is much weaker compared to AMPAR because of nearly 90° effective phase lag introduced by synaptic filtering. However, $\lambda$ depends on the NMDAR, GABAR as well as AMPAR conductances and external rate indirectly via their effect on the operating point of the neuron's input current-output frequency response function. The direct dependence manifests the essential influence of the AMPAR synaptic conductance on the strength of an excitatory-inhibitory feedback loop via fast excitatory to excitatory and excitatory to inhibitory recurrent connections. The indirect dependence manifests the influence of the synaptic conductances and external rate on the location of the operating point on the current-frequency response curve and, therefore, the slope of the response function. The steepness of the slope, in turn, determines the amplitude of the neuron's response to dynamically varying input current and, therefore, affects the strength of excitatory feedback.

The analytic expression for the oscillation growth rate $\lambda$ also reveals the differences and similarities in how AMPAR and NMDAR, both of which mediate recurrent excitation, influence the stability of asynchronous state and transition to synchronous oscillations. Both AMPAR and NMDAR conductances affect $\lambda$ indirectly by influencing the amplitude of the neuron's response to varying input current. However, because AMPA currents are much faster than NMDA currents, unlike NMDAR, AMPAR conductance also affects $\lambda$ directly by influencing the strength of fast excitatory feedback.

## Relation to prior studies of NMDAR function and oscillatory dynamics

Previous work (*Wang, 1999*) suggested that NMDAR mediated recurrent currents have a stabilizing effect on the network activity. Compte and colleagues (*Compte et al., 2000*) carried out spiking network simulations with different relative contributions of the NMDAR and AMPAR mediated currents to the recurrent excitation and showed that with less NMDA but more AMPA currents, the asynchronous steady state becomes unstable and neurons begin to synchronize, leading to network oscillations in the gamma band. At first glance, these simulation results seem to contradict

the experimental findings in *Zick et al., 2018*. Indeed, in the neural recording experiments blocking NMDAR caused desynchronization of neurons, whereas in the simulations (*Compte et al., 2000*) the reduction of NMDAR currents provoked strong synchronization. Our model and theoretical analysis allows to explain this apparent paradox. In general, the asynchronous state becomes unstable and oscillation emerges when an excitatory feedback from the fast AMPA currents becomes sufficiently strong and is followed by a strong inhibitory feedback from the slower GABA currents (*Brunel and Wang, 2003*; *Compte et al., 2000*; *Tsodyks et al., 1997*; *Wang, 1999*). As explained above, the excitatory feedback can be enhanced via different mechanisms involving direct and indirect influence of synaptic parameters on the instability growth rate. In *Compte et al., 2000*, the concurrent increasing AMPAR and decreasing NMDAR conductances nullifies the indirect effect because contributions from the changes in the NMDAR and AMPAR mediated currents to the average total synaptic current, in essence, cancel each other. As a result, the operating point of the response function, defined by the average total current, does not change. However, due to the direct effect of the AMPAR on the instability growth rate $\lambda$, increasing AMPAR conductance enhances the excitatory-inhibitory feedback loop leading to the destabilization of the asynchronous activity and emergence of synchronous oscillations. In our model, by contrast, there is no direct effect on the instability growth rate because the AMPAR conductance is kept fixed, and the enhancement of recurrent excitatory feedback is entirely due to the indirect mechanism. It is achieved through external rate increase at a certain strength of the NMDAR conductance resulting in the neuron's operating point shift toward a steeper slope above the point of the critical network. This induces network oscillation and synchronization of neurons as observed in monkey PFC when NMDAR is not blocked (*Zick et al., 2018*). However, when the NMDAR conductance is set to zero, the average total synaptic current is reduced, and the operating point moves down to such locus that it cannot be shifted above the point of the critical network by the same increase in external rate. As a result, external rate increase no longer provides a strong enough excitatory feedback, the network remains in asynchronous state, and no increase in synchrony occurs, consistent with observations in *Zick et al., 2018* when NMDAR is blocked.

## Relation to prior studies of NMDAR function and working memory

In monkeys performing a memory-guided saccade task, prefrontal neurons exhibit persistent activity that is associated with the maintenance of information in working memory (*Chafee and Goldman-Rakic, 1998*; *Funahashi et al., 1989*; *Goldman-Rakic, 1995*). Prior theoretical studies have investigated circuit and synaptic mechanisms that can generate persistent activity in recurrent prefrontal networks, specifically addressing how reducing NMDAR function destabilizes attractor states (patterns of stable neural activity) in these networks during a delay period (when the memory of the stimulus must be retained) leading to working memory deficits (*Calvin and Redish, 2021*; *Compte et al., 2000*; *Funahashi et al., 1989*; *Goldman-Rakic, 1987*; *Loh et al., 2007*; *Murray et al., 2014*). In one seminal study by *Compte et al., 2000*, the authors investigated the robustness of working memory storage against external synaptic noise and distraction stimuli in attractor networks. They showed that a concomitant increase of NMDAR- and GABAR-mediated currents leads to an increase of persistent activity and to a decrease of spontaneous activity, thereby enhancing the resistance of the network to distractors (*Brunel and Wang, 2001*; *Compte et al., 2000*). In another prominent work, *Murray et al., 2014*, employing an attractor network model, investigated the neural and behavioral effects of synaptic disinhibition induced by the malfunction of NMDAR mediated synapses targeting inhibitory neurons. They demonstrated that disinhibition resulted in a broadening of stimulus selective persistent activity at the neural level, with a concomitant loss of precision, increase in variability over time, and increase in distractibility of stored information at the behavioral level. Although these modeling studies provide important mechanistic insight into prefrontal network dynamics underlying working memory, and potentially, working memory deficits in schizophrenia (*Goldman-Rakic, 1999*), they do not address the topic of the current study, which is how slow NMDAR recurrent excitation and external input received by the network jointly influence spike timing dynamics at the neuron level and oscillatory dynamics at the population level in the presence of fast AMPA excitation and GABA inhibition. Thus, no prior modeling study captures the relationship between NMDAR synaptic mechanisms, spike timing, and network oscillations that we have observed in neural recordings (*Kummerfeld et al., 2020*; *Zick et al., 2018*), and for which we provide a theoretical explanation in the current report.

## Spike timing disruptions and rewiring of prefrontal local circuits via STDP

We previously hypothesized that reduced synchrony at the level of spiking neurons (*Zick et al., 2022*; *Zick et al., 2018*) could disconnect prefrontal local circuits via spike-timing dependent synaptic plasticity (STDP; *Dan and Poo, 2004*; *Feldman, 2012*), contributing to the reduction in dendritic spine density that has been observed in postmortem analysis of prefrontal cortex in schizophrenia (*Glantz and Lewis, 2000*; *MacDonald et al., 2017*). However, the interaction between neural synchrony and synaptic connectivity in networks incorporating STDP is hard to predict, as changes in connectivity patterns and neural dynamics are mutually dependent and interact in complex ways as connectivity and synchrony influence each other over time. Perhaps for this reason, prior theoretical studies incorporating STDP into spiking networks have obtained divergent results with respect to how STDP changes the pattern of synaptic connections between neurons in networks, and whether synchronous inputs to the neurons are required for STDP to influence the pattern of synaptic connections. For example, STDP operating on random spiking in neurons can either lead to the formation of structured stable connections between neurons in the absence of synchronous inputs (forming neural 'groups') (*Izhikevich et al., 2004*), or not (*Morrison et al., 2007*), depending on the assumptions incorporated into the models. Similarly, correlated external input to recurrent networks incorporating STDP can either fail to produce structured synaptic connections between neurons (*Morrison et al., 2007*), or it can lead to the formation of such structured connections (*Litwin-Kumar and Doiron, 2014*) depending on the specifics of the simulations. Key parameters that could influence the diversity of outcomes among studies include whether (*Izhikevich et al., 2004*) or not (*Morrison et al., 2007*) axonal conduction delays and the geometry of recurrent connections are incorporated into the models (since circuit architecture and associated signal conduction delays powerfully influences when action potentials arrive at pre- and postsynaptic elements), as well as the specific form of the STDP rule employed (*Babadi and Abbott, 2013*; *Bono and Clopath, 2017*; *Izhikevich et al., 2004*; *Morrison et al., 2007*). Based on these results, it seems reasonable that distortions of spike timing dynamics in prefrontal networks may alter the pattern of neural connections via STDP in schizophrenia. However, the diversity of results obtained from theoretical studies of STDP outlined above make it difficult to conclude that the reduction in synchronous spiking we observed would lead to synaptic disconnection via STDP, imposing important constraints on our prior hypothesis (*Zick et al., 2022*; *Zick et al., 2018*), although this remains a possibility. Network simulations that accurately incorporate as many of these biological variables as possible may be useful in predicting how spike timing changes that may emerge downstream of schizophrenia risk factors would be likely to influence synaptic connectivity in the human cortex. In addition, as noted, genetic linkage studies have implicated altered NMDAR function in schizophrenia (*Fromer et al., 2014*; *Schizophrenia Working Group of the Psychiatric Genomics Consortium, 2014*). Since NMDAR play a central role in the molecular mechanisms that implement STDP in the brain, disruption of NMDAR synaptic transmission in schizophrenia may alter STDP directly, independently of the impact of disrupted NMDAR function on neural spiking dynamics in the disease state.

## Potential U-shaped relation between NMDAR function and spike synchrony

We had previously reported that blocking NMDAR in monkeys (*Zick et al., 2018*) and deleting a schizophrenia risk gene (*Dgcr8*) in mice (*Zick et al., 2022*), both reduced the frequency of synchronous, 0-lag spiking between prefrontal neurons. *Dgcr8* encodes a protein involved in the synthesis of miRNA, which in turn bind to mRNA and suppress their translation into proteins, including mRNA coding for NMDAR subunits (*Corbel et al., 2015*). Deleting *Dgcr8* would therefore be expected to reduce miRNA synthesis and increase translation of mRNA coding for NMDAR subunits. Given these considerations, the convergent spike desynchronization we observed in monkey drug and mouse genetic models could be explained by an inverted U-shaped relationship wherein either too little NMDAR function (as produced by NMDAR blockade in monkeys) or too much NMDAR function (as predicted to result from deletion of *Dgcr8* in mice) decreases the frequency of 0-lag spiking between prefrontal neurons (*Zick et al., 2022*; *Zick et al., 2018*). An inverted U-shaped relationship has been reported between the level of D1 dopamine receptor stimulation and the strength of persistent neural activity in prefrontal neurons during working memory tasks wherein small doses of an agonist amplify

persistent activity, and larger doses degrade it (*Vijayraghavan et al., 2007*). However, additional experimental data are needed to establish that spike synchrony exhibits a similar inverted U-shaped relation to NMDAR function, insofar as our prior neural recording studies did not test a U-shaped relationship directly (*Zick et al., 2022*; *Zick et al., 2018*). These studies did not for example contrast the effect of low versus high doses of an NMDAR agonist (such as NMDA) on spike synchrony in the monkey model, nor relate reduction in spike synchrony specifically to the upregulation of NMDAR subunit expression in the mouse model (rather than the many other proteins regulated by miRNA that are dependent on *Dgcr8*).

Results we present in the current study establish a theoretical basis and circuit mechanism explaining how reduction of NMDAR synaptic function implicated in schizophrenia could lead to the desynchronization of neural activity in prefrontal recurrent circuits. We provide evidence that spiking networks situated close to a boundary in the synaptic parameter space separating asynchronous and synchronous activity states can explain a variety of biological observations. These include the emergence of 0-lag synchronous spiking between individual prefrontal neurons when external inputs to the network push it across this state boundary, and failure of synchronous spiking to emerge between prefrontal neurons when NMDAR synaptic currents are reduced, as we have observed in neural recordings in primate prefrontal cortex (*Kummerfeld et al., 2020*; *Zick et al., 2022*; *Zick et al., 2018*).

## Materials and methods
### Experimental data
For the present theoretical study, we used experimental data obtained in our previous work (*Zick et al., 2018*). Here, we provide brief descriptions of the experimental task, NMDAR antagonist regimen, and neurophysiological recording methodology employed in that work; details have been reported in *Blackman et al., 2016*; *Zick et al., 2018*.

### Experimental task
Male rhesus macaque monkeys (8–10 kg) were trained to perform the dot-pattern expectancy (DPX) task. This task is closely related to the AX-CPT (continuous performance task) except that dot patterns replace letters as stimuli. During each trial of the DPX tasks, monkeys maintained gaze fixated on a central target as a cue stimulus (1,000ms), followed by a delay period (1,000ms), and a probe stimulus (500ms) were presented. Monkeys were rewarded for moving a joystick to the left if the cue-probe sequence had been AX (69% of trials), or to the right if any other cue-probe sequence had been presented (AY, BX, BY, collectively 31% of trials). Since the correct response to the X-probe depended on the preceding cue (A or B), the task required both working memory and cognitive control. Both The DPX and AX-CPT measure specific cognitive control impairments in schizophrenia (*Barch et al., 2003*; *Jones et al., 2010*).

### Neurophysiological recording
In our previous study (*Zick et al., 2018*), we recorded neural activity from the region of the principal sulcus (centered on Brodmann's areas 46) in the dorsolateral prefrontal cortex of two macaques performing the DPX task. We found that 0-lag synchrony while present in both monkeys was much stronger in one than the other animal. For comparison to spiking dynamics in the present neural network simulation, we used neurophysiological recording data from the monkey that exhibited the strongest 0-lag spike correlation during task performance (*Zick et al., 2018*). For neurophysiological recording, we used a computer-controlled electrode drive (System Eckhorn, Thomas Recording, GmbH) advancing 16, closely spaced, independently movable glass coated platinum/tungsten microelectrodes into the prefrontal cortex. Electrodes were spaced 400 µm apart, and interelectrode distances in the array spanned 400–1,400 µm. Moving the electrodes in depth and the position of the array within recording chambers over days made it possible to isolate the spiking activity of different neural ensembles, each containing 15–30 individually isolated, simultaneously recorded neurons. The database included in the present study consisted of 47 neural ensembles containing a total of 893 prefrontal neurons. Spike correlation was evaluated within ensembles of simultaneously recorded ensembles using spike trains recorded during DPX task performance (*Zick et al., 2018*).

## NMDAR antagonist regimen

We examined the effect of systemic administration of an NMDAR antagonist (phencyclidine, 0.25–0.30 mg/kg IM) on spike timing dynamics in prefrontal local circuits. Neural activity was recorded in a Naive condition (before first exposure to drug), and a Drug condition (following systemic drug administration) (*Zick et al., 2018*).

## Spike correlation and synchrony

To estimate correlation between spiking activity of simultaneously recorded neuron pairs as a function of time, we used a similar approach described in *Zick et al., 2018*. Correlation is evaluated from spiking activity observed during a time window $\Delta T$ around a given instant of time $t$. The window size $\Delta T$, thus, defines the temporal resolution of time resolved correlation. The interval $\Delta T$ is subdivided into small time bins of width $\Delta t$. Activity of neuron $i$ in a given trial at a time bin $t'$ is represented by a binary variable $\xi_i(t')$ that can take on two values: 1 if in the time bin $t'$ one or more spikes are present, and 0 if there are no spikes. Correspondingly, time-lagged joint spike activity of neurons $i$ and $j$ is described by the product $\xi_i(t') \times \xi_j(t' + \tau)$: it is 1 if neuron $i$ fired a spike in the time bin $t'$ and neuron $j$ fired a spike in the time bin $t' + \tau$; otherwise, it is 0. The duration of the bin $\Delta t$, thus, defines the spike coincidence window. We assume that spike firing statistics of neurons do not change during the interval $\Delta T$, so that low order moments of the binary variables, such as the mean spike frequencies $\nu_i = \overline{\xi_i(t')}$ and $\nu_j = \overline{\xi_j(t')}$ and the mean joint spike frequency $\rho_{ij}(\tau) = \overline{\xi_i(t') \times \xi_j(t' + \tau)}$, can be reliably estimated by averaging over $\Delta T/\Delta t$ time bins (bars $\overline{\phantom{.}}$ above the expressions denote time averaging operation). To avoid a contribution to correlation from possible cross-trial non-stationarity (slow covariation) of neural activity, for each neuron pair correlation is estimated from single trials and then averaged over all trials. Spiking correlation between neurons $i$ and $j$ in a single trial is characterized by the observed frequency of joint spikes $\rho_{ij}(\tau)$ normalized by the expected joint spike frequency $\nu_i \times \nu_j$ if activity of the neurons were independent: $\rho_{ij}(\tau)/(\nu_i \times \nu_j)$. We then average this ratio over the trials to obtain time-lagged correlation of spiking activity as $c_{ij}(\tau) = \langle \rho_{ij}(\tau)/(\nu_i \times \nu_j) \rangle$, where angular brackets $\langle \cdot \rangle$ denote trial averaging operation. Finally, $c_{ij}(\tau)$ is averaged over the population of simultaneously recorded pairs resulting in the population average spike correlation $C(\tau)$. Spike synchrony is defined as 0-lag correlation.

To accurately estimate spike synchrony and time-lagged correlation in PFC circuits, it is necessary to keep the value of time bin $\Delta t$, controlling the spike coincidence window, sufficiently small, within 1–2ms (no more than one spike occurred in a bin). On the other hand, the firing rates of PFC neurons are relatively low, on the order of 10 Hz. Therefore, to increase the number of counts of joint spike events and improve the estimate of spike synchrony while keeping $\Delta t$ small (and, thus, spike synchrony resolution sufficiently high), one needs to increase the duration of time window $\Delta T$ and/or the number of trials $K$ However, $\Delta T$ should be kept sufficiently short so that during this interval spiking activity remains nearly stationary, whereas $K$ cannot be made arbitrarily large because it is limited by practical considerations.

These experimental restrictions, as a result, impose constraints on the firing rates of the neurons in the pair. To derive a meaningful criterion for selecting 'good' neuron pairs, we note that for a reliable estimation of the mean joint spike firing frequency, which is a second order statistic, one needs quadratically more experimental samples than for a reliable estimation of the mean spike frequency, which is a first order statistic. We also note that the expected joint spike frequency if neurons in the pair were independent is simply given as the product of their mean spike frequencies. It is this quantity that is used as a reference (normalization) for the quantification of spike correlation strength. Therefore, to reliably estimate the joint spike firing frequency from available samples of a given pair, one should be confident that at least when assuming that neurons fire independently, a sufficiently accurate estimation of the expected joint spike frequency from these samples is possible. This, in turn, means that, given the neuron firing rates $\nu_i$ and $\nu_j$, the average total number of counts of joint spikes $(\nu_i \Delta t)(\nu_j \Delta t)(\Delta T/\Delta t)K$ observed in $\Delta T/\Delta t$ bins in $K$ trials predicted under the assumption of independence and calculated from experimental samples should be 'detectable', that is, it should be at least greater than 1. This condition results in a constraint for the geometric mean, $\bar{\nu}_{ij} = \sqrt{\nu_i \nu_j}$, of the firing rates of neuron pairs: $\bar{\nu}_{ij} > 1/\sqrt{K\Delta T \Delta t}$. The typical values for the time window and spike coincidence window are $\Delta T \sim 100$ ms and $\Delta t \sim 1$ ms. Given that the number of correct trials in the DPX task were on the order of $K \sim 200$, this

means that the geometric mean firing rate of neuron pairs, for which a reliable estimation of synchrony can be achieved, should be at least 7 Hz.

## Network model

The network consists of $N$ leaky integrate and fire neurons (see, e.g., **Dayan and Abbott, 2001**), of which $N_\text{E} = 0.8N$ are excitatory and $N_\text{I} = 0.2N$ are inhibitory (**Abeles, 1991**; **Braitenberg and Schüz, 1998**). Neurons are connected randomly with a probability $p$, so that, on average, each neuron receives $C_\text{E} = pN_\text{E}$ connections from excitatory and $C_\text{I} = pN_\text{I}$ from inhibitory neurons. In the framework of mean field consideration, the network is large ($N \gg 1$) and connections are sparse ($p \ll 1$) but the average number of connections received by individual neurons, $C$, is large ($C = pN \gg 1$). In most simulations, networks consisted of $N = 5 \cdot 10^3$ neurons that were randomly connected with the probability $p = 0.2$ and, therefore each neuron, on average, received $C = 10^3$ connections. In addition, each neuron also receives $C_\text{X}$ external connections from excitatory neurons outside of the network that fire spikes independently according to a Poisson process with rate $\nu_\text{X}$.

The dynamics of the membrane potential $V(t)$ of a neuron below the spike firing potential threshold $\theta$ obeys the standard leaky integrate and fire equation:

$$C_\text{m} \frac{dV(t)}{dt} = -g_\text{m} \left(V(t) - V_\text{L}\right) - i_\text{syn}(t),$$ (8)

where $C_\text{m}$ is the cell membrane capacitance, $g_\text{m}$ is the membrane leak conductance, $V_\text{L}$ is the resting potential, and $i_\text{syn}(t)$ is the total synaptic current. When the membrane potential reaches the threshold $\theta$, the neuron fires a spike, the potential is reset to $V_\text{rst}$, and the neuron becomes insensitive to its input for the duration of a refractory period $\tau_\text{rp}$. Both excitatory and inhibitory neurons have $\theta = -50$ mV, $V_\text{L} = -70$ mV, and $V_\text{rst} = -55$ mV. For excitatory neurons $C_\text{m} = 0.5$ nF, $g_\text{m} = 25$ nS, $\tau_\text{rp} = 2$ ms, and for inhibitory neurons $C_\text{m} = 0.2$ nF, $g_\text{m} = 20$ nS, $\tau_\text{rp} = 1$ ms (see, e.g., **Koch, 2004**).

The total synaptic input for each neuron is a linear sum of four components:

$$i_\text{syn}(t) = i_\text{AMPA}(t) + i_\text{NMDA}(t) + i_\text{GABA}(t) + i_\text{X}(t),$$ (9)

where $i_\text{AMPA}$ and $i_\text{NMDA}$ correspond to recurrent excitatory currents mediated by AMPA and NMDA receptors, respectively, $i_\text{GABA}$ corresponds to inhibitory currents mediated by GABA receptors, and $i_\text{X}$ corresponds to external currents mediated by AMPA receptors. The purpose of external currents is twofold: (i) to represent the noisy inputs due to the background synaptic activity and (ii) to convey neural signals from outside of the network.

The description of component synaptic currents of a postsynaptic neuron follows **Wang, 1999**:

$$i_\text{AMPA}(t) = g_\text{AMPA} \left(V(t) - V_\text{E}\right) \sum_j s_{\text{AMPA},j}(t)$$ (10)

$$i_\text{NMDA}(t) = \frac{g_\text{NMDA} \left(V(t) - V_\text{E}\right)}{1 + \left[\text{Mg}^{2+}\right]/\gamma \exp\left(-\beta V(t)\right)} \sum_j s_{\text{NMDA},j}(t)$$ (11)

$$i_\text{GABA}(t) = g_\text{GABA} \left(V(t) - V_\text{I}\right) \sum_j s_{\text{GABA},j}(t)$$ (12)

$$i_\text{X}(t) = g_\text{X} \left(V(t) - V_\text{E}\right) \sum_j s_{\text{X},j}(t),$$ (13)

where synaptic reversal potentials $V_\text{E} = 0$ mV and $V_\text{I} = -70$ mV. NMDAR mediated currents have voltage dependence controlled by the extracellular magnesium concentration (**Jahr and Stevens, 1990**): $\beta = 0.062$ mV$^{-1}$, $\gamma = 3.57$ mM, $\left[\text{Mg}^{2+}\right] = 1$ mM. The gating variable $s_{R,j}(t)$, describes the temporal course of postsynaptic currents received from the presynaptic neuron $j$ mediated by the receptor $R$, where $R = $ X, AMPA, NMDA, GABA. For a spike train generated by a presynaptic neuron with emission times $\{t_k\}$, the temporal dynamics of the gating variable obeys the equations

$$\begin{cases} \tau_\text{r} \dfrac{dx(t)}{dt} = -x(t) + \tau_* \sum_k \delta(t - t_k - \tau_\text{l}) \\ \tau_\text{d} \dfrac{ds(t)}{dt} = -s(t) + x(t) \end{cases},$$ (14)

where $\tau_l$, $\tau_r$ and $\tau_d$ are, respectively, latency, rising, and decay time constants. Their values are $\tau_{AMPA,l} = 1$ mS, $\tau_{AMAP,r} = 0.2$ ms, $\tau_{AMPA,d} = 2$ ms for AMPAR mediated currents (**Zhou and Hablitz, 1998**), $\tau_{NMDA,l} = 1$ mS, $\tau_{NMDA,r} = 2$ ms, $\tau_{NMDA,d} = 100$ ms for NMDAR mediated currents (**Hestrin et al., 1990**), and $\tau_{GABA,l} = 1$ mS, $\tau_{GABA,r} = 0.5$ ms, $\tau_{GABA,d} = 5$ ms for GABAR-mediated currents (**Gupta et al., 2000**). The time integral of $s(t)$ in response to a presynaptic spike equals $\tau_*$ and, thus, is independent of the temporal shape of $s(t)$, which is determined by the rising and decay time constants that are specific to each receptor type. Because the charge flowing to the cell is determined by the product of the time integral of $s(t)$ and the maximal conductance, we set $\tau_*$ to be the same for all types of receptors, so that the charge entry mediated by each type of receptor is parametrized, in essence, solely by the corresponding maximal conductance parameter.

## Network simulations

In all direct network simulations, the numerical integration of the coupled differential equations describing the dynamics of membrane potentials and synaptic variables of all cells and synapses were carried out using a custom MATALAB (The MathWorks) code implementing a second order Runge-Kutta method with interpolation of spike firing times between integration time steps $\Delta t$ (**Hansel et al., 1998**). In most simulations $\Delta t = 0.1$ ms.

## Mean field approximation

To derive maximal synaptic conductance parameters $g_{X,\alpha}$, $g_{AMPA,\alpha}$, $g_{NMDA,\alpha}$, $g_{GABA,\alpha}$ ($\alpha = E, I$) providing prescribed neural firing rates $\nu_E$ and $\nu_I$, we used mean field analysis (**Amit and Brunel, 1997**; **Brunel, 2000**; **van Vreeswijk and Sompolinsky, 1996**) extended to networks of neurons with realistic, conductance based synapses (**Brunel and Wang, 2001**; **Renart et al., 2003**). For simplicity, we disregard the heterogeneity of synaptic connectivity and assume that each neuron receives $C_E$ excitatory and $C_I$ inhibitory connections. In the mean field approximation synaptic inputs are described in terms of their average and their fluctuations arising from both external and recurrent inputs. To this end, the sums of gating variables in **Equations 10–13** are replaced by their respective population averages $\tau_* S_R^0$, where $R$ designates the type of the synapse, and

$$S_X^0 = C_X \nu_X, \quad S_{AMPA}^0 = C_E \nu_E, \quad S_{NMDA}^0 = C_E \nu_E, \quad S_{GABA}^0 = C_I \nu_I. \tag{15}$$

The voltage dependence of NMDAR conductance is linearized around the mean value of the potential $\langle V \rangle$:

$$\frac{\left(V(t) - V_E\right)}{1 + \left[Mg^{2+}\right]/\gamma \exp\left(-\beta V(t)\right)} \approx \frac{V(t) - V_E}{\kappa} + \beta \frac{\left(V(t) - \langle V \rangle\right)\left(\langle V \rangle - V_E\right)\left(\kappa - 1\right)}{\kappa^2}, \tag{16}$$

where $\kappa = 1 + \left[Mg^{2+}\right]/\gamma \exp\left(-\beta \langle V \rangle\right)$. After these simplifications, average components of synaptic currents for excitatory ($\alpha = E$) and inhibitory ($\alpha = I$) populations can be written as

$$I_{X,\alpha}^0 = g_{X,\alpha}\left(\langle V_\alpha \rangle - V_E\right)\tau_* S_X^0 = J_{X,\alpha} S_X^0 \tag{17}$$

$$I_{AMPA,\alpha}^0 = g_{AMPA,\alpha}\left(\langle V_\alpha \rangle - V_E\right)\tau_* S_{AMPA}^0 = J_{AMPA,\alpha} S_{AMPA}^0 \tag{18}$$

$$I_{NMDA,\alpha}^0 = g_{NMDA,\alpha}/\kappa\left(\langle V_\alpha \rangle - V_E\right)\tau_* S_{NMDA}^0 = J_{NMDA,\alpha} S_{NMDA}^0 \tag{19}$$

$$I_{GABA,\alpha}^0 = g_{GABA,\alpha}\left(\langle V_\alpha \rangle - V_I\right)\tau_* S_{GABA}^0 = J_{GABA,\alpha} S_{GABA}^0, \tag{20}$$

where $\langle V_\alpha \rangle$ is the average membrane potential, and $J_{R,\alpha}$ is the effective strength of the $R$-receptor mediated synapse, expressed as the total charge entering the postsynaptic neuron due to a single presynaptic spike. In this framework, the system of equations describing the dynamics of membrane potentials for each of $N_E$ excitatory and $N_I$ inhibitory neurons is reduced to equations describing the dynamics of membrane potentials $V_E(t)$ and $V_I(t)$ of just two neurons representing, respectively, excitatory, E, and inhibitory, I, populations (**Brunel and Wang, 2001**; **Renart et al., 2003**):

$$\tau_\alpha \frac{dV_\alpha(t)}{dt} = -\left(V_\alpha(t) - V_L\right) + \mu_\alpha + \sigma_\alpha \sqrt{\tau_\alpha}\eta_\alpha(t), \quad \alpha = E, I, \tag{21}$$

where $V_\mathrm{L}$ is the resting potential, $\tau_\alpha$ is the effective membrane time constant, $\mu_\alpha$ is the effective mean synaptic input, $\sigma_\alpha$ is the magnitude of the fluctuations in the synaptic input, and $\eta_\alpha(t)$ is the time course of these fluctuations:

$$\tau_\alpha = \frac{C_{\mathrm{m},\alpha}}{g_{\mathrm{m},\alpha} S_\alpha} \tag{22}$$

$$S_\alpha = 1 + T_{\mathrm{X},\alpha} \nu_\mathrm{X} + T_{\mathrm{AMPA},\alpha} \nu_\mathrm{E} + \left(T_{\mathrm{NMDA1},\alpha} + T_{\mathrm{NMDA2},\alpha}\right) \nu_\mathrm{E} + T_{\mathrm{GABA},\alpha} \nu_\mathrm{I} \tag{23}$$

$$T_{\mathrm{X},\alpha} = \frac{g_{\mathrm{X},\alpha} C_\mathrm{X} \tau_*}{g_{\mathrm{m},\alpha}} \tag{24}$$

$$T_{\mathrm{AMPA},\alpha} = \frac{g_{\mathrm{AMPA},\alpha} C_\mathrm{E} \tau_*}{g_{\mathrm{m},\alpha}} \tag{25}$$

$$T_{\mathrm{NMDA1},\alpha} = \frac{g_{\mathrm{NMDA},\alpha} C_\mathrm{E} \tau_*}{g_{\mathrm{m},\alpha} \, \kappa} \tag{26}$$

$$T_{\mathrm{NMDA2},\alpha} = \beta \frac{g_{\mathrm{NMDA},\alpha} C_\mathrm{E} \tau_* \left(\langle V_\alpha \rangle - V_\mathrm{E}\right) \left(\kappa - 1\right)}{g_{\mathrm{m},\alpha} \, \kappa^2} \tag{27}$$

$$T_{\mathrm{GABA},\alpha} = \frac{g_{\mathrm{GABA},\alpha} C_\mathrm{I} \tau_*}{g_{\mathrm{m},\alpha}} \tag{28}$$

$$\mu_\alpha = \frac{\left(T_{\mathrm{X},\alpha} \nu_\mathrm{X} + T_{\mathrm{AMPA},\alpha} \nu_\mathrm{E} + T_{\mathrm{NMDA1},\alpha} \nu_\mathrm{E}\right) \left(V_\mathrm{E} - V_\mathrm{L}\right)}{S_\alpha} + \frac{T_{\mathrm{NMDA2},\alpha} \nu_\mathrm{E} \left(\langle V_\alpha \rangle - V_\mathrm{L}\right) + T_{\mathrm{GABA},\alpha} \nu_\mathrm{I} \left(V_\mathrm{I} - V_\mathrm{L}\right)}{S_\alpha}. \tag{29}$$

In the absence of spiking and fluctuations, the average membrane potential would equal $\mu_\alpha + V_\mathrm{L}$ (**Equation 21**). The average membrane potential $\langle V_\alpha \rangle$ of spiking neuron in the presence of synaptic noise can be calculated from the distribution of potentials obtained in **Brunel and Hakim, 1999** and is given by (**Renart et al., 2003**)

$$\langle V_\alpha \rangle = \mu_\alpha + V_\mathrm{L} - \left(\theta - V_\mathrm{rst}\right) \nu_\alpha \tau_\alpha - \left(\mu_\alpha + V_\mathrm{L} - V_\mathrm{rst}\right) \nu_\alpha \tau_{\mathrm{rp},\alpha}. \tag{30}$$

The total synaptic noise $\sigma_\alpha^2$ characterizing fluctuations in the input that result from random arrival of spikes is approximated as the sum of the fluctuations in the external and recurrent inputs (**Fourcaud and Brunel, 2002**):

$$\sigma_\alpha^2 = \sigma_{\mathrm{X},\alpha}^2 + \sigma_{\mathrm{AMPA},\alpha}^2 + \sigma_{\mathrm{NMDA},\alpha}^2 + \sigma_{\mathrm{GABA},\alpha}^2, \tag{31}$$

where

$$\sigma_{R,\alpha}^2 = \frac{J_{R,\alpha}^2 S_R^0 \tau_\alpha}{C_{\mathrm{m},\alpha}^2}, \quad R = \mathrm{X, AMPA, NMDA, GABA}. \tag{32}$$

$\eta_\alpha(t)$ is a Gaussian process with zero mean, $\langle \eta_\alpha(t) \rangle = 0$, and an exponentially decaying correlation function, $\langle \eta_\alpha(t) \eta_\alpha(t') \rangle \propto \exp\left(-\left|t - t'\right| / \tau_{\mathrm{syn},\alpha}\right)$, which is due to synaptic filtering with effective time constant $\tau_{\mathrm{syn},\alpha}$ (**Fourcaud and Brunel, 2002**):

$$\tau_{\mathrm{syn},\alpha} = \frac{\sigma_\alpha^2}{\dfrac{\sigma_{\mathrm{X},\alpha}^2}{\tau_{\mathrm{AMPA}}} + \dfrac{\sigma_{\mathrm{AMPA},\alpha}^2}{\tau_{\mathrm{AMPA}}} + \dfrac{\sigma_{\mathrm{NMDA},\alpha}^2}{\tau_{\mathrm{NMDA}}} + \dfrac{\sigma_{\mathrm{GABA},\alpha}^2}{\tau_{\mathrm{GABA}}}}, \tag{33}$$

where $\tau_{\mathrm{AMPA}} = \tau_{\mathrm{AMPA,l}} + \tau_{\mathrm{AMPA,r}} + \tau_{\mathrm{AMPA,d}}$, $\tau_{\mathrm{NMDA}} = \tau_{\mathrm{NMDA,l}} + \tau_{\mathrm{NMDA,r}} + \tau_{\mathrm{NMDA,d}}$, $\tau_{\mathrm{GABA}} = \tau_{\mathrm{GABA,l}} + \tau_{\mathrm{GABA,r}} + \tau_{\mathrm{GABA,d}}$ are effective synaptic time constants for AMPAR, NMDAR, and GABAR-mediated currents, respectively. In addition, because of sparse connectivity, the correlation of the fluctuations in the synaptic inputs of excitatory and inhibitory populations is neglected: $\langle \eta_\mathrm{E}(t) \eta_\mathrm{I}(t') \rangle = 0$. The firing rate $\nu_\alpha$ of a neuron, whose potential is governed by **Equation 21**, is given by a current-frequency relationship $\phi_\alpha\left(\mu_\alpha, \sigma_\alpha\right)$ that is a function of the mean and fluctuating part of synaptic input (**Brunel and Sergi, 1998**; **Fourcaud and Brunel, 2002**):

$$\phi_\alpha\left(\mu_\alpha, \sigma_\alpha\right) = \left(\tau_{\mathrm{rp},\alpha} + \tau_\alpha \int_{a(\mu_\alpha,\sigma_\alpha)}^{b(\mu_\alpha,\sigma_\alpha)} dx \sqrt{\pi} \exp(x^2)\left(1 + \mathrm{erf}(x)\right)\right)^{-1}, \tag{34}$$

where

$$a\left(\mu_\alpha, \sigma_\alpha\right) = \frac{V_{\mathrm{rst}} - V_{\mathrm{L}} - \mu_\alpha}{\sigma_\alpha} \tag{35}$$

$$b\left(\mu_\alpha, \sigma_\alpha\right) = \frac{\theta - V_{\mathrm{L}} - \mu_\alpha}{\sigma_\alpha}\left(1 + 0.5\frac{\tau_{\mathrm{syn},\alpha}}{\tau_\alpha}\right) + 1.03\sqrt{\frac{\tau_{\mathrm{syn},\alpha}}{\tau_\alpha}} - 0.5\frac{\tau_{\mathrm{syn},\alpha}}{\tau_\alpha}. \tag{36}$$

Since $\mu_\alpha$ and $\sigma_\alpha$ themselves depend on the population firing rates $\nu_{\mathrm{E}}$ and $\nu_{\mathrm{I}}$, the two coupled frequency-current equations

$$\begin{cases} \nu_{\mathrm{E}} = \phi_{\mathrm{E}}\left(\mu_{\mathrm{E}}\left(\nu_{\mathrm{E}}, \nu_{\mathrm{I}}\right), \sigma_{\mathrm{E}}\left(\nu_{\mathrm{E}}, \nu_{\mathrm{I}}\right)\right) \\ \nu_{\mathrm{I}} = \phi_{\mathrm{I}}\left(\mu_{\mathrm{I}}\left(\nu_{\mathrm{E}}, \nu_{\mathrm{I}}\right), \sigma_{\mathrm{I}}\left(\nu_{\mathrm{E}}, \nu_{\mathrm{I}}\right)\right) \end{cases} \tag{37}$$

provide a self-consistent description of the network in stationary states, that is regimes of network dynamics when the population average quantities such as firing rates and synaptic inputs are constant in time. In the framework of our model, synaptic conductances $g_{\mathrm{X},\alpha}$, $g_{\mathrm{AMPA},\alpha}$, $g_{\mathrm{NMDA},\alpha}$, $g_{\mathrm{GABA},\alpha}$ ($\alpha = \mathrm{E}, \mathrm{I}$) and the external spike rate $\nu_{\mathrm{X}}$ are system parameters controlling the regime of network dynamics; they enter to the mean field analysis through expressions for $\mu_\alpha$, and $\sigma_\alpha$. If these parameters are given, one can solve the self-consistent equations to obtain predicted by the mean field approximation population firing rates $\nu_{\mathrm{E}}^0$ and $\nu_{\mathrm{I}}^0$ in a stationary state of the network. Conversely, once external $\nu_{\mathrm{X}}$ and population spike rates $\nu_{\mathrm{E}}^0$ and $\nu_{\mathrm{I}}^0$ are specified, the self-consistent equations could be solved to find the values of synaptic conductance parameters $g_{\mathrm{X},\alpha}$, $g_{\mathrm{AMPA},\alpha}$, $g_{\mathrm{NMDA},\alpha}$, $g_{\mathrm{GABA},\alpha}$ ($\alpha = \mathrm{E}, \mathrm{I}$) that correspond to these spike rates. However, because there are eight unknown parameters and only two equations, to find a unique solution one would need six additional equations imposing constraints on conductance parameters.

## Model parametrization

We derive three of these equations by implementing a commonly used constraint (e.g. *Brunel and Wang, 2003*; *Compte et al., 2000*) that equalizes the ratio of synaptic conductance parameters for component currents in excitatory and inhibitory neurons. Since each component current is proportional to its respective synaptic conductance, this constraint implies that the balance between different components of average synaptic currents $I_{\mathrm{X},\alpha}^0$, $I_{\mathrm{AMPA},\alpha}^0$, $I_{\mathrm{NMDA},\alpha}^0$, $I_{\mathrm{GABA},\alpha}^0$ for excitatory ($\alpha = \mathrm{E}$) and inhibitory ($\alpha = \mathrm{I}$) populations is the same, thus providing the following three equations:

$$\frac{I_{\mathrm{NMDA,E}}^0}{I_{\mathrm{GABA,E}}^0} = \frac{I_{\mathrm{NMDA,I}}^0}{I_{\mathrm{GABA,I}}^0}, \quad \frac{I_{\mathrm{AMPA,E}}^0}{I_{\mathrm{GABA,E}}^0} = \frac{I_{\mathrm{AMPA,I}}^0}{I_{\mathrm{GABA,I}}^0}, \quad \frac{I_{\mathrm{X,E}}^0}{I_{\mathrm{GABA,E}}^0} = \frac{I_{\mathrm{X,I}}^0}{I_{\mathrm{GABA,I}}^0}. \tag{38}$$

As a result, whenever the ratio of synaptic conductances and/or component currents is involved, the index $\alpha$ designating the type of the neuron can be dropped.

Two additional equations are obtained by fixing the balance between inhibition and two-component recurrent excitation at certain values:

$$\frac{I_{\mathrm{NMDA}}^0}{I_{\mathrm{GABA}}^0} = q_1, \qquad \frac{I_{\mathrm{AMPA}}^0}{I_{\mathrm{GABA}}^0} = q_2 \tag{39}$$

The last constraint is provided in terms of the relative magnitude of average external current of excitatory neurons, $I_{\mathrm{X,E}}^0$:

$$\frac{I_{\mathrm{X,E}}^0}{I_{\theta,\mathrm{E}}^0} = q_3, \tag{40}$$

where $I_{\theta,\mathrm{E}}^0$ is the current that is needed for an excitatory neuron to reach firing threshold $\theta$ in absence of recurrent feedback. This approach allowed to parametrize network dynamics in terms of three

parameters expressed as ratios of absolute values of average synaptic currents, $I_{\text{AMPA}}/I_{\text{GABA}}$, $I_{\text{NMDA}}/I_{\text{GABA}}$, and $I_{\text{X,E}}/I_{\theta,\text{E}}$, characterizing the balance between components of recurrent excitation and inhibition, and the balance between external input and firing threshold. For a given external spike rate $\nu_{\text{X}}$ and fixed values of these three parameters, we are now able to solve the self-consistent equations for the eight synaptic conductances that provide the prescribed population firing rates $\nu_{\text{E}}^0$ and $\nu_{\text{I}}^0$ in a stationary state of the network.

We are interested in the asynchronous stationary state in which neurons fire spikes irregularly and at low rates, like neurons in prefrontal cortex. When mean synaptic inputs $\mu_\alpha$ are well below threshold $\theta$, firing is driven by the synaptic fluctuations $\sigma_\alpha$ around the mean input, therefore, resulting in irregular spike trains and low rates (*Renart et al., 2003*). Given that the number of synaptic connections received by individual neurons is large and network connectivity is sparse, solutions of self-consistent equations providing the subthreshold regime for $\mu_\alpha$ and, thus, low rate asynchronous network dynamics, arise when inhibition strongly dominates recurrent excitation and the mean external inputs are around or above threshold $\theta$ (*Brunel, 2000*; *Renart et al., 2003*; *van Vreeswijk and Sompolinsky, 1996*). Thus, for the network to be in asynchronous irregular state the three system parameters characterizing the balance between recurrent excitation and inhibition, and the relative strength of external inputs should be within certain bounds: $I_{\text{AMPA}}/I_{\text{GABA}} + I_{\text{NMDA}}/I_{\text{GABA}} < 1$, and $I_{\text{X,E}}/I_{\theta,\text{E}} \gtrsim 1$.

## Linear stability analysis

We perform a linear stability analysis of the asynchronous state (*Abbott and van Vreeswijk, 1993*; *Brunel and Hakim, 1999*) on the basis of an analytical consideration in *Brunel and Wang, 2003*. To understand if the network develops instability caused by fluctuations in population firing rates, we consider small deviations from the stationary population rates $\nu_{\text{E}}^0$ and $\nu_{\text{I}}^0$. In order to analyze the resulting network behavior, the mean field approach and self-consistent equations providing population mean firing rates $\nu_{\text{E}}^0$ and $\nu_{\text{I}}^0$ are extended to describe the dynamics of population rates $\nu_{\text{E}}(t)$ and $\nu_{\text{I}}(t)$.

In the framework of mean field approximation, each component of synaptic current is determined by the product of effective synaptic strength $J$ and average gating variable $S$ (*Equations 17–20* for the steady state consideration). The dynamics of $S$ is governed by the same type of equations as for the gating variable $s$ of an individual synapse in a given postsynaptic neuron (*Equation 14*), except that the instantaneous rate of spikes $\sum_k \delta\left(t - t_k - \tau_l\right)$ arriving from the presynaptic cell is replaced by the instantaneous average rate of spikes, $C_{\alpha_R}\nu_{\alpha_R}\left(t - \tau_l\right)$, arriving from all presynaptic cells making the same type of synapse in the postsynaptic neuron:

$$\begin{cases} \tau_{\text{r}}\dfrac{dx(t)}{dt} = -x(t) + C_{\alpha_R}\nu_{\alpha_R}\left(t - \tau_l\right) \\ \tau_{\text{d}}\dfrac{dS_R(t)}{dt} = -S_R(t) + x(t) \end{cases}, \tag{41}$$

where $R$ designates the type of the synapse ($R = $ X, AMPA, NMDA, GABA), and $\alpha_R$ designates the presynaptic population establishing these synapses ($\alpha_R = $ X, E for glutamatergic and $\alpha_R = $ I for GABAergic synapse). Since external firing rate $\nu_{\text{X}}$ is stationary, the gating variable for external current is constant in time: $S_{\text{X}} = C_{\text{X}}\nu_{\text{X}}$. For recurrent currents, the temporal course of $S_R$ is dependent on the instantaneous presynaptic population activity $\nu_{\alpha_R}(t)$. Consequently, the total synaptic input current $I_{\text{syn}}(t)$, given as a sum of contributions from external and recurrent components

$$I_{\text{syn}}(t) = J_{\text{X}}S_{\text{X}} + J_{\text{AMPA}}S_{\text{AMPA}}(t) + J_{\text{NMDA}}S_{\text{NMDA}}(t) + J_{\text{GABA}}S_{\text{GABA}}(t), \tag{42}$$

depends on the population firing rates $\nu_{\text{E}}(t)$ and $\nu_{\text{I}}(t)$. The output firing rate of population neurons, in turn, is determined by the input current and can be modeled in terms of an input-output response function $F$.

In general, the input-output relationship $\nu(t) = F\left(I_{\text{syn}}(t)\right)$ depends on the spectral characteristics of the input current, resulting in frequency dependent phase shifts and/or amplitude modulations between the oscillatory components of $I_{\text{syn}}$ and $\nu$. However, it has been shown (*Brunel et al., 2001*; *Fourcaud and Brunel, 2002*) that the output rate in the leaky integrate and fire neuron model follows instantaneously the temporal variations in its synaptic input current given that synaptic noise is sufficiently strong and synaptic time constant is comparable with membrane time constant. That is, in

these conditions, the response does not exhibit a phase shift, and its amplitude is independent of the frequency of oscillatory components of the input current. As a result, even if the input current is varying in time, the input-output function $F$ can be approximated by the current-frequency response function $\phi$, given by *Equation 34*, describing the output due to the steady input current.

In the framework of mean field approximation, the output rates $\phi_E\left(I_{\text{syn,E}}(t)\right)$ and $\phi_I\left(I_{\text{syn,I}}(t)\right)$ for excitatory and inhibitory populations must be the same as the instantaneous presynaptic population rates $\nu_E(t)$ and $\nu_I(t)$ because both presynaptic and output rates are of the same populations. This requirement results in two self-consistent equations:

$$\begin{cases} \nu_E(t) = \phi_E\left(I_{\text{syn,E}}\left(\nu_E(t), \nu_I(t)\right)\right) \\ \nu_I(t) = \phi_I\left(I_{\text{syn,I}}\left(\nu_E(t), \nu_I(t)\right)\right) \end{cases}. \tag{43}$$

Since the amplitudes of firing rate deviations from the rates in asynchronous steady state are small, $\phi\left(I_{\text{syn}}(t)\right)$ can be linearized about the input current $I_{\text{syn}}^0$ in asynchronous state as:

$$\phi\left(I_{\text{syn}}(t)\right) \approx \phi\left(I_{\text{syn}}^0\right) + \frac{d\phi\left(I_{\text{syn}}^0\right)}{dI_{\text{syn}}}\left(I_{\text{syn}}(t) - I_{\text{syn}}^0\right). \tag{44}$$

With this approximation, the self-consistent equations for excitatory and inhibitory populations become

$$\begin{cases} \nu_E(t) = \nu_E^0\left(1 + A_E \dfrac{I_{\text{syn,E}}(t) - I_{\text{syn,E}}^0}{I_{\text{syn,E}}^0}\right) \\ \nu_I(t) = \nu_I^0\left(1 + A_I \dfrac{I_{\text{syn,I}}(t) - I_{\text{syn,I}}^0}{I_{\text{syn,I}}^0}\right) \end{cases}, \tag{45}$$

where $A_\alpha = \frac{I_{\text{syn},\alpha}^0}{\nu_\alpha^0}\frac{d\phi_\alpha\left(I_{\text{syn},\alpha}^0\right)}{dI_{\text{syn},\alpha}}$ is the dimensionless slope of the current-frequency response function at the current value in asynchronous state, expressed as the ratio between the relative changes in the firing rate and the input current (**Brunel and Wang, 2003**).

The self-consistent equations *Equation 45* together with *Equation 41* for the gating variables and *Equation 42* for the total synaptic current describe approximate firing rate dynamics of excitatory and inhibitory populations. To determine if the network develops oscillatory instability caused by small fluctuations in population firing rates, we seek solutions for the rates $\nu_E(t)$ and $\nu_I(t)$ in which initially small (with relative amplitudes $|\varepsilon_E| \ll 1$ and $|\varepsilon_I| \ll 1$) oscillatory perturbations that can change exponentially with time are added to the stationary rates $\nu_E^0$ and $\nu_I^0$ such that: $\nu_\alpha(t) = \nu_\alpha^0\left(1 + |\varepsilon_\alpha|\exp(\lambda t)\cos(\omega t + \varphi_\alpha)\right)$ or, equivalently, in complex form

$$\nu_\alpha(t) = \nu_\alpha^0\left(1 + \varepsilon_\alpha \exp(\lambda t + i\omega t)\right), \quad \alpha = E, I, \tag{46}$$

where $\lambda$ is the rate of perturbation growth, $\omega$ is the oscillation frequency, and $\varepsilon_\alpha$ is complex accounting for a possible shift in oscillation phase $\varphi_\alpha$ between the two populations. We can now replace the firing rates in *Equation 41* with these expressions to solve the two equations and determine the synaptic variables $S_R(t)$ for recurrent currents mediated by $R =$ AMPA, NMDA, GABA receptors:

$$S_R(t) = S_R^0\left[1 + \varepsilon_{\alpha_R}Q_R(\lambda, \omega)\exp\left(\lambda t + i\omega t - i\Phi_R(\lambda, \omega)\right)\right], \tag{47}$$

where

$$Q_R(\lambda, \omega) = \frac{\exp(-\lambda\tau_{R,\text{l}})}{\sqrt{\left((1 + \lambda\tau_{R,\text{r}})^2 + \omega^2\tau_{R,\text{r}}^2\right)\left((1 + \lambda\tau_{R,\text{d}})^2 + \omega^2\tau_{R,\text{d}}^2\right)}} \tag{48}$$

and

$$\Phi_R(\lambda, \omega) = \omega\tau_{R,\text{l}} + \text{atan}\left(\frac{\omega\tau_{R,\text{r}}}{1 + \lambda\tau_{R,\text{r}}}\right) + \text{atan}\left(\frac{\omega\tau_{R,\text{d}}}{1 + \lambda\tau_{R,\text{d}}}\right). \tag{49}$$

The components of synaptic currents and the total currents $I_{\text{syn,E}}(t)$ and $I_{\text{syn,I}}(t)$ can now be calculated and inserted into the linearized self-consistent *Equation 45* for population firing rates. Taking into account that the balance between the components of synaptic currents in excitatory and inhibitory populations is equal, we arrive at the following set of two equations

$$\begin{cases} X_{\text{AMPA}}(\lambda,\omega)\cos\left(\Phi_{\text{AMPA}}(\lambda,\omega)\right) + X_{\text{NMDA}}(\lambda,\omega)\cos\left(\Phi_{\text{NMDA}}(\lambda,\omega)\right) - X_{\text{GABA}}(\lambda,\omega)\cos\left(\Phi_{\text{GABA}}(\lambda,\omega)\right) = 1 \\ X_{\text{AMPA}}(\lambda,\omega)\sin\left(\Phi_{\text{AMPA}}(\lambda,\omega)\right) + X_{\text{NMDA}}(\lambda,\omega)\sin\left(\Phi_{\text{NMDA}}(\lambda,\omega)\right) - X_{\text{GABA}}(\lambda,\omega)\sin\left(\Phi_{\text{GABA}}(\lambda,\omega)\right) = 0 \end{cases}$$

(50)

and the relationship between the relative amplitudes:

$$\varepsilon_{\text{E}}A_{\text{I}} = \varepsilon_{\text{I}}A_{\text{E}},$$

(51)

where

$$X_{\text{AMPA}}(\lambda,\omega) = A_{\text{E}}\frac{I_{\text{AMPA}}}{I_{\text{syn}}}Q_{\text{AMPA}}(\lambda,\omega)$$

(52)

$$X_{\text{NMDA}}(\lambda,\omega) = A_{\text{E}}\frac{I_{\text{NMDA}}}{I_{\text{syn}}}Q_{\text{NMDA}}(\lambda,\omega)$$

(53)

$$X_{\text{GABA}}(\lambda,\omega) = A_{\text{I}}\frac{I_{\text{GABA}}}{I_{\text{syn}}}Q_{\text{GABA}}(\lambda,\omega).$$

(54)

Solving *Equation 50*, we obtain the rate of perturbation growth $\lambda$ and the oscillation frequency $\omega$. Because both $A_{\text{E}}$ and $A_{\text{I}}$ are real, the linear relationship between the amplitudes $\varepsilon_{\text{E}}$ and $\varepsilon_{\text{I}}$ given by *Equation 51* means that there is no phase lag between firing rates of excitatory and inhibitory populations.

## Analytical consideration of the dependence of oscillation growth rate on network parameters

To further elucidate how specifically synaptic conductances $g_{\text{AMPA}}$, $g_{\text{NMDA}}$, $g_{\text{GABA}}$, and external rate $v_{\text{X}}$ affect synchrony, we linearize the mean field equations *Equation 37* and equations *Equation 50* for the stability analysis around the point $\{g^*_{\text{AMPA},\{\text{E,I}\}}, g^*_{\text{NMDA},\{\text{E,I}\}}, g^*_{\text{GABA},\{\text{E,I}\}}, v^*_{\text{X}}\}$ corresponding to the critical state network where $\lambda = 0$. We then derive an analytical approximation for the oscillation growth rate $\lambda$ describing its dependence on the synaptic conductances and external rate in the vicinity of this point.

### Linearization of mean field equations

Approximate analytic description of the changes in the population firing rates $\Delta v_{\text{E}}$ and $\Delta v_{\text{I}}$ due to small changes in the synaptic conductances and external rate can be obtained by linearizing the current-frequency response function $\phi$, providing population firing rates $v_{\text{E}}$ and $v_{\text{I}}$ as a function of synaptic conductances and external rate. We note that the function $\phi$ (*Equations 34–36*) explicitly depends on the mean effective synaptic input $\mu$, synaptic noise $\sigma$, membrane time constant $\tau$, and synaptic time constant $\tau_{\text{syn}}$, which in turn depend on the synaptic conductances and external rate (*Equations 22–33*). Thus, changes in the firing rates $\Delta v_{\text{E}}$ and $\Delta v_{\text{I}}$ in response to small changes in the synaptic conductances $\Delta g_{\text{AMPA}}$, $\Delta g_{\text{NMDA}}$, $\Delta g_{\text{GABA}}$, and external rate $\Delta v_{\text{X}}$ can be approximated as:

$$\Delta v_{\alpha} = \frac{d\phi_{\alpha}}{d\mu_{\alpha}}\Delta\mu_{\alpha} + \frac{d\phi_{\alpha}}{d\sigma_{\alpha}}\Delta\sigma_{\alpha} + \frac{d\phi_{\alpha}}{d\tau_{\alpha}}\Delta\tau_{\alpha} + \frac{d\phi_{\alpha}}{d\tau_{\text{syn},\alpha}}\Delta\tau_{\text{syn},\alpha}, \qquad \alpha = \text{E, I}.$$

(55)

The dominant contribution to $\Delta v_{\alpha}$ is due to the change in synaptic input, $\Delta\mu_{\alpha}$. Contributions from the remaining terms are relatively small, with the largest contribution being due to the change in the effective membrane time constant, $\Delta\tau_{\alpha}$. Therefore, the expression for $\Delta v_{\alpha}$ can be simplified by retaining only the terms involving $\Delta\mu_{\alpha}$ and $\Delta\tau_{\alpha}$:

$$\Delta v_{\alpha} \approx \frac{d\phi_{\alpha}}{d\mu_{\alpha}}\Delta\mu_{\alpha} + \frac{d\phi_{\alpha}}{d\tau_{\alpha}}\Delta\tau_{\alpha}, \qquad \alpha = \text{E, I}.$$

(56)

$\Delta\mu_\alpha$ and $\Delta\tau_\alpha$ are expressed through the relative changes in synaptic conductances $\Delta g_{\text{AMPA}}/g^*_{\text{AMPA}}$, $\Delta g_{\text{NMDA}}/g^*_{\text{NMDA}}$, $\Delta g_{\text{GABA}}/g^*_{\text{GABA}}$, external rate $\Delta\nu_X/\nu^*_X$, and the changes in population rates $\Delta\nu_E$ and $\Delta\nu_I$:

$$\Delta\mu_\alpha = a^\mu_{\alpha E}\Delta\nu_E + a^\mu_{\alpha I}\Delta\nu_I + b^\mu_{X,\alpha}\frac{\Delta\nu_X}{\nu^*_X} + \sum_R b^\mu_{R,\alpha}\frac{\Delta g_R}{g^*_R} \tag{57}$$

$$\Delta\tau_\alpha = a^\tau_{\alpha E}\Delta\nu_E + a^\tau_{\alpha I}\Delta\nu_I + b^\tau_{X,\alpha}\frac{\Delta\nu_X}{\nu^*_X} + \sum_R b^\tau_{R,\alpha}\frac{\Delta g_R}{g^*_R}, \tag{58}$$

where $R = \{\text{AMPA, NMDA, GABA}\}$, and

$$\begin{aligned} a^\mu_{\alpha E} &= \frac{(T_{\text{AMPA},\alpha} + T_{\text{NMDA1},\alpha})(V_E - V_L) + T_{\text{NMDA2},\alpha}(V_0 - V_L)}{S_\alpha} \\ &\quad - \frac{\mu_\alpha(T_{\text{AMPA},\alpha} + T_{\text{NMDA1},\alpha} + T_{\text{NMDA2},\alpha})}{S_\alpha} \end{aligned} \tag{59}$$

$$a^\mu_{\alpha I} = \frac{T_{\text{GABA},\alpha}(V_I - \mu_\alpha - V_L)}{S_\alpha} \tag{60}$$

$$b^\mu_{\text{AMPA},\alpha} = \frac{T_{\text{AMPA},\alpha}(V_E - \mu_\alpha - V_L)}{S_\alpha}\nu_E \tag{61}$$

$$b^\mu_{\text{NMDA},\alpha} = \frac{T_{\text{NMDA1},\alpha}(V_E - \mu_\alpha - V_L) + T_{\text{NMDA2},\alpha}(V_0 - \mu_\alpha - V_L)}{S_\alpha}\nu_E \tag{62}$$

$$b^\mu_{\text{GABA},\alpha} = \frac{T_{\text{GABA},\alpha}(V_I - \mu_\alpha - V_L)}{S_\alpha}\nu_I \tag{63}$$

$$b^\mu_{X,\alpha} = \frac{T_{X,\alpha}(V_E - \mu_\alpha - V_L)}{S_\alpha}\nu_X \tag{64}$$

$$a^\tau_{\alpha E} = -\frac{T_{\text{AMPA},\alpha} + T_{\text{NMDA1},\alpha} + T_{\text{NMDA2},\alpha}}{S_\alpha}\tau_\alpha \tag{65}$$

$$a^\tau_{\alpha I} = -\frac{T_{\text{GABA},\alpha}}{S_\alpha}\tau_\alpha \tag{66}$$

$$b^\tau_{\text{AMPA},\alpha} = -\frac{T_{\text{AMPA},\alpha}}{S_\alpha}\tau_\alpha\nu_E \tag{67}$$

$$b^\tau_{\text{NMDA},\alpha} = -\frac{T_{\text{NMDA1},\alpha} + T_{\text{NMDA2},\alpha}}{S_\alpha}\tau_\alpha\nu_E \tag{68}$$

$$b^\tau_{\text{GABA},\alpha} = -\frac{T_{\text{GABA},\alpha}}{S_\alpha}\tau_\alpha\nu_I \tag{69}$$

$$b^\tau_{X,\alpha} = -\frac{T_{X,\alpha}}{S_\alpha}\tau_\alpha\nu_X. \tag{70}$$

Inserting expressions for $\Delta\mu_\alpha$ and $\Delta\tau_\alpha$ into **Equation 56**, we obtain a closed system of linear equations for the changes in the firing rates of excitatory and inhibitory populations in response to small changes in the synaptic conductances and external rates. In matrix form these equations can be written as

$$\Delta\mathbf{v} = \mathbf{a}\Delta\mathbf{v} + \mathbf{b}\Delta\mathbf{p}, \tag{71}$$

where

$$\mathbf{a} = \begin{bmatrix} \phi'_{\mu,E}a^\mu_{EE} + \phi'_{\tau,E}a^\tau_{EE} & \phi'_{\mu,E}a^\mu_{EI} + \phi'_{\tau,E}a^\tau_{EI} \\ \phi'_{\mu,I}a^\mu_{IE} + \phi'_{\tau,I}a^\tau_{IE} & \phi'_{\mu,I}a^\mu_{II} + \phi'_{\tau,I}a^\tau_{II} \end{bmatrix}, \quad \Delta\mathbf{v} = \begin{bmatrix} \Delta\nu_E \\ \Delta\nu_I \end{bmatrix} \tag{72}$$

$$
b^T = \begin{bmatrix}
\phi'_{\mu,E}b^{\mu}_{X,E} + \phi'_{\tau,E}b^{\tau}_{X,E} & \phi'_{\mu,I}b^{\mu}_{X,I} + \phi'_{\tau,I}b^{\tau}_{X,I} \\
\phi'_{\mu,E}b^{\mu}_{AMPA,E} + \phi'_{\tau,E}b^{\tau}_{AMPA,E} & \phi'_{\mu,I}b^{\mu}_{AMPA,I} + \phi'_{\tau,I}b^{\tau}_{AMPA,I} \\
\phi'_{\mu,E}b^{\mu}_{NMDA,E} + \phi'_{\tau,E}b^{\tau}_{NMDA,E} & \phi'_{\mu,I}b^{\mu}_{NMDA,I} + \phi'_{\tau,I}b^{\tau}_{NMDA,I} \\
\phi'_{\mu,E}b^{\mu}_{GABA,E} + \phi'_{\tau,E}b^{\tau}_{GABA,E} & \phi'_{\mu,I}b^{\mu}_{GABA,I} + \phi'_{\tau,I}b^{\tau}_{GABA,I}
\end{bmatrix}, \quad
\Delta\mathbf{p} = \begin{bmatrix}
\Delta v_X/v^*_X \\
\Delta g_{AMPA}/g^*_{AMPA} \\
\Delta g_{NMDA}/g^*_{NMDA} \\
\Delta g_{GABA}/g^*_{GABA}
\end{bmatrix}.
$$

(73)

Here, the elements of matrices $\mathbf{a}$ and $\mathbf{b}$ are constants defined by the point in the network parameter space around which the mean field equations are linearized. Components of the vector $\Delta\mathbf{v}$ are the changes in the firing rates of excitatory and inhibitory populations due to the changes in the synaptic conductances and external rate given by the components of vector $\Delta\mathbf{p}$. Taking into account that $\phi'_{\tau,\alpha}b^{\tau}_{R,\alpha} \ll \phi'_{\mu,\alpha}b^{\mu}_{R,\alpha}$ and that $\mu_\alpha + V_L \approx \langle V_\alpha \rangle$, we neglect the $\phi'_{\tau,\alpha}b^{\tau}_{R,\alpha}$ terms in $\mathbf{b}$ and replace $\mu_\alpha + V_L$ with $\langle V_\alpha \rangle$. With these approximations $\mathbf{b}$ simplifies to:

$$
b \approx \mathbf{b}_0 \begin{bmatrix}
I_X/I_{GABA} \\
I_{AMPA}/I_{GABA} \\
I_{NMDA}/I_{GABA} \\
-1
\end{bmatrix}^T, \quad
\mathbf{b}_0 = \begin{bmatrix}
\phi'_{\mu,E}I_{GABA,E}/g_{m,E}S_E \\
\phi'_{\mu,I}I_{GABA,I}/g_{m,I}S_I
\end{bmatrix}.
$$

(74)

Equation *Equation 71* can now be rewritten as

$$
(\mathbf{a} - \mathbf{I})\,\Delta\mathbf{v} + \mathbf{b}_0 \begin{bmatrix}
I_X/I_{GABA} \\
I_{AMPA}/I_{GABA} \\
I_{NMDA}/I_{GABA} \\
-1
\end{bmatrix}^T \Delta\mathbf{p} = 0,
$$

(75)

where $\mathbf{I}$ is the identity matrix. Solving this equation for $\Delta\mathbf{v}$ we obtain

$$
\Delta\mathbf{v} = \mathbf{W} \begin{bmatrix}
I_X/I_{GABA} \\
I_{AMPA}/I_{GABA} \\
I_{NMDA}/I_{GABA} \\
-1
\end{bmatrix}^T \Delta\mathbf{p},
$$

(76)

or in component form

$$
\Delta v_E = W_E \left( \frac{I_X}{I_{GABA}} \frac{\Delta v_X}{v^*_X} + \frac{I_{AMPA}}{I_{GABA}} \frac{\Delta g_{AMPA}}{g^*_{AMPA}} + \frac{I_{NMDA}}{I_{GABA}} \frac{\Delta g_{NMDA}}{g^*_{NMDA}} - \frac{\Delta g_{GABA}}{g^*_{GABA}} \right)
$$

(77)

$$
\Delta v_I = W_I \left( \frac{I_X}{I_{GABA}} \frac{\Delta v_X}{v^*_X} + \frac{I_{AMPA}}{I_{GABA}} \frac{\Delta g_{AMPA}}{g^*_{AMPA}} + \frac{I_{NMDA}}{I_{GABA}} \frac{\Delta g_{NMDA}}{g^*_{NMDA}} - \frac{\Delta g_{GABA}}{g^*_{GABA}} \right),
$$

(78)

where $\mathbf{W} = \begin{bmatrix} W_E & W_I \end{bmatrix}^T$ is given by

$$
\mathbf{W} = -(\mathbf{a} - \mathbf{I})^{-1}\mathbf{b}_0.
$$

(79)

In summary, equations *Equations 77, 78* describe changes in the excitatory, $\Delta v_E$, and inhibitory, $\Delta v_I$, population firing rates due to the small relative changes in the synaptic conductances $\Delta g_{AMPA}/g^*_{AMPA}$, $\Delta g_{NMDA}/g^*_{NMDA}$, $\Delta g_{GABA}/g^*_{GABA}$, and external rate $\Delta v_X/v^*_X$.

## Linearization of equations for oscillatory instability analysis

Changes in synaptic parameters result not only in the changes of population firing rates, but also affect the stability of population dynamics. To understand the precise role played by the synaptic conductances and external input in the destabilization of the steady dynamics and emergence of network oscillation near the boundary between asynchronous and synchronous states, we derive an

approximate analytic description of the change in the rate of oscillatory instability growth $\Delta\lambda$ and the change in the oscillation frequency $\Delta\omega$ caused by small changes in the synaptic conductances and external rate. For this purpose, we linearize equations *Equation 50* for $\lambda$ and $\omega$ around the point $\{g^*_{\mathrm{AMPA},\{E,I\}}, g^*_{\mathrm{NMDA},\{E,I\}}, g^*_{\mathrm{GABA},\{E,I\}}, \nu^*_X\}$ corresponding to the critical state network that is on the boundary between steady and oscillatory states where $\lambda = 0$. We do this by taking the differentials with respect to the synaptic variables $\Phi_R$ and $X_R$, ($R$ = AMPA, NMDA, GABA) that, in turn, depend on $\lambda$ and $\omega$:

$$\begin{cases} \Delta X_{\mathrm{AMPA}} \cos\left(\Phi_{\mathrm{AMPA}}\right) - X_{\mathrm{AMPA}} \sin\left(\Phi_{\mathrm{AMPA}}\right) \Delta\Phi_{\mathrm{AMPA}} + \Delta X_{\mathrm{NMDA}} \cos\left(\Phi_{\mathrm{NMDA}}\right) \\ -X_{\mathrm{NMDA}} \sin\left(\Phi_{\mathrm{NMDA}}\right) \Delta\Phi_{\mathrm{NMDA}} - \Delta X_{\mathrm{GABA}} \cos\left(\Phi_{\mathrm{GABA}}\right) + X_{\mathrm{GABA}} \sin\left(\Phi_{\mathrm{GABA}}\right) \Delta\Phi_{\mathrm{GABA}} = 0 \\ \\ \Delta X_{\mathrm{AMPA}} \sin\left(\Phi_{\mathrm{AMPA}}\right) + X_{\mathrm{AMPA}} \cos\left(\Phi_{\mathrm{AMPA}}\right) \Delta\Phi_{\mathrm{AMPA}} + \Delta X_{\mathrm{NMDA}} \sin\left(\Phi_{\mathrm{NMDA}}\right) \\ +X_{\mathrm{NMDA}} \cos\left(\Phi_{\mathrm{NMDA}}\right) \Delta\Phi_{\mathrm{NMDA}} - \Delta X_{\mathrm{GABA}} \sin\left(\Phi_{\mathrm{GABA}}\right) - X_{\mathrm{GABA}} \cos\left(\Phi_{\mathrm{GABA}}\right) \Delta\Phi_{\mathrm{GABA}} = 0 \end{cases}, \tag{80}$$

The parameter $X_R$ (see *Equations 52–54*) characterizes the relative attenuation in the strength of the underlying synapse due to the $R$-current dynamics. In addition to the dependency on $\lambda$ and $\omega$ through $Q_R$ (*Equation 48*), $X_R$ depends directly on its corresponding synaptic conductance $g_R$ and indirectly on all the synaptic conductances and external rate through its dependency on the slope $\phi'_{I_{\mathrm{syn}},\alpha_R}$ of the current-frequency response function. The change $\Delta X_R$ due to small variations in the synaptic conductances and external rate is given by

$$\Delta X_R = X_R \left( \frac{\Delta g_R}{g^*_R} + \frac{\Delta\phi'_{I_{\mathrm{syn}},\alpha_R}}{\phi'_{I_{\mathrm{syn}},\alpha_R}} + \frac{\Delta Q_R}{Q_R} \right), \tag{81}$$

where $\phi'_{I_{\mathrm{syn}},\alpha_R}$, $X_R$, $Q_R$, and $\Phi_R$ are constants whose values are defined by the point $\{g^*_{\mathrm{AMPA},\{E,I\}}, g^*_{\mathrm{NMDA},\{E,I\}}, g^*_{\mathrm{GABA},\{E,I\}}, \nu^*_X\}$ in the synaptic parameter space around which the stability analysis equations are linearized.

The relative change $\Delta Q_R/Q_R$ can be obtained from *Equation 48*:

$$\frac{\Delta Q_R}{Q_R} = -\tau^{(1)}_R \Delta\lambda - \tau^{(2)}_R \Delta\omega, \tag{82}$$

and the change in $\Phi_R$ from *Equation 49*:

$$\Delta\Phi_R = \tau^{(1)}_R \Delta\omega - \tau^{(2)}_R \Delta\lambda, \tag{83}$$

where

$$\tau^{(1)}_R = \tau_{R,\mathrm{l}} + \frac{\tau_{R,\mathrm{r}}}{1 + \left(\omega\tau_{R,\mathrm{r}}\right)^2} + \frac{\tau_{R,\mathrm{d}}}{1 + \left(\omega\tau_{R,\mathrm{d}}\right)^2} \tag{84}$$

$$\tau^{(2)}_R = \omega \left( \frac{\tau^2_{R,\mathrm{r}}}{1 + \left(\omega\tau_{R,\mathrm{r}}\right)^2} + \frac{\tau^2_{R,\mathrm{d}}}{1 + \left(\omega\tau_{R,\mathrm{d}}\right)^2} \right), \tag{85}$$

and $\omega$ is the oscillation frequency at the critical state. Inserting expressions for $\Delta Q_R/Q_R$ into equations *Equation 81* for $\Delta X_R$ and, subsequently, expressions for $\Delta\Phi_R$ and $\Delta X_R$ into equations *Equation 80*, we obtain a system of two linear equations for $\Delta\lambda$ and $\Delta\omega$:

$$\begin{cases} T_+ \Delta\omega + T_- \Delta\lambda = \Delta\xi_{\mathrm{AMPA}} + \Delta\xi_{\mathrm{NMDA}} - \Delta\xi_{\mathrm{GABA}} \\ T_- \Delta\omega - T_+ \Delta\lambda = \Delta\zeta_{\mathrm{AMPA}} + \Delta\zeta_{\mathrm{NMDA}} - \Delta\zeta_{\mathrm{GABA}} \end{cases}, \tag{86}$$

where

$$T_+ = X_{\mathrm{AMPA}} \tau^+_{\mathrm{AMPA}} + X_{\mathrm{NMDA}} \tau^+_{\mathrm{NMDA}} - X_{\mathrm{GABA}} \tau^+_{\mathrm{GABA}} \tag{87}$$

$$T_- = X_{\text{AMPA}} \tau_{\text{AMPA}}^- + X_{\text{NMDA}} \tau_{\text{NMDA}}^- - X_{\text{GABA}} \tau_{\text{GABA}}^- \tag{88}$$

$$\tau_R^+ = \tau_R^{(1)} \sin(\Phi_R) + \tau_R^{(2)} \cos(\Phi_R) \tag{89}$$

$$\tau_R^- = \tau_R^{(1)} \cos(\Phi_R) - \tau_R^{(2)} \sin(\Phi_R) \tag{90}$$

$$\Delta \xi_R = X_R \cos(\Phi_R) \left( \frac{\Delta g_R}{g_R^*} + \frac{\Delta \phi'_{I_{\text{syn}, \alpha_R}}}{\phi'_{I_{\text{syn}, \alpha_R}}} \right) \tag{91}$$

$$\Delta \zeta_R = -X_R \sin(\Phi_R) \left( \frac{\Delta g_R}{g_R^*} + \frac{\Delta \phi'_{I_{\text{syn}, \alpha_R}}}{\phi'_{I_{\text{syn}, \alpha_R}}} \right) \tag{92}$$

and $R = $ AMPA, NMDA, GABA. Solving the system of equations **Equation 86** for $\Delta \lambda$ and $\Delta \omega$ we obtain:

$$\Delta \lambda = \Lambda_{\text{AMPA}} \left( \frac{\Delta g_{\text{AMPA}}}{g_{\text{AMPA}}^*} + \frac{\Delta \phi'_{I_{\text{syn},E}}}{\phi'_{I_{\text{syn},E}}} \right) + \Lambda_{\text{NMDA}} \left( \frac{\Delta g_{\text{NMDA}}}{g_{\text{NMDA}}^*} + \frac{\Delta \phi'_{I_{\text{syn},E}}}{\phi'_{I_{\text{syn},E}}} \right) - \Lambda_{\text{GABA}} \left( \frac{\Delta g_{\text{GABA}}}{g_{\text{GABA}}^*} + \frac{\Delta \phi'_{I_{\text{syn},I}}}{\phi'_{I_{\text{syn},I}}} \right) \tag{93}$$

$$\Delta \omega = \Omega_{\text{AMPA}} \left( \frac{\Delta g_{\text{AMPA}}}{g_{\text{AMPA}}^*} + \frac{\Delta \phi'_{I_{\text{syn},E}}}{\phi'_{I_{\text{syn},E}}} \right) + \Omega_{\text{NMDA}} \left( \frac{\Delta g_{\text{NMDA}}}{g_{\text{NMDA}}^*} + \frac{\Delta \phi'_{I_{\text{syn},E}}}{\phi'_{I_{\text{syn},E}}} \right) - \Omega_{\text{GABA}} \left( \frac{\Delta g_{\text{GABA}}}{g_{\text{GABA}}^*} + \frac{\Delta \phi'_{I_{\text{syn},I}}}{\phi'_{I_{\text{syn},I}}} \right). \tag{94}$$

Here, $\Lambda_R$ and $\Omega_R$ are constants defined by the parameters of the critical state network around which the stability analysis equations are linearized:

$$\Lambda_R = \frac{X_R}{T_+^2 + T_-^2} \left( T_+ \sin(\Phi_R) + T_- \cos(\Phi_R) \right) \tag{95}$$

$$\Omega_R = \frac{X_R}{T_+^2 + T_-^2} \left( T_+ \cos(\Phi_R) - T_- \sin(\Phi_R) \right), \tag{96}$$

or, equivalently,

$$\Lambda_R = \frac{X_R}{T_0} \cos(\Phi_R + \Phi_0) \tag{97}$$

$$\Omega_R = -\frac{X_R}{T_0} \sin(\Phi_R + \Phi_0), \tag{98}$$

where $T_0 = \sqrt{T_+^2 + T_-^2}$ and $\Phi_0 = -\text{atan}(T_+/T_-)$.

Note that while $\Delta \lambda$ and $\Delta \omega$ given by equations **Equations 93 and 94** depend directly on the changes in the synaptic conductances, they also depend indirectly on these parameters and the change in external rate through the terms involving $\Delta \phi'_{I_{\text{syn},E}}$ and $\Delta \phi'_{I_{\text{syn},I}}$ characterizing changes in the slopes of the current-frequency response functions of excitatory and inhibitory neurons. To calculate these changes due to the changes in the synaptic conductances and external rate, we note that

$$\frac{\Delta \phi'_{I_{\text{syn}}}}{\phi'_{I_{\text{syn}}}} = \frac{\Delta \left( \frac{d\phi}{d\mu} \frac{d\mu}{dI_{\text{syn}}} \right)}{\frac{d\phi}{d\mu} \frac{d\mu}{dI_{\text{syn}}}} = \frac{\frac{d\mu}{dI_{\text{syn}}} \Delta \left( \frac{d\phi}{d\mu} \right) + \frac{d\phi}{d\mu} \Delta \left( \frac{d\mu}{dI_{\text{syn}}} \right)}{\frac{d\phi}{d\mu} \frac{d\mu}{dI_{\text{syn}}}} = \frac{\Delta \phi'_{\mu}}{\phi'_{\mu}} + \frac{\Delta \left( \frac{d\mu}{dI_{\text{syn}}} \right)}{\frac{d\mu}{dI_{\text{syn}}}}. \tag{99}$$

Taking into account the linear relationship $\mu \sim -I_{\text{syn}}/g_m S$ between the effective synaptic input $\mu$ and total synaptic current $I_{\text{syn}}$, we arrive at

$$\frac{\Delta \phi'_{I_{\text{syn}, \alpha}}}{\phi'_{I_{\text{syn}, \alpha}}} = \frac{\Delta \phi'_{\mu, \alpha}}{\phi'_{\mu, \alpha}} - \frac{\Delta S_\alpha}{S_\alpha}, \qquad \alpha = \text{E, I.} \tag{100}$$

As in the case of the change in the current-frequency response function $\Delta \phi_{\mu, \alpha}$, the dominant contribution to the change in the slope of the response function $\Delta \phi'_{\mu, \alpha}$ is coming from the change in the

synaptic input $\Delta\mu_\alpha$, while the change in the effective membrane time constant $\Delta\tau_\alpha$, similarly, is the next largest contribution. Therefore, $\Delta\phi'_{\mu,\alpha}$ can be approximated as

$$\Delta\phi'_{\mu,\alpha} \approx \frac{d^2\phi_\alpha}{d\mu_\alpha^2}\Delta\mu_\alpha + \frac{d^2\phi_\alpha}{d\tau_\alpha d\mu_\alpha}\Delta\tau_\alpha. \tag{101}$$

Note also that using *Equation 22* one can express the relative change $\Delta S_\alpha/S_\alpha$ through the change $\Delta\tau_\alpha$ as $\Delta S_\alpha/S_\alpha = -\Delta\tau_\alpha/\tau_\alpha$. Inserting expressions for $\Delta\phi'_{\mu,\alpha}$ and $\Delta S_\alpha/S_\alpha$ into *Equation 100* we obtain

$$\frac{\Delta\phi'_{I_{\mathrm{syn},\alpha}}}{\phi'_{I_{\mathrm{syn},\alpha}}} = \frac{\phi''_{\mu\mu,\alpha}}{\phi'_{\mu,\alpha}}\Delta\mu_\alpha + \left(\frac{\phi''_{\tau\mu,\alpha}}{\phi'_{\mu,\alpha}} + \frac{1}{\tau_\alpha}\right)\Delta\tau_\alpha. \tag{102}$$

Equations for $\Delta\mu_\alpha$ and $\Delta\tau_\alpha$ in terms of the changes in the synaptic conductances, external rate, and the resulting changes in the population firing rates $\Delta\nu_E$ and $\Delta\nu_I$ have been already derived and are given by *Equations 57 and 58*. We replace $\Delta\nu_E$ and $\Delta\nu_I$ in these equations with the solution obtained from the linearization of the mean field equations given, respectively, by *Equations 77, 78*. Next, by inserting the resulting $\Delta\mu_\alpha$ and $\Delta\tau_\alpha$ into *Equation 102*, we obtain expressions describing the relative changes in the slopes of the response functions for excitatory and inhibitory neurons due to the small changes in the synaptic conductances and external rate. In matrix form, these expressions can be written as

$$\begin{bmatrix} \dfrac{\Delta\phi'_{I_{\mathrm{syn},E}}}{\phi'_{I_{\mathrm{syn},E}}} \\[2ex] \dfrac{\Delta\phi'_{I_{\mathrm{syn},I}}}{\phi'_{I_{\mathrm{syn},I}}} \end{bmatrix} = -\widetilde{\mathbf{a}}\mathbf{W}\begin{bmatrix} I_X/I_{\mathrm{GABA}} \\ I_{\mathrm{AMPA}}/I_{\mathrm{GABA}} \\ I_{\mathrm{NMDA}}/I_{\mathrm{GABA}} \\ -1 \end{bmatrix}^T \Delta\mathbf{p} + \widetilde{\mathbf{b}}\Delta\mathbf{p}, \tag{103}$$

where

$$\widetilde{\mathbf{a}} = \begin{bmatrix} \dfrac{\phi''_{\mu\mu,E}}{\phi'_{\mu,E}}a^\mu_{\mathrm{EE}} + \left(\dfrac{\phi''_{\tau\mu,E}}{\phi'_{\mu,E}} + \dfrac{1}{\tau_E}\right)a^\tau_{\mathrm{EE}} & \dfrac{\phi''_{\mu\mu,E}}{\phi'_{\mu,E}}a^\mu_{\mathrm{EI}} + \left(\dfrac{\phi''_{\tau\mu,E}}{\phi'_{\mu,E}} + \dfrac{1}{\tau_E}\right)a^\tau_{\mathrm{EI}} \\[3ex] \dfrac{\phi''_{\mu\mu,I}}{\phi'_{\mu,I}}a^\mu_{\mathrm{IE}} + \left(\dfrac{\phi''_{\tau\mu,I}}{\phi'_{\mu,I}} + \dfrac{1}{\tau_I}\right)a^\tau_{\mathrm{IE}} & \dfrac{\phi''_{\mu\mu,I}}{\phi'_{\mu,I}}a^\mu_{\mathrm{II}} + \left(\dfrac{\phi''_{\tau\mu,I}}{\phi'_{\mu,I}} + \dfrac{1}{\tau_I}\right)a^\tau_{\mathrm{II}} \end{bmatrix} \tag{104}$$

$$\widetilde{\mathbf{b}}^T = \begin{bmatrix} \dfrac{\phi''_{\mu\mu,E}}{\phi'_{\mu,E}}b^\mu_{\mathrm{X,E}} + \left(\dfrac{\phi''_{\tau\mu,E}}{\phi'_{\mu,E}} + \dfrac{1}{\tau_E}\right)b^\tau_{\mathrm{X,E}} & \dfrac{\phi''_{\mu\mu,I}}{\phi'_{\mu,I}}b^\mu_{\mathrm{X,I}} + \left(\dfrac{\phi''_{\tau\mu,I}}{\phi'_{\mu,I}} + \dfrac{1}{\tau_I}\right)b^\tau_{\mathrm{X,I}} \\[3ex] \dfrac{\phi''_{\mu\mu,E}}{\phi'_{\mu,E}}b^\mu_{\mathrm{AMPA,E}} + \left(\dfrac{\phi''_{\tau\mu,E}}{\phi'_{\mu,E}} + \dfrac{1}{\tau_E}\right)b^\tau_{\mathrm{AMPA,E}} & \dfrac{\phi''_{\mu\mu,I}}{\phi'_{\mu,I}}b^\mu_{\mathrm{AMPA,I}} + \left(\dfrac{\phi''_{\tau\mu,I}}{\phi'_{\mu,I}} + \dfrac{1}{\tau_I}\right)b^\tau_{\mathrm{AMPA,I}} \\[3ex] \dfrac{\phi''_{\mu\mu,E}}{\phi'_{\mu,E}}b^\mu_{\mathrm{NMDA,E}} + \left(\dfrac{\phi''_{\tau\mu,E}}{\phi'_{\mu,E}} + \dfrac{1}{\tau_E}\right)b^\tau_{\mathrm{NMDA,E}} & \dfrac{\phi''_{\mu\mu,I}}{\phi'_{\mu,I}}b^\mu_{\mathrm{NMDA,I}} + \left(\dfrac{\phi''_{\tau\mu,I}}{\phi'_{\mu,I}} + \dfrac{1}{\tau_I}\right)b^\tau_{\mathrm{NMDA,I}} \\[3ex] \dfrac{\phi''_{\mu\mu,E}}{\phi'_{\mu,E}}b^\mu_{\mathrm{GABA,E}} + \left(\dfrac{\phi''_{\tau\mu,E}}{\phi'_{\mu,E}} + \dfrac{1}{\tau_E}\right)b^\tau_{\mathrm{GABA,E}} & \dfrac{\phi''_{\mu\mu,I}}{\phi'_{\mu,I}}b^\mu_{\mathrm{GABA,I}} + \left(\dfrac{\phi''_{\tau\mu,I}}{\phi'_{\mu,I}} + \dfrac{1}{\tau_I}\right)b^\tau_{\mathrm{GABA,I}} \end{bmatrix}. \tag{105}$$

The elements of matrices $\widetilde{\mathbf{a}}$ and $\widetilde{\mathbf{b}}$ are constants defined by the parameters of the critical state network. Noting that $\left(\dfrac{\phi''_{\tau\mu,\alpha}}{\phi'_{\mu,\alpha}} + \dfrac{1}{\tau_\alpha}\right)b^\tau_{R,\alpha} \ll \dfrac{\phi''_{\mu\mu,\alpha}}{\phi'_{\mu,\alpha}}b^\mu_{R,\alpha}$, we neglect the $\left(\dfrac{\phi''_{\tau\mu,\alpha}}{\phi'_{\mu,\alpha}} + \dfrac{1}{\tau_\alpha}\right)b^\tau_{R,\alpha}$ terms in $\widetilde{\mathbf{b}}$ and, as in the calculation of the change in the current-frequency response function, replace $\mu_\alpha + V_L$ with $\langle V_\alpha\rangle$. With these approximations $\widetilde{\mathbf{b}}$ simplifies to:

$$\widetilde{\mathbf{b}} \approx \widetilde{\mathbf{b}}_0 \begin{bmatrix} I_X/I_{GABA} \\ I_{AMPA}/I_{GABA} \\ I_{NMDA}/I_{GABA} \\ -1 \end{bmatrix}^T, \quad \widetilde{\mathbf{b}}_0 = \begin{bmatrix} \dfrac{\phi''_{\mu\mu,E}}{\phi'_{\mu,E}} \dfrac{I_{GABA,E}}{g_{m,E}S_E} \\ \dfrac{\phi'_{\mu\mu,I}}{\phi'_{\mu,I}} \dfrac{I_{GABA,I}}{g_{m,I}S_I} \end{bmatrix}. \tag{106}$$

Equation *Equation 103* can now be written as

$$\begin{bmatrix} \dfrac{\Delta\phi'_{I_{syn,E}}}{\phi'_{I_{syn,E}}} \\ \dfrac{\Delta\phi'_{I_{syn,I}}}{\phi'_{I_{syn,I}}} \end{bmatrix} = \mathbf{U} \begin{bmatrix} I_X/I_{GABA} \\ I_{AMPA}/I_{GABA} \\ I_{NMDA}/I_{GABA} \\ -1 \end{bmatrix}^T \Delta\mathbf{p}, \tag{107}$$

or in component form

$$\frac{\Delta\phi'_{I_{syn,E}}}{\phi'_{I_{syn,E}}} = U_E \left( \frac{I_X}{I_{GABA}} \frac{\Delta v_X}{v_X^*} + \frac{I_{AMPA}}{I_{GABA}} \frac{\Delta g_{AMPA}}{g_{AMPA}^*} + \frac{I_{NMDA}}{I_{GABA}} \frac{\Delta g_{NMDA}}{g_{NMDA}^*} - \frac{\Delta g_{GABA}}{g_{GABA}^*} \right) \tag{108}$$

$$\frac{\Delta\phi'_{I_{syn,I}}}{\phi'_{I_{syn,I}}} = U_I \left( \frac{I_X}{I_{GABA}} \frac{\Delta v_X}{v_X^*} + \frac{I_{AMPA}}{I_{GABA}} \frac{\Delta g_{AMPA}}{g_{AMPA}^*} + \frac{I_{NMDA}}{I_{GABA}} \frac{\Delta g_{NMDA}}{g_{NMDA}^*} - \frac{\Delta g_{GABA}}{g_{GABA}^*} \right), \tag{109}$$

where $\mathbf{U} = \begin{bmatrix} U_E & U_I \end{bmatrix}^T$ is given by

$$\mathbf{U} = -\tilde{a}\mathbf{W} + \widetilde{\mathbf{b}}_0. \tag{110}$$

Equations *Equations 108 and 109* describing the relative changes in the slopes of the response functions for excitatory and inhibitory neurons can now be combined with *Equations 93 and 94* to account for both direct and indirect dependence of the change in the oscillation growth rate $\Delta\lambda$ and change in the oscillation frequency $\Delta\omega$ on the small relative changes in the synaptic conductances and external rate.

## Numerical solutions

Self-consistent mean field equations for the eight conductance parameters, and linear stability equations for the perturbation growth rate $\lambda$ and the oscillation frequency $\omega$ were both solved numerically using custom codes written in MATLAB (The MathWorks) with the aid of *fsolve* function.

## Acknowledgements

We would like to provide our special thanks to Apostolos P Georgopoulos for his intellectual contributions to the study of the temporal correlations in networks and the temporal dynamics of neural ensembles that provided part of the motivation for this work, as well as for his constant support and encouragement. We thank Dean Evans for lab and project management as well as his assistance with surgeries, animal care, and neural recordings; Dale Boeff for his assistance with neurophysiological recording system design and construction, as well as computer programming for signal processing and data analysis; Sofia Sakellaridi for her assistance with neural recordings; Olivia Newman for preliminary analysis. This work was performed while RKB was employed at the University of Minnesota. The opinions expressed in this article are the author's own and do not reflect the views of the Food and Drug Administration, the Department of Health and Human Services, or the United States Government. Support for this work was provided by the Wilfred Wetzel Graduate Fellowship (to RKB), the National Institute of General Medical Sciences (T32 GM008244 and T32 HD007151 to RKB), and the National Institute of Mental Health (R25 MH101076 to RKB, and R01MH107491 and P50MH119569 to MVC). Partial funding for this study was provided by the University of Minnesota Foundation and the U.S. Department of Veterans Affairs. The sponsors had no role in the current study design, analysis or interpretation, or in the writing of this paper. The contents do not represent the views of the

U.S. Department of Veterans Affairs, the National Institutes of Health, the Department of Health and Human Services, or the United States Government.

## Additional information

### Funding

| Funder | Grant reference number | Author |
|---|---|---|
| National Institute of Mental Health | R01MH107491 | Matthew V Chafee |
| National Institute of Mental Health | P50MH119569 | Matthew V Chafee |
| National Institute of Mental Health | R25 MH101076 | Rachael K Blackman |
| National Institute of General Medical Sciences | T32 GM008244 | Rachael K Blackman |
| National Institute of General Medical Sciences | T32 HD007151 | Rachael K Blackman |
| University of Minnesota Foundation | | Bagrat Amirikian |
| U.S. Department of Veterans Affairs | | Bagrat Amirikian |

The funders had no role in study design, data collection and interpretation, or the decision to submit the work for publication.

### Author contributions

David A Crowe, Software, Formal analysis, Validation, Investigation, Writing – review and editing; Andrew Willow, Software; Rachael K Blackman, Validation, Investigation, Writing – review and editing; Adele L DeNicola, Investigation; Matthew V Chafee, Conceptualization, Funding acquisition, Validation, Investigation, Writing – original draft, Writing – review and editing; Bagrat Amirikian, Conceptualization, Software, Formal analysis, Supervision, Visualization, Methodology, Writing – original draft, Project administration, Writing – review and editing

### Author ORCIDs

Matthew V Chafee (iD) http://orcid.org/0000-0001-9289-0239
Bagrat Amirikian (iD) http://orcid.org/0000-0001-8080-0902

### Ethics

All animal care and experimental procedures conformed to National Institutes of Health (NIH) guidelines and were approved by the Institutional Animal Care and Use Committees of the Minneapolis Veterans Administration Medical Center (VA Animal Component of Research Protocol Number 170601).

### Decision letter and Author response

Decision letter https://doi.org/10.7554/eLife.79352.sa1
Author response https://doi.org/10.7554/eLife.79352.sa2

## Additional files

### Supplementary files

• MDAR checklist

### Data availability

The current manuscript is a computational study. Neural data analyzed in this paper are available at the Mendeley Data repository. Custom MATLAB codes used for simulations, mean field and stability

analysis are available at the GitHub: https://github.com/amirikian/SCZ-Synaptic-Circuit-Failure-Model (copy archived at *Amirikian, 2024*).

The following dataset was generated:

| Author(s) | Year | Dataset title | Dataset URL | Database and Identifier |
|-----------|------|---------------|-------------|-------------------------|
| Chafee MV | 2024 | Neural data accompanying Crowe et al. 'A prefrontal network operating near steady and oscillatory states links spike desynchronization and synaptic deficits in schizophrenia' | https://doi.org/10.17632/bt6j9gyz5t.1 | Mendeley Data, 10.17632/bt6j9gyz5t.1 |

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

## Appendix 1

## Accuracy of analytical approximation for the instability growth rate

*Appendix 1—figure 1* shows the comparison of the exact value of $\lambda$ obtained by numerically solving the mean field and stability analysis equations, with the analytical approximation given by *Equations 1, 2*. The comparison is performed separately for each of the three synaptic conductances and external rate by varying one of the underlying parameters while the remaining three keeping constant at their critical values. It is seen that in all cases, the analytical expression for $\lambda$ provides a very good approximation of the exact relationship.

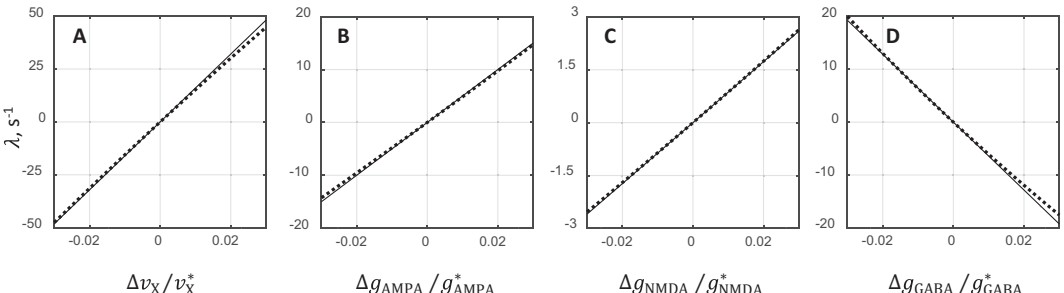

**Appendix 1—figure 1.** Comparison between exact solution and analytical approximation for the oscillatory instability growth rate $\lambda$. The comparison is performed separately for each of the three synaptic conductances and external rate by varying one of the underlying parameters while the remaining three keeping constant at their critical values. It is seen that in all cases, the analytical expression for $\lambda$ provides a very good approximation of the exact relationship. The plots show the rate $\lambda$ as a function of the relative deviation from the critical value of external rate (A), AMPAR conductance (B), NMDAR conductance (C), and GABAR conductance (D). The comparison is performed by varying the underlying parameter while keeping the other parameters at their critical values. The solid lines show analytical approximations for $\lambda$ given by *Equations 1, 2*, and the dotted lines are the exact solutions obtained from the mean field and stability analysis equations.

# Appendix 2

## Contributions from $\Lambda_{\text{AMPA}}$, $\Lambda_{\text{NMDA}}$, and $\Lambda_{\text{GABA}}$ terms to the instability growth rate

The reasons of why $\Lambda_{\text{NMDA}}, \Lambda_{\text{GABA}} \ll \Lambda_{\text{AMPA}}$ can be understood by considering analytical expressions of the factors $\Lambda_{\text{AMPA}}$, $\Lambda_{\text{NMDA}}$, and $\Lambda_{\text{GABA}}$. These quantities have a form $\Lambda_R = X_R \cos\left(\Phi_R + \Phi_0\right)/T_0$, where $X_R$ characterizes the relative attenuation in the strength of the corresponding synapse due to the $R$-current dynamics, $\Phi_R$ is the phase lag introduced by $R$-current synaptic filtering, and $T_0$ and $\Phi_0$ are constants (see Materials and methods). For illustration purposes, it is convenient to use a geometric interpretation of $\Lambda_R$ in which it is associated with a vector emanating from the origin of a cartesian plane, with the length $X_R/T_0$, and the angle $\Phi_R + \Phi_0$ between the vector and the $x$-axis. Then, the projection of this vector on the $x$-axis (i.e. its $x$-component) is $\Lambda_R$.

*Appendix 2—figure 1* shows vectors $L_{\text{AMPA}}$, $L_{\text{NMDA}}$, and $L_{\text{GABA}}$ corresponding to the values of $\Lambda_{\text{AMPA}}$, $\Lambda_{\text{NMDA}}$, and $\Lambda_{\text{GABA}}$ in the crttical network. It is seen that the length of the vector $L_{\text{NMDA}}$ is nearly zero and, therefore, its projection $\Lambda_{\text{NMDA}}$ is nearly zero as well. The length of $L_{\text{NMDA}}$ being very short means that the relative attenuation in the strength of NMDA synapse $X_{\text{NMDA}} \ll 1$. This is because the characteristic synaptic time constant $\tau_{\text{NMDA}} \sim 100$ ms of NMDA synapse is much longer than the period of the $\gamma$-band oscillation $1/f_{\text{ntwrk}} \sim 15$ ms emerging in the critical network at the boundary between the steady and oscillatory states. This results in a strong attenuation of the oscillatory component of NMDAR mediated current that is quantified by the $X_{\text{NMDA}}$ numeric value (*Equations 48, 53*).

The vector $L_{\text{GABA}}$, in contrast, has a length comparable with the length of $L_{\text{AMPA}}$. However, because its angle $\Phi_{\text{GABA}} + \Phi_0$ is close to 90°, the $L_{\text{GABA}}$ projection $\Lambda_{\text{GABA}}$ is much smaller than the projection $\Lambda_{\text{AMPA}}$ of the vector $L_{\text{AMPA}}$. This occurs because the phase shift $\Phi_{\text{GABA}}$ introduced by GABA-current synaptic filtering is lagging the phase shift $\Phi_{\text{AMPA}}$ introduced by AMPA-current filtering. This difference in the phase delays, in turn, is due to the fact that the GABA synapse ($\tau_{\text{GABA}} \sim 5$ ms) is slower than the AMPA synapse ($\tau_{\text{AMPA}} \sim 2$ ms). Thus, while both $\Lambda_{\text{NMDA}}$ and $\Lambda_{\text{GABA}}$ are much smaller than $\Lambda_{\text{AMPA}}$, the primary reasons are different.

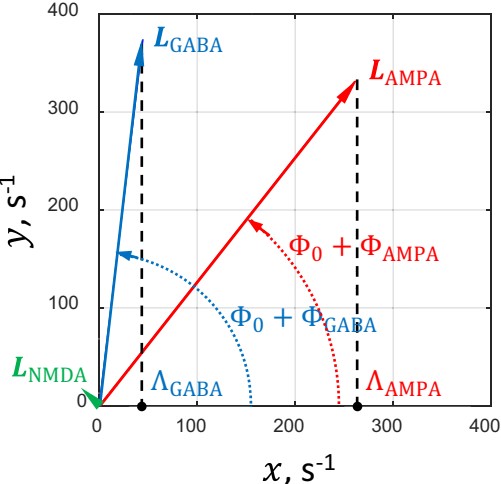

**Appendix 2—figure 1.** Geometric interpretation of the factors $\Lambda_{\text{AMPA}}$, $\Lambda_{\text{NMDA}}$, and $\Lambda_{\text{GABA}}$. For illustration purposes, the factors $\Lambda_{\text{AMPA}}$, $\Lambda_{\text{NMDA}}$, and $\Lambda_{\text{GABA}}$ (*Equation 1*) are associated with corresponding vectors $L_{\text{AMPA}}$, $L_{\text{NMDA}}$, and $L_{\text{GABA}}$ (see text for details). The projections of these vectors on the $x$-axis are equal to their respective factors. The vectors $L_{\text{AMPA}}$ and $L_{\text{GABA}}$ corresponding to the values of $\Lambda_{\text{AMPA}}$ and $\Lambda_{\text{GABA}}$ in the critical network are shown in red and blue, respectively. The green arrowhead at the origin relates to the vector $L_{\text{NMDA}}$ corresponding to $\Lambda_{\text{NMDA}}$. Due to the small size of the vector, it cannot be depicted in scale on this plot.

## Appendix 3

## Direct and indirect pathways for synaptic conductances and external rate affecting synchrony

*Equation 6* for the oscillatory instability growth rate $\lambda$ illuminates specific pathways by which modulations of synaptic conductances and external rate affect synchrony. In general, when fast excitatory feedback becomes sufficiently strong and is followed by slow and powerful inhibition the asynchronous state becomes unstable and synchronous oscillation emerges (*Brunel and Wang, 2003*; *Compte et al., 2000*; *Tsodyks et al., 1997*; *Wang, 1999*). The intuitive explanation is as follows. An increase of activity in the excitatory population caused by a fluctuation rapidly surges due to the fast excitatory feedback. The buildup of excess excitation, in turn, activates the inhibitory population. The growth of excitation continues until slow rising inhibition eventually suppresses excitatory population activity. As a result, activity of inhibitory population decays as well, and the next oscillation cycle starts. *Equation 5* explicitly shows that the strong excitatory feedback necessary to induce oscillation can emerge via two pathways. One is manifested in the indirect dependence of $\lambda$ on the synaptic parameters via their effect on the operating point of the current-frequency response function. Changes in the synaptic conductances and external rate producing an increase in the average total synaptic current ($\Delta I_{\text{syn}} > 0$) shift the operating point of excitatory population toward a steeper slope of the current-frequency response function ($\Delta \phi'_{I_{\text{syn}}} > 0$). As a result, the amplitude of the neuron's response to dynamically varying input increases, giving rise to stronger excitatory feedback. If AMPAR conductance is fixed, $\Delta I_{\text{syn}} > 0$ results in $\lambda > 0$ (*Equation 5*). Therefore, changes in the synaptic parameters that increase the average total synaptic currents, in general, have a destabilizing effect on the asynchronous activity and favor synchronous oscillations.

It should be noted that the change in the slope with a shift in the operating point is critical for the indirect mechanism functionality. If the response function were linear and, thus, the slope is constant, independent of $I_{\text{syn}}$ ($U_{\text{E}} = 0$, *Equation 5*), a change in the operating point $\Delta I_{\text{syn}}$ will still affect the neuron's average firing frequency. However, the neuron's response amplitude to dynamically varying input will be independent of the operating point, i.e., the average input current $I_{\text{syn}}$. As a result, shifts in the operating point will not change the strength of excitatory feedback. In the case of integrate-and-fire neurons, the input current-output frequency relationship is a monotonically increasing non-linear function ($U_{\text{E}} > 0$). Consequently, when the operating point shifts due to increasing average input current ($\Delta I_{\text{syn}} > 0$), the amplitude of the neuron's response to the same variation in the input current increases since the slope of the response function becomes steeper, giving rise to stronger excitatory feedback.

The second pathway is manifested in the direct dependence of $\lambda$ on the AMPAR conductance. An increase in the conductance results in stronger excitatory to excitatory connections and excitatory to inhibitory connections that, in turn, amplify fast AMPAR mediated excitatory feedback and slow GABAR mediated inhibition. If other parameters remain unchanged, $\Delta g_{\text{AMPA}} > 0$ also results in $\Delta I_{\text{syn}} > 0$ (*Equation 4*). This means that because of the AMPAR conductance increase, both direct and indirect pathways tend to make $\lambda > 0$ (*Equation 5*). Thus, increased fast excitation, due to the stronger AMPAR mediated current, and enhanced response to varying input, due to the shift in the operating point, destabilize the asynchronous dynamic and push the network toward synchronous oscillations.

## Appendix 4

### The mechanism by which NMDAR influences network synchronization

In this section we provide a more formal consideration of the mechanism for the transition between the steady and oscillatory states, and the lack thereof when the NMDAR is blocked. For convenience, we rewrite here *Equation 7* for the instability growth rate $\lambda$:

$$\frac{\lambda}{\Lambda_{\text{AMPA}}} = U_{\text{E}} \left( \frac{I_{\text{X}}}{I_{\text{GABA}}} \frac{\Delta v_{\text{X}}}{v_{\text{X}}^*} + \frac{I_{\text{NMDA}}}{I_{\text{GABA}}} \frac{\Delta g_{\text{NMDA}}}{g_{\text{NMDA}}^*} \right).$$

(111)

The expression in the brackets is proportional to the change $\Delta I_{\text{syn}}$ in the average total synaptic current (*Equation 4*). Without loss of generality, we assume that the drug-naive condition corresponds to the network for which $\Delta g_{\text{NMDA}} = 0$ and $\Delta v_{\text{X}} = -\delta < 0$. Thus, initially $\Delta I_{\text{syn}} < 0$ and the response function operating point is below the point of the critical network. As a result, $\lambda < 0$ and the network is in the asynchronous steady state. When external input increases by $2\delta$, so that $\Delta v_{\text{X}} = +\delta > 0$, the total synaptic current increases as well, so that $\Delta I_{\text{syn}}$ becomes positive and the operating point shifts above the point of the critical network. This results in $\lambda > 0$ and, therefore, the network transitions to the synchronous oscillatory state. This transition is due to the indirect dependence of $\lambda$ on the external rate change $\Delta v_{\text{X}}$ via its effect on the operating point of the current-frequency response function. In the drug condition, the NMDAR conductance is reduced and, if it becomes sufficiently small so that $\frac{\Delta g_{\text{NMDA}}}{g_{\text{NMDA}}^*} < -\frac{\delta}{v_{\text{X}}^*} \frac{I_{\text{X}}}{I_{\text{NMDA}}}$, the same increase in external rate will not result in positive $\lambda$. As a result, no oscillatory instability develops, and the network remains in the asynchronous steady state. This is because the decrease in the NMDAR conductance, which like $\Delta v_{\text{X}}$ influences $\lambda$ indirectly, reduces the average total synaptic current $I_{\text{syn}}$ and brings down the operating point to such locus that it cannot be shifted above the operating point of the critical network by the same $2\delta$ increase in the external rate.

When the oscillatory instability growth rate $\lambda$ on the left-hand side of *Equation 111* is fixed at zero, it becomes the equation for the critical hyperplane in the synaptic parameter space separating the asynchronous and synchronous states. In the context of our model, we are interested in the network states within the $\left( v_{\text{X}}/v_{\text{X}}^*, g_{\text{NMDA}}/g_{\text{NMDA}}^* \right)$ subspace and, thus, *Equation 111* becomes the equation for the line

$$0 = I_{\text{X}} \frac{\Delta v_{\text{X}}}{v_{\text{X}}^*} + I_{\text{NMDA}} \frac{\Delta g_{\text{NMDA}}}{g_{\text{NMDA}}^*}.$$

(112)

This critical line is the analytical approximation of the exact critical line shown in the state diagram depicted in *Figure 6C* that was obtained by solving the mean field and stability analysis equations numerically. *Appendix 4—figure 1* shows the comparison between the exact critical line (dotted line) and its approximation given by *Equation 112* (solid line). It is seen that these lines virtually overlap and, thus, the accuracy of approximation is very good.

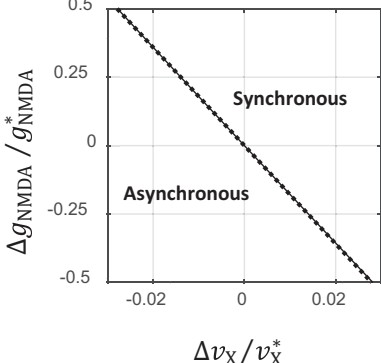

**Appendix 4—figure 1.** Comparison between exact and analytical approximation of the critical line separating asynchronous and synchronous states. The solid line shows the analytical approximation of the critical line given

by *Equation 112*. The dotted line corresponds to the exact solution obtained from the mean field and stability analysis equations. On the scale of this plot, the two curves virtually overlap.

# Appendix 5

## Special case of the zero net effect of the variation of synaptic parameters on the average total synaptic current

Here we consider an interesting case (see a specific example below) when variations in the synaptic conductances and external rate are such that they have no net effect on the average total synaptic current because the sum of their contributions cancels out. In this case, $\Delta I_{\text{syn}} = 0$ and there will be no shifts in the operating points and, therefore, no change in the firing rates and slopes of the response functions (*Equation 3*). As a result, despite the synaptic parameter changes there will not be any indirect contribution to $\lambda$ (*Equation 1*). However, even though the background firing rates remain unchanged, the oscillatory instability growth rate $\lambda$ will be affected via its direct dependence on the changes in the synaptic conductances.

Such case was studied in *Compte et al., 2000*. Specifically, Compte and colleagues observed in simulations that the concurrent increase in the AMPAR and decrease in the NMDAR conductances, so that the total strength of excitation remains fixed, leads to the destabilization of the asynchronous steady state and emergence of synchronous oscillations. Assuming that the network is close to the boundary between the asynchronous and synchronous states, this phenomenon can be explained in terms of *Equation 6* for the oscillatory instability growth rate $\lambda$. In this case, it simplifies to

$$\frac{\lambda}{\Lambda_{\text{AMPA}}} = \frac{\Delta g_{\text{AMPA}}}{g^*_{\text{AMPA}}} + U_{\text{E}} \left( \frac{I_{\text{AMPA}}}{I_{\text{GABA}}} \frac{\Delta g_{\text{AMPA}}}{g^*_{\text{AMPA}}} + \frac{I_{\text{NMDA}}}{I_{\text{GABA}}} \frac{\Delta g_{\text{NMDA}}}{g^*_{\text{NMDA}}} \right) . \tag{113}$$

The expression in the brackets is proportional to the change $\Delta I_{\text{syn}}$ in the average total synaptic current (*Equation 4*). It can be seen that decreasing the NMDAR conductance by $\Delta g^0_{\text{NMDA}} < 0$ and simultaneously increasing the AMPAR conductance in the amount of $\Delta g^0_{\text{AMPA}} = -\Delta g^0_{\text{NMDA}} \frac{I_{\text{NMDA}}}{I_{\text{AMPA}}} \frac{g^*_{\text{AMPA}}}{g^*_{\text{NMDA}}} > 0$ results in zero for the expression in the brackets. This means that the contributions from the changes in the NMDAR and AMPAR mediated currents to the average total synaptic current $I_{\text{syn}}$ cancel each other and $\Delta I_{\text{syn}} = 0$. Thus, in this case, although the AMPAR and NMDAR conductances change, the operating point of the response function does not. However, because $\lambda$ also depends directly on the AMPAR conductance (the first term in the r.h.s. of *Equation 113*) and $\Delta g^0_{\text{AMPA}} > 0$, the instability growth rate $\lambda$ becomes positive and synchronous oscillations emerge due to the stronger excitatory-inhibitory feedback loop.

