## [Editor Report]

This valuable modeling study proposes a local circuit mechanism based on a network of recurrently connected excitatory and inhibitory neurons for the recently reported effect that NMDA receptor antagonists cause a drastic reduction of prefrontal neural synchronization in preparation for motor responses in a cognitive task. This mechanism is convincingly supported by simulations of spiking networks and a thorough analysis of the parameter dependency of network dynamics using mean-field theory. The work will be of general interest to computational neuroscientists, and especially for those interested in computational psychiatry.

---

## [Decision Letter]

**Decision letter after peer review:**

Thank you for submitting your article "A prefrontal network model operating near steady and oscillatory states links spike desynchronization and synaptic deficits in schizophrenia" for consideration by *eLife*. Your article has been reviewed by 3 peer reviewers, including Albert Compte as the Reviewing Editor and Reviewer #1, and the evaluation has been overseen by John Huguenard as the Senior Editor. The following individual involved in the review of your submission has agreed to reveal their identity: Yashar Ahmadian (Reviewer #2).

The manuscript is currently not suitable for publication in *eLife*, but we are willing to consider a revised version that would fully address all concerns of the reviewers. The reviewers have discussed their reviews with one another, and the Reviewing Editor has drafted this to help you prepare a revised submission.

Essential revisions:

The reviewers consider that the manuscript has potential but it requires extensive revisions and new simulations to support its claims. Because these are substantial revisions, a revised manuscript will be critically reviewed in depth again and sent back out to the original reviewers.

The interest in the computational results depends critically on the ability of the model to provide a convincing explanation of the experimental data (Zick et al. 2018, 2021). For this, it is essential to:

1) consider not only the dynamics of synchronization (Figure 1) but also the dynamics of firing rate, and not only the period immediately preceding motor action but also the dynamics of the effects following motor action and around the stimulus cue. The mechanisms of the computational model make specific predictions as to how the dynamics of firing rate and oscillatory synchronization will be associated, and this should be specifically validated against the experimental data.

2) consider how the model accommodates a U-shape relationship between NMDAR modulation and network synchrony. Existing evidence shows that 0-lag synchrony in prefrontal networks is affected by manipulations that reduce NMDAR function (Zick et al. 2018) and by manipulations that enhance NMDAR function (Zick et al. 2021). The computational model presented in this manuscript does not show this U-shape behavior and the discussion does not mention this.

In addition, the theoretical understanding of the model should be enhanced by:

3) expanding the theoretical analysis to provide more intuition on the way NMDA works the way it does, and clarification of its distinct role in synchronization (in contrast to AMPA).

4) investigating how synchrony in the model reacts to transient external inputs and how the model can produce weakly oscillatory synchrony.

Finally, all reviewers were concerned about the emphasis on the association with schizophrenia, so we recommend:

5) moving speculative links to schizophrenia to the Discussion section. Also, please broaden the scope of this discussion with available literature on how synchrony and STDP interact in computational models, and with alternative explanations based on changes in NMDAR-dependent synaptic plasticity irrespective of synchronization.

*Reviewer #1 (Recommendations for the authors):*

Here are some specific recommendations for improvement of the manuscript:

1) The last two paragraphs in the introduction are largely discussional in spirit. The paragraph before last does not make much sense here, at a point where the reader is not yet familiar with the details of the model proposed. The very last paragraph is also mostly speculative in how the results in this manuscript would be important for schizophrenia and would appear more natural in the Discussion section.

2) The data presented in Figure 1 could be presented more fully in order to provide more constraints for the model so as to enhance its biological plausibility. For instance, it would be most interesting to have a parallel plot of the firing rate in these same neurons to see how neuronal activity evolves through the task. Neurons show a response to the cue stimulus (Zick et al. 2018), which may be similar to the response at the time of the response, and that is something that the model would have to explain: how does the control network react with strong synchronization when it receives inputs prior to the response but instead does not show any synchronization when it receives inputs at the time of the cue? Also, it would be good to show panel b extended for a time window after motor response time t=0 to compare the time course of firing rate and synchrony increases. This could also constrain further the network model.

3) In Figure 1 it would be good to have a dashed horizontal line at y=0 to separate correlated (y>0) from anti-correlated (y<0) activity more clearly. I would also recommend changing the x-axis label ("time lag") because this is now used in cross-correlation functions in Figure 4 and they are intrinsically different. The synchrony plot should also include error bars and proper inference about differences between the two conditions at different time points. For clarity, it would be nice to indicate with a shaded area or horizontal bar the period of cue stimulus presentation, too.

4) Briefly mention in line 167 what spiking neuron model you are using

5) Mention of "GABA receptors" in line 173 suggests that the model includes both GABA_A and GABA_B receptors.

6) Verb missing in "and mediated" in line 174.

7) In Figure 3 the color code red/blue means two different things: excitatory vs. inhibitory neurons, and steady vs. critical networks. This is confusing and should be clarified with different color codes for each thing.

*Reviewer #2 (Recommendations for the authors):*

1) I assume elucidation of the precise mechanism by which external inputs or NMDA affect synchrony can be achieved by analytically inspecting the direct and indirect dependence of the real part of the relevant complex eigenvalue on the synaptic conductances and external inputs, either directly or indirectly via the dependence of linearized neural gains in the background operating point (which itself depends on the external input and the synaptic parameters).

2) The authors do refer to the Compte et al. 2000 study that found that when the total strength of excitatory connectivity is fixed, tilting the relative strength of NMDAR and AMPAR towards the former disfavors synchronous oscillations (with relatively high frequency in or near the γ band). Previous studies (e.g. Wang and Brunel 2003, Tsodyks et al. 1997, and Compte et al. 2000) have pointed out the role of the negative feedback loop between excitatory and inhibitory populations mediated by the fast receptors AMPA and GABA in γ-band oscillations. Again I think the mean-field theory and the stability analysis based on it could be used to elucidate the differences and similarities in the pathways by which NMDAR and AMPAR affect the transition to synchrony. For example, the role of NMDAR could be via the explicit dependence of the growth rate of oscillations (the eigenvalue real part) on g_NMDA, or (most likely) via the effect of NMDAR-based excitation on background rates (which indirectly affect the eigenvalue).

3) Regarding the link to "disconnection" via STDP. Various studies have found that STDP in (initially randomly connected) networks in the asynchronous state can maintain random connectivity (Morrison et al. 2007 – DOI: 10.1162/neco.2007.19.6.1437) or even lead to the emergence of structured connectivity (Izhikevich et al. 2004, Babadi and Abbott 2013, and Litwin-Kumar and Doiron 2014 -- DOIs: 10.1093/cercor/bhh053, 10.1371/journal.pcbi.1002906, 10.1038/ncomms6319). In lieu of evidence (either previous studies or new simulations/modeling by the authors) for this claim, I suggest weakening the claim to saying that the NMDAR-related reduction in synchrony, as a specific disruption in the patterns of precise spike timing, can lead to abnormal changes in patterns of synaptic connectivity via STDP. I also suggest citing the above-mentioned (or similar) studies on previous studies of STDP in asynchronous (or synchronous) states, or any other studies that support specific claims by the authors in this regard.

Other suggestions:

– Figure 6 top: here the cross-correlation plot based on monkey data (as in the model) exhibits nonzero lag "γ" peaks. But these seem to be absent in the data shown in the Neuron paper (e.g. Figure 4D therein). Can the authors comment on this seeming discrepancy? Is this because data only from one of the monkeys (showing stronger effects) was included in the current paper? Can the authors also comment on the flatness of the CCH in the "initial probe period" in the monkey data in contrast to the small 0-lag peak in the model CCH?

– Lines 104-105: I found the meaning of *balance between* in this sentence vague/unclear. What does the balance between NMDA inputs and oscillatory activity mean?

– I suggest the authors also mention (e.g. around lines 476-477 where they discuss their assumption 1, and as further support for this assumption) that in their empirical data (Neuron paper) they do find an increase in PFC neural firing rates in the pre-response period, which will presumably also arise in their model from the increase in external inputs.

– Line 781: "themes" → "times".

– Line 838: I would explain where this equation comes from (I understand it accounts for the effect of reset on mean membrane potential), and also what v_α denotes.

*Reviewer #3 (Recommendations for the authors):*

Here are issues that I believe the authors should address before the paper can be published:

1. In Figure 1, the authors only show the synchrony before the motor response. I believe it would be instructive to show also what happens after motor response, to visualize how temporally precise is this increase. It would also be useful to show the time course of average firing rates. The explanation of the authors relies on an increase in external input to generate an increase in synchrony – this increase in external input should be reflected in an increase in the average firing rate as well.

2. Figure 6 shows a strong difference between the network model and data – side peaks in spike correlations are much more prominent in the model than in the data. Thus, synchronization in the data is much less oscillatory than in the model. Is there a way in the model to get such weakly oscillatory synchrony? If not, can the authors speculate as to what additional mechanisms would lead to a reduction of γ power, preserving zero lag synchrony?

3. In the data, the increase in synchrony is transient. In the simulations, the author considers the case of stationary external inputs. I believe the authors should also investigate the case of transient increases of external inputs leading to a more realistic time course of synchrony. Such a scenario could also help in solving

the issue mentioned in 2.

[Editors' note: further revisions were suggested prior to acceptance, as described below.]

Thank you for resubmitting your work entitled "A prefrontal network model operating near steady and oscillatory states links spike desynchronization and synaptic deficits in schizophrenia" for further consideration by *eLife*. Your revised article has been evaluated by John Huguenard (Senior Editor) and a Reviewing Editor.

The manuscript has been improved but there are some remaining issues that need to be addressed, as outlined below:

1. The dynamics of average firing rates in Figure 1 should mark clearly the various relevant periods of the task on the time axis: probe interval, motor response, etc.

2. The new theoretical analyses in this revision have greatly extended the length of the main text, but the additional insight provided is not clearly explained and in its current form it can lead to confusion. It is suggested that most of this material is moved to a supplementary document and a more concise and clearly delineated section in the main text provides the fundamental message of how NMDARs influence network synchronization.

*Reviewer #3 (Recommendations for the authors):*

The paper has been considerably improved and I would like to congratulate the authors for their hard work. There are still a couple of issues though that I think should be addressed to improve the paper further.

1. In response to one of the main concerns of the reviewers, the authors have expanded Figure 1 to include the dynamics of average firing rates in the data and the effect of NMDAR blockade on this dynamic. This is a welcome addition, but I think this figure could be improved to better understand the time course of the task. Currently, there is no discussion at all about when the motor response is happening and so the left and right panels seem totally disconnected. One has to go back to the original paper by Zick et al. to understand when the motor response happens. I would suggest extending the left panels to include the whole trial (including the `probe' interval), adding where the mean response time is similar to Figure 4 in the Zick et al. 2018 paper. Or at the very least indicate when is the mean response time (time 0 in right panels) with respect to the time axis of the left panels.

2. In response to another concern, the authors have considerably expanded their theoretical analysis with a detailed description of an approximation for the instability growth rate. However, while the approximation itself represents an interesting new derivation, I did not find that it brings much insight, and in fact in some ways introduces additional confusion, for the following reasons. (i) The main new result of this new analysis is the new Equation (1). In this equation, the LambdaRs play critical roles and are the subject of much of the later discussion, however, their meaning is not really explained in the main text and one has to go to the Methods to understand what they really are. (ii) The new Figure (10) indicates that the growth rate is primarily determined by the AMPA term and that the GABA term plays little role. This is quite confusing as oscillations in such networks relies on GABAR mediated negative feedback. How can the authors explain this fact? (iii) After reading the new part, I ended up not having much more insight about mechanisms. My guess at this point is that NMDAR influences the synchronisation properties of the network by providing an additional excitatory drive, thereby playing a similar role as the external excitatory input. I have the feeling that is what the authors are trying to say when they say that the NMDAR term acts through changes in phiprime (the slope of the transfer function), which is exactly how the external inputs act, but it feel like it could be explained in a much clearer way.

---

## [Author Response]

Essential revisions:The reviewers consider that the manuscript has potential but it requires extensive revisions and new simulations to support its claims. Because these are substantial revisions, a revised manuscript will be critically reviewed in depth again and sent back out to the original reviewers.The interest in the computational results depends critically on the ability of the model to provide a convincing explanation of the experimental data (Zick et al. 2018, 2021). For this, it is essential to:1) consider not only the dynamics of synchronization (Figure 1) but also the dynamics of firing rate, and not only the period immediately preceding motor action but also the dynamics of the effects following motor action and around the stimulus cue. The mechanisms of the computational model make specific predictions as to how the dynamics of firing rate and oscillatory synchronization will be associated, and this should be specifically validated against the experimental data.

We have extended our analysis of both the simulated and recorded neural data to include modulation of both synchrony and firing rate during the DPX task (in the neural data) and in response to transient inputs to the PFC network (in the simulated data), both in the baseline case, and following reduction of NMDAR synaptic transmission. We added panels C and D to Figure 1 to illustrate modulation of firing rate in the neural data and the influence of NMDAR blockade on neural firing rates. This new analysis shows (1) firing rate in monkey prefrontal cortex ramped gradually up during the delay period considerably before the motor response, (2) spike synchrony increased more abruptly and dramatically immediately before the motor response, and (3) blocking NMDAR attenuated both the increase in spike synchrony and the increase in firing rate (new Figure 1C, D). In addition, we have extended the time of analysis of both synchrony and firing rate to encompass the response as well as the time period following the response. To facilitate the comparison of the dynamics of firing rate and synchrony between experimental data and the computational model we modified the location of the critical network in the parameter space by changing the balance (ratio) between NMDAR and GABAR mediated currents from 0.2 to 0.15. We, correspondingly, repeated all previous simulations for the new value of the underlying parameter and updated the manuscript figures accordingly. These new figures are very similar to the previous ones. Next, we extended our consideration to transient external inputs to the spiking network to emulate time-varying inputs to prefrontal networks that were likely to drive task-locked changes in firing rates (Figure 1C, D), as suggested by Reviewers (see Point 4, below), and provide the results of the simulations in the new Figure 7. Figure 7 demonstrates that modulation of firing rate and spike synchrony in the network simulation exhibit several features that were evident in the recorded neural data. Namely, increases in spike rate were early and gradual following transient increase in network inputs, whereas increases in spike synchrony were delayed and abrupt in relation to transient increase in network inputs in the network simulation (Figure 7B, C; black), as we saw in the neural data (Figure 1C, D; black). In addition, the increase in both firing rate and spike synchrony were attenuated by reducing NMDAR synaptic currents in the model (Figure 7B, C; magenta) as they were in the neural data (Figure 1C, D; magenta). Thus, there was considerable agreement between the simulated and recorded neural data, although the neural data additionally exhibited several unique features, including additional dynamics after the response (Figure 1C, D) that likely reflected complex modulation of synaptic inputs during this time. The new simulation results characterizing the network response to transient inputs both in terms of spike synchrony and spike rate are provided in the Results section titled ‘Explaining the effects of blocking NMDAR observed in primate PFC by the prefrontal circuit model’, starting at Line 367.

2) consider how the model accommodates a U-shape relationship between NMDAR modulation and network synchrony. Existing evidence shows that 0-lag synchrony in prefrontal networks is affected by manipulations that reduce NMDAR function (Zick et al. 2018) and by manipulations that enhance NMDAR function (Zick et al. 2021). The computational model presented in this manuscript does not show this U-shape behavior and the discussion does not mention this.

We are grateful to the Reviewers for raising this possibility, which we consider in a new section we have added to the revised Discussion entitled ‘Potential U-shaped relation between NMDAR function and spike synchrony’ (starting at Line 777). There we note the possibility that either too little or too much NMDAR synaptic activity could impair spike synchrony, in a manner analogous to the U-shaped relationship between D1 receptor activation and prefrontal persistent activity as reported by Vijayraghavan, Arnsten and colleagues (2007), for example. However, existing neurophysiological data do not establish yet that the relationship between spike synchrony and NMDAR function has an inverted U-shape. Specifically, we did not test the dose-response relationship between spike synchrony and NMDAR modifying drugs in the primate drug model, nor establish that NMDAR upregulation was responsible for reduced spike synchrony in the mouse genetic model (Deleting *Dgcr8* would alter expression of many downstream proteins). The absence of a U-shaped relationship between NDMAR function and spike synchrony in the model from this perspective does not seem to represent a disagreement with available biological data.

In addition, the theoretical understanding of the model should be enhanced by:3) expanding the theoretical analysis to provide more intuition on the way NMDA works the way it does, and clarification of its distinct role in synchronization (in contrast to AMPA).

We have now extended the theoretical consideration and obtained an analytic approximation for the oscillatory instability growth rate λ describing the dependence on the AMAPR, NMDAR, GABAR synaptic conductances and external rate. Based on this consideration, we have now provided a substantially more detailed theoretical account of the precise mechanism implemented in our model for the transition between the steady and oscillatory states and the lack thereof when the NMDAR conductance is blocked. Also, the closed form analytic expression for λ allowed us to reveal the differences and similarities in how AMPAR and NMDAR, both of which mediate recurrent excitation, influence the stability of asynchronous state and transition to synchronous oscillations. The results of this new theoretical analysis are presented in the new section “Dependence of oscillatory instability growth rate on synaptic parameters” in the Results and include four new figures (Figures 9-12). The mathematical details are given in the new section “Analytical consideration of the dependence of oscillation growth rate on network parameters” in the Methods.

4) investigating how synchrony in the model reacts to transient external inputs and how the model can produce weakly oscillatory synchrony.

In the revised manuscript we extended our consideration to transient external inputs. To this end, we carried out a new series of simulations in which stationary external input received by network neurons was replaced with a time-varying input that had a trapezoid-like temporal profile. Results of these new simulations are summarized in the new Figure 7 mentioned above. Also, as it was suggested by Reviewer #3, we found that the oscillatory synchrony became slightly weaker, which could be appreciated from relatively smaller off-zero lag peaks in the population average spike correlation function as compared with the case in which external input was stationary. More generally, as one can appreciate from Figure 4 panels A2 and C2 and Figure 5B the strength of oscillatory synchrony depends on how far from the oscillatory instability boundary the synchronous network state is: the closer it is, the weaker oscillatory synchrony is, as it can be appreciated by decreasing off-zero lag peaks while the 0-lag peak remains present.

Finally, all reviewers were concerned about the emphasis on the association with schizophrenia, so we recommend:5) moving speculative links to schizophrenia to the Discussion section. Also, please broaden the scope of this discussion with available literature on how synchrony and STDP interact in computational models, and with alternative explanations based on changes in NMDAR-dependent synaptic plasticity irrespective of synchronization.

These are good points and we thank the reviewers for raising them. We have made the suggested changes by limiting the link between the current study and schizophrenia in the Introduction to the motivation for the original neurophysiological experiments (as this link dictated the pharmacological and genetic manipulations we employed in the animal models). We have also added a new section to the Discussion with the heading ‘Spike timing disruptions and rewiring of prefrontal local circuits via STDP’ where we discuss the complexity of the interaction between STDP, synchrony, and connectivity in prior modeling studies. Namely, it is difficult to predict whether loss of synchronous spiking would cause disconnection via STDP without additional data. We acknowledge this constraint on our original hypothesis that asynchrony would cause disconnection considering these prior theoretical studies in this new section.

Reviewer #1 (Recommendations for the authors):Here are some specific recommendations for improvement of the manuscript:1) The last two paragraphs in the introduction are largely discussional in spirit. The paragraph before last does not make much sense here, at a point where the reader is not yet familiar with the details of the model proposed. The very last paragraph is also mostly speculative in how the results in this manuscript would be important for schizophrenia and would appear more natural in the Discussion section.

These are good points and we thank the reviewers for raising them. We have made the suggested changes by limiting the link between the current study and schizophrenia in the Introduction to the motivation for the original neurophysiological experiments (as this link dictated the pharmacological and genetic manipulations we employed in the animal models). Specifically, we have removed speculative links between the model results we present and synaptic disconnection in schizophrenia. In addition, we have removed the second to last paragraph in the original submission describing the relationship to prior modeling studies. Detailed comparison to prior work is now restricted to the Discussion as recommended.

2) The data presented in Figure 1 could be presented more fully in order to provide more constraints for the model so as to enhance its biological plausibility. For instance, it would be most interesting to have a parallel plot of the firing rate in these same neurons to see how neuronal activity evolves through the task. Neurons show a response to the cue stimulus (Zick et al. 2018), which may be similar to the response at the time of the response, and that is something that the model would have to explain: how does the control network react with strong synchronization when it receives inputs prior to the response but instead does not show any synchronization when it receives inputs at the time of the cue? Also, it would be good to show panel b extended for a time window after motor response time t=0 to compare the time course of firing rate and synchrony increases. This could also constrain further the network model.

We completely agree that the neural data used in Figure 1 should be presented more fully.

1) In the revised Figure 1, we have now added parallel plots displaying the time course of population average firing rate observed in neural data under the two conditions. We have also added a new figure, Figure 7, showing how population spike rate and synchrony obtained from spike trains of simulated networks in response to transient inputs evolve in time and compare the results of simulations with neural data presented in the revised Figure 1.

2) “how does the control network react with strong synchronization when it receives inputs prior to the response but instead does not show any synchronization when it receives inputs at the time of the cue?” This is an important point, that we address now in a new Figure 6—figure supplement 2. First, it can be now appreciated from Figure 1C and D that the population firing rate response to the cue and initial probe periods is weaker than the response at the time of the motor response. The likely reason is that during the cue/probe periods, as it is commonly assumed in modeling studies, only a small group of neurons (typically <10% in network models) receive the external/selective inputs. Second, to understand whether our model is consistent with the experimental observation that during the cue and initial probe periods no increase in the spike synchrony occurs, we carried out a series of simulations in which the fraction of neurons that received the increase in external spike rate varied from 0 to 1. The plot in Figure 6—figure supplement 2 shows spike synchrony obtained from the spike trains of simulated networks as a function of the fraction f. In this approach, the cue and probe presentations correspond to the case when relatively small fraction of neurons (f<0.2) receive the increase in external input, in contrast to the case of the pre-response period, when in the framework of our model all neurons receive increased input (f=1). It is seen that for small f the spike synchrony remains week, which is in agreement with the experimental observations.

3) We have now extended the panel B in Figure 1, as well as the new panel D displaying the time evolution of population firing rate, beyond the motor response time *t*=0 allowing the comparison of the time courses of synchrony and firing rate observed in neural data. In the revised Results section “Explaining the effects of blocking of NMDAR observed in primate PFC by the prefrontal circuit model” we also carry out a similar comparison for data obtained from simulated networks presented in new Figure 7 mentioned above. In addition, we compare the simulated data with neural data presented in the revised Figure 1.

3) In Figure 1 it would be good to have a dashed horizontal line at y=0 to separate correlated (y>0) from anti-correlated (y<0) activity more clearly. I would also recommend changing the x-axis label ("time lag") because this is now used in cross-correlation functions in Figure 4 and they are intrinsically different. The synchrony plot should also include error bars and proper inference about differences between the two conditions at different time points. For clarity, it would be nice to indicate with a shaded area or horizontal bar the period of cue stimulus presentation, too.

We thank the reviewer for the valuable suggestions.

1) In the revised Figure 1 we have now added a dashed horizontal line at y=0.

2) Thank you for noticing the mislabeled axis: we have now changed the “time lag” label to “time”.

3) We have now included the standard errors depicted as shaded bands in the synchrony plots.

4) We have now marked the time points in Figure 1 when the two conditions are statistically different by asterisks and provide the details in the figure legend.

5) We have now indicated by a yellow shaded area the period of cue stimulus presentation.

4) Briefly mention in line 167 what spiking neuron model you are using

We have added a statement saying that we are using leaky integrate-and-fire neuron model (line 124).

5) Mention of "GABA receptors" in line 173 suggests that the model includes both GABA_A and GABA_B receptors.

We have appended the subscript A to GABA to remove the uncertainty (line 130).

6) Verb missing in "and mediated" in line 174.

We have inserted the missing verb (line 131).

7) In Figure 3 the color code red/blue means two different things: excitatory vs. inhibitory neurons, and steady vs. critical networks. This is confusing and should be clarified with different color codes for each thing.

We agree that the choice of the colors is confusing. We have now changed the color code for the firing rates of excitatory/inhibitory neurons in Figure 3 from red/blue to black/green, respectively.

Reviewer #2 (Recommendations for the authors):1) I assume elucidation of the precise mechanism by which external inputs or NMDA affect synchrony can be achieved by analytically inspecting the direct and indirect dependence of the real part of the relevant complex eigenvalue on the synaptic conductances and external inputs, either directly or indirectly via the dependence of linearized neural gains in the background operating point (which itself depends on the external input and the synaptic parameters).

Thank you for this valuable suggestion. We have now extended the theoretical consideration and obtained an analytic approximation for the oscillatory instability growth rate λ describing the dependence on the AMAPR, NMDAR, GABAR synaptic conductances and external rate. We have accomplished this by linearizing the mean field and linear stability analysis equations near the boundary between the asynchronous and synchronous states where λ=0 . We have shown that λ essentially directly depends only on the AMPAR conductance. However, λ depends on the NMDAR, GABAR as well as AMPAR conductances and external rate indirectly via their effect on the operating point of the neuron’s input current-output frequency response function. The results of this new theoretical analysis are presented in the new section “Dependence of oscillatory instability growth rate on synaptic parameters” in the Results and include three new figures. The mathematical details are given in the new section “Analytical consideration of the dependence of oscillation growth rate on network parameters” in the Methods.

In the new Results section, we have now provided a more detailed theoretical account of the precise mechanism implemented in our model for the transition between the steady and oscillatory states and the lack thereof when the NMDAR conductance is blocked. We do this in terms of the equation for the instability growth rate λ describing the dependence on the NMDAR conductance and external rate in closed form. We also show that the critical line separating the asynchronous and synchronous states in the synaptic parameter space predicted by the analytical approximation is in excellent agreement with the exact line obtained by solving the mean-filed and linear stability analysis equations numerically. We believe that these and other related results described in the new Results section considerably improved the manuscript, and we are very grateful to the reviewer for giving us the opportunity to provide a deeper analytical insight into the involved synaptic mechanisms.

2) The authors do refer to the Compte et al. 2000 study that found that when the total strength of excitatory connectivity is fixed, tilting the relative strength of NMDAR and AMPAR towards the former disfavors synchronous oscillations (with relatively high frequency in or near the γ band). Previous studies (e.g. Wang and Brunel 2003, Tsodyks et al. 1997, and Compte et al. 2000) have pointed out the role of the negative feedback loop between excitatory and inhibitory populations mediated by the fast receptors AMPA and GABA in γ-band oscillations. Again I think the mean-field theory and the stability analysis based on it could be used to elucidate the differences and similarities in the pathways by which NMDAR and AMPAR affect the transition to synchrony. For example, the role of NMDAR could be via the explicit dependence of the growth rate of oscillations (the eigenvalue real part) on g_NMDA, or (most likely) via the effect of NMDAR-based excitation on background rates (which indirectly affect the eigenvalue).

As implied by the reviewer, the analytic expression for the oscillation growth rate λ allowed us to reveal the differences and similarities in how AMPAR and NMDAR, both of which mediate recurrent excitation, influence the stability of the asynchronous state and the transition to synchronous oscillations. Both AMPAR and NMDAR conductances affect λ indirectly by influencing the amplitude of the neuron’s response to varying input current. However, AMPAR conductance also affects λ directly by influencing the strength of fast excitatory feedback. We have now added the corresponding text in the Discussion section.

In the new Results section mentioned in point (1), we explain that the direct dependence manifests the essential influence of the AMPAR synaptic conductance on the strength of excitatory-inhibitory feedback loops via fast excitatory to excitatory and excitatory to inhibitory recurrent connections. The indirect dependence manifests the influence of the synaptic parameters, including NMDAR and external rate, on the location of the operating point on the neuron’s current-frequency response curve and, therefore, the slope of the response function. The steepness of the slope, in turn, determines the amplitude of the neuron’s response to dynamically varying input current and, therefore, affects the strength of the feedback loop.

Also, in the new Results section, we have provided a theoretical account of the simulation results observed in Compte et al., 2000, when concurrent increase in the AMPAR and decrease in the NMDAR conductances leads to synchronous oscillations. As in the case of our model, the explanation is given in terms of the equation for the instability growth rate λ that describes the direct (AMPAR) and indirect (both AMPAR and NMDAR) synaptic conductance contributions to the emergence of oscillations. In addition, we have also revised the Discussion section where we address the apparent contradiction between Compte et al., 2000, simulations and Zick et al., 2018, experiments, emphasizing the critical role played by the direct and indirect mechanisms by which NMDAR and AMPAR affect the transition from the asynchronous to the synchronous dynamic.

3) Regarding the link to "disconnection" via STDP. Various studies have found that STDP in (initially randomly connected) networks in the asynchronous state can maintain random connectivity (Morrison et al. 2007 – DOI: 10.1162/neco.2007.19.6.1437) or even lead to the emergence of structured connectivity (Izhikevich et al. 2004, Babadi and Abbott 2013, and Litwin-Kumar and Doiron 2014 -- DOIs: 10.1093/cercor/bhh053, 10.1371/journal.pcbi.1002906, 10.1038/ncomms6319). In lieu of evidence (either previous studies or new simulations/modeling by the authors) for this claim, I suggest weakening the claim to saying that the NMDAR-related reduction in synchrony, as a specific disruption in the patterns of precise spike timing, can lead to abnormal changes in patterns of synaptic connectivity via STDP. I also suggest citing the above-mentioned (or similar) studies on previous studies of STDP in asynchronous (or synchronous) states, or any other studies that support specific claims by the authors in this regard.

We thank the reviewer for raising this important point. It motivated us to reexamine our initial hypothesis that asynchrony at the level of spiking neurons would necessarily be expected to disconnect recurrent networks considering prior theoretical studies examining STDP. We consider constraints on our hypothesis imposed by the prior theoretical studies in a new section we have added to the Discussion (‘Spike timing disruptions and rewiring of prefrontal local circuits via STDP’, see new text starting on line 739). There we acknowledge that the relationship between spike timing dynamics and synaptic connectivity in networks incorporating STDP is hard to predict because changes in spike synchrony and synaptic connectivity influence one another in complex ways after STDP is implemented and spiking patterns are allowed to sculpt synaptic connectivity. We thank the reviewer for pointing out this caveat and we now acknowledge it as a constraint on our prior hypothesis in the Discussion. However, to our understanding, prior network stimulations incorporating STDP have obtained divergent results as to how synchrony and connectivity influence one another that appear to depend strongly on the initial assumptions of the models. These crucial factors appear to include delays in signal conduction imposed by axons, synapses, and dendrites, and how network simulations model these delays. For example, delays in signal conduction imposed by the length of axons in recurrent connections between pyramidal neurons can be on the order of tens of ms, e.g., very large in comparison to the STDP window. These delays reflecting anatomical details about cortical network architecture therefore have profound effects on synaptic plasticity via STDP mechanisms (as demonstrated in the Izhikevich et al., 2004 paper). Indeed, how these delays are modeled appears to contribute to divergent results between theoretical studies (comparing the Izhikevich et al.,2004 and Morrison et al., 2007 papers). Synchronous spiking is not required to maintain structure in some STDP networks (Izhikevich 2004) yet correlated synchronous spiking can induce structure in other STDP networks (Litwin-Kumar and Doiron, 2014), according to the specific assumptions adopted. We are grateful to the reviewer for posing these questions, they seem essential. We hope these points and the data we provide will motivate further integrated computational and experimental studies of cortical local circuit dynamics.

Other suggestions:– Figure 6 top: here the cross-correlation plot based on monkey data (as in the model) exhibits nonzero lag "γ" peaks. But these seem to be absent in the data shown in the Neuron paper (e.g. Figure 4D therein). Can the authors comment on this seeming discrepancy? Is this because data only from one of the monkeys (showing stronger effects) was included in the current paper? Can the authors also comment on the flatness of the CCH in the "initial probe period" in the monkey data in contrast to the small 0-lag peak in the model CCH?

1) The absence of nonzero “γ” lag peaks in Figure 4, Neuron paper, is likely due to the fact that the cross-correlation depicted in Figure 4D is the average of the cross-correlation function over the trial time. Since the strongest spike synchrony is restricted to the end of the trial around the time of the motor response, flanking peaks in the CCH may be washed when averaging over the trial, with only the stronger, central peak remaining. In addition, restriction of the current data to the animal with the strongest spike synchrony may have contributed to this difference.

2) Flatness of the CCH in the "initial probe period" in the monkey data in contrast to the small 0-lag peak in the model CCH. This question is now addressed in Figure 6—figure supplement 1. We note that in a large (network size N≫1) and sparsely connected network (connection probability p≪1), when the population average firing rate is stationary and the network is in the asynchronous state, spike synchrony vanishes in the limit N→∞,p→0, while the average number of connections of individual neurons C=pN remains finite. The small 0-lag peak seen for the asynchronous network in panel Figure 6 A2 is due to the effect of the finite size of the network (Brunel and Hakim, 1999) in the simulations. The plot in Figure 6—figure supplement 1 shows spike synchrony obtained from the spike trains of the asynchronous steady (red circles) and synchronous oscillatory networks (blue circles) as a function of the simulated network size (from left to right, N= 20,000, 10,000, 5,000, and 3,333). The red and blue lines represent linear fits to the corresponding simulated data. It can be seen that, as the sizes of the networks increase, the synchrony of the asynchronous network extrapolates to zero in the limit N→∞, whereas for the synchronous network it remains finite.

– Lines 104-105: I found the meaning of *balance between* in this sentence vague/unclear. What does the balance between NMDA inputs and oscillatory activity mean?

We agree this was a vague statement and we have removed it from the Introduction in the revision.

– I suggest the authors also mention (e.g. around lines 476-477 where they discuss their assumption 1, and as further support for this assumption) that in their empirical data (Neuron paper) they do find an increase in PFC neural firing rates in the pre-response period, which will presumably also arise in their model from the increase in external inputs.

Thank you for the suggestion. We added a corresponding statement and referred to a new panel D in Figure 1 showing the time course of population firing rate in empirical data (line 344).

– Line 781: "themes" → "times".

Corrected (line 936).

– Line 838: I would explain where this equation comes from (I understand it accounts for the effect of reset on mean membrane potential), and also what v_α denotes.

Thank you for alerting us to this oversight. We now explain that this equation is for the average membrane potential ⟨Vα⟩. It was calculated in Renart et al., 2003, using the distribution of potentials obtained in Brunel and Hakim, 1999, and accounts for neuron spiking (i.e., the effect of reset and refractory period) and synaptic noise (line 1000).

Reviewer #3 (Recommendations for the authors):Here are issues that I believe the authors should address before the paper can be published:1. In Figure 1, the authors only show the synchrony before the motor response. I believe it would be instructive to show also what happens after motor response, to visualize how temporally precise is this increase. It would also be useful to show the time course of average firing rates. The explanation of the authors relies on an increase in external input to generate an increase in synchrony – this increase in external input should be reflected in an increase in the average firing rate as well.

Thank you for these valuable suggestions. We have now added new panels C, and D to Figure 1 that show the time courses of average firing rates. We also extended panel B as well as the new panel D beyond the time of the motor response. It can be now appreciated from Figure 1B that the peak of the synchrony modulation matches the onset of the motor response. Comparing Figures 1B and 1D one can also appreciate that, consistent with the model’s assumption about the increase in external input, the empirical firing rates gradually increase before the motor response. We have added a corresponding statement (see new text at Lines 110-115) to support our assumption about the increase in external input by referring to the increase in the empirical firing rates shown in the new panel D, Figure 1. Additionally, in a new Figure 7 we have now provided a similar comparison between the time courses of synchrony, population spike rate, and external input for the network model based on spike trains obtained in a series of new simulations in which the network received transient external inputs (see point 3 below).

2. Figure 6 shows a strong difference between the network model and data – side peaks in spike correlations are much more prominent in the model than in the data. Thus, synchronization in the data is much less oscillatory than in the model. Is there a way in the model to get such weakly oscillatory synchrony? If not, can the authors speculate as to what additional mechanisms would lead to a reduction of γ power, preserving zero lag synchrony?

In general, as one can appreciate from Figure 4 panels A2 and C2 and Figure 5B the strength of oscillatory synchrony depends on how far from the oscillatory instability boundary the synchronous network state is: the closer it is, the weaker oscillatory synchrony is, as it can be appreciated by decreasing off-zero lag peaks while the 0-lag peak remains present. This suggests that inputs to prefrontal networks in the neural data push the network just past the state boundary into the domain of oscillatory activity, whereas we consider larger excursions through the synaptic parameter space in our simulations to better characterize the full range of effects.

3. In the data, the increase in synchrony is transient. In the simulations, the author considers the case of stationary external inputs. I believe the authors should also investigate the case of transient increases of external inputs leading to a more realistic time course of synchrony. Such a scenario could also help in solving the issue mentioned in 2.

In the revised manuscript we extended our consideration to transient external inputs. To this end, we carried out a new series of simulations in which stationary external input received by network neurons was replaced with a time-varying input that had a trapezoid-like temporal profile. Results of these new simulations are presented in the Results section “Explaining the effects of blocking of NMDAR observed in primate PFC by the prefrontal circuit model,” and are summarized in the new Figure 7 mentioned above providing the time courses for synchrony and firing rate obtained from spike trains of simulated networks in response to transient external inputs. In the revised Results section, we now compare the dynamics of firing rate and synchrony and association between them observed in simulations with those observed in experiments. Figure 7 demonstrates that modulation of firing rate and spike synchrony in response to a transient input to the network exhibit several features that were evident in the recorded neural data. Namely, increases in spike rate were early and gradual following transient increase in network inputs, whereas increases in spike synchrony were delayed and abrupt in relation to transient increase in network inputs in the network simulation (Figure 7B, C; black), as we saw in the neural data (Figure 1C, D; black). We acknowledge, however, that while our relatively simple model qualitatively is consistent with dynamical features of the firing rate and synchrony observed in primate PFC, there are some quantitative discrepancies in firing rates.

[Editors’ note: what follows is the authors’ response to the second round of review.]

The manuscript has been improved but there are some remaining issues that need to be addressed, as outlined below:1. The dynamics of average firing rates in Figure 1 should mark clearly the various relevant periods of the task on the time axis: probe interval, motor response, etc.

We expanded the time axis in the left panels of Figure 1 to include the whole trial period. The probe presentation time is now indicated by green shaded area whereas the mean and standard deviation of the motor response time by color coded horizontal bars.

2. The new theoretical analyses in this revision have greatly extended the length of the main text, but the additional insight provided is not clearly explained and in its current form it can lead to confusion. It is suggested that most of this material is moved to a supplementary document and a more concise and clearly delineated section in the main text provides the fundamental message of how NMDARs influence network synchronization.

We have now moved large parts of this material, including three figures, from Results to newly added Appendices 1-5. We have also revised the remaining part in the Results section to improve the clarity of the text.

Reviewer #3 (Recommendations for the authors):The paper has been considerably improved and I would like to congratulate the authors for their hard work. There are still a couple of issues though that I think should be addressed to improve the paper further.

Thank you for the positive comments and valuable suggestions.

1. In response to one of the main concerns of the reviewers, the authors have expanded Figure 1 to include the dynamics of average firing rates in the data and the effect of NMDAR blockade on this dynamic. This is a welcome addition, but I think this figure could be improved to better understand the time course of the task. Currently, there is no discussion at all about when the motor response is happening and so the left and right panels seem totally disconnected. One has to go back to the original paper by Zick et al. to understand when the motor response happens. I would suggest extending the left panels to include the whole trial (including the `probe' interval), adding where the mean response time is similar to Figure 4 in the Zick et al. 2018 paper. Or at the very least indicate when is the mean response time (time 0 in right panels) with respect to the time axis of the left panels.

As suggested, we expanded the time axis in the left panels to include the whole trial period. The probe presentation time is now indicated by green shaded area whereas the mean and standard deviation of the motor response time by color coded horizontal bars. We also added accompanying explanation in the figure legend (lines 1491-1493).

2. In response to another concern, the authors have considerably expanded their theoretical analysis with a detailed description of an approximation for the instability growth rate. However, while the approximation itself represents an interesting new derivation, I did not find that it brings much insight, and in fact in some ways introduces additional confusion, for the following reasons. (i) The main new result of this new analysis is the new Equation (1). In this equation, the LambdaRs play critical roles and are the subject of much of the later discussion, however, their meaning is not really explained in the main text and one has to go to the Methods to understand what they really are. (ii) The new Figure (10) indicates that the growth rate is primarily determined by the AMPA term and that the GABA term plays little role. This is quite confusing as oscillations in such networks relies on GABAR mediated negative feedback. How can the authors explain this fact? (iii) After reading the new part, I ended up not having much more insight about mechanisms. My guess at this point is that NMDAR influences the synchronisation properties of the network by providing an additional excitatory drive, thereby playing a similar role as the external excitatory input. I have the feeling that is what the authors are trying to say when they say that the NMDAR term acts through changes in phiprime (the slope of the transfer function), which is exactly how the external inputs act, but it feel like it could be explained in a much clearer way.

(i) We thank the reviewer for noting that the meaning of the factors ΛAMPA, ΛNMDA, and ΛGABA have not been clearly explained in the main text. We believe that this fact also caused the confusion expressed in point (ii), which we address below. We have now revised the corresponding section of the manuscript (lines 465-477). The factors ΛAMPA, ΛNMDA, and ΛGABA govern the strength of the direct and indirect contributions of the changes in the synaptic conductances ΔgAMPA, ΔgNMDA, and ΔgGABA to the oscillatory instability. The direct contribution of the conductance change ΔgR is due to the explicit dependence of λ on the ΔgR in Equation 1. The indirect contribution is due to the implicit dependence of λ on the ΔgR in Equation 1 via the changes in the slopes of the current-frequency response functions ΔϕIsyn,E′ and ΔϕIsyn,I′ of excitatory and inhibitory neurons, respectively. These changes in the slopes occur because the change in the conductance ΔgR produces changes in the average total synaptic currents of excitatory and inhibitory neurons. These, in turn, change the operating points of the response functions and, hence, the slopes of these functions at the operating points (lines 436-443). We explain that the strength of the direct contribution of the change in the synaptic conductance ΔgR to the instability growth rate λ is determined solely by the corresponding factor ΛR via the term ΛRΔgRgR∗ in Equation 1. However, the strength of the indirect contribution of the ΔgR is determined by all three, ΛAMPA, ΛNMDA, and ΛGABA, factors via the terms involving changes in the slopes ΔϕIsyn,E′ and ΔϕIsyn,I′. In addition, we now provide a specific example for the change in the GABAR conductance ΔgGABA (lines 471-477) and explain the resulting contributions to the instability growth rate λ determined by each of the ΛAMPA, ΛNMDA, and ΛGABA factors.

(ii) Figure 9 (Figure 10 in previous version) demonstrates that contributions to the oscillatory instability growth rate λ from the terms involving ΛNMDA and ΛGABA in Equation 1 are small compared to the contribution involving the term ΛAMPA. Figure 9 also demonstrates that this does not mean that changes in the NMDAR and GABAR conductances do not affect the oscillatory instability. Indeed, one can see from Figure 9 C and D that changes in the instability growth rate λ (black line) in response to the changes in the NMDAR (panel C) and GABAR (panel D) conductances are comparable to the changes in λ in response to the change in the AMPAR conductance (panel B). This is because changes in the NMDAR and GABAR conductances, while do not affect directly, influence the oscillatory instability growth rate λ indirectly by changing the average total synaptic current (excitatory drive) and, therefore, the operating point of the neuron’s current-frequency response function. We provide this explanation in a new paragraph (lines 485-490).

(iii) The reviewer’s understanding of our explanation is correct: NMDAR influences the synchronisation properties of the network by providing an additional excitatory drive, thereby playing a similar role as the external excitatory input. To improve the overall readability and clarity of this section we have now moved some parts of it, including three figures, from the main text to newly added Appendices 1-5. We have also revised the main text to provide a more concise and clear explanation of how the NMDAR conductance and external input modulations affect synchrony (lines 506-524).